



# Imagery classification of stream stage to support ephemeral stream monitoring

Sarah E. Ogle[1], Garrett McGurk[1], Anahita Jensen[2], Fred Martin Ralph[1], and Morgan C. Levy[3, 4]

[1]Center for Western Weather and Water Extremes, Scripps Institution of Oceanography, University of California San Diego, San Diego, California, USA
[2]University of California Los Angeles, Los Angeles, California, USA
[3]Scripps Institution of Oceanography, University of California San Diego, San Diego, California, USA
[4]School of Global Policy and Strategy, University of California San Diego, San Diego, California, USA

*Correspondence to:* Sarah Ogle (seogle@ucsd.edu)

**Abstract.** Intermittent rivers and ephemeral streams (IRES) constitute a large fraction of global river networks, provide important ecosystem services, and are increasing in number with climate change. Yet, observing stage and calculating discharge in IRES can be technologically and methodologically challenging. To address this problem, we develop a method to classify relative stage categories from field camera imagery, creating a time series of categorical flow states without the need for direct stage measurements. Specifically, we employ a Logistic Regression model to classify conditions of no water, low water levels, or high water levels for an ephemeral stream located in the upper Russian River watershed of California (U.S.). We trained our algorithm using hourly field camera images from 2017-2023, and validated the image classifications with 15-minute continuous stage observations. We then used image classifications to perform quality control on the continuous stage time series. Next, we compared the image classifications to publicly accessible modeled discharge from the NOAA National Water Model CONUS Retrospective Dataset. We discuss how in-situ monitoring including field cameras and the classification of field camera imagery, combined with surface meteorology and soil moisture observations, provides detailed hydrologic information important for understanding how climate change affects IRES. Because the image classification approach is transferable to other ephemeral stream sites equipped only with field cameras, this methodology provides a low-cost option for observing relative stage on sparsely-measured IRES that can augment existing hydrologic modeling used by water managers.

## 1 Introduction

Global climate models (GCMs) agree that California (CA) will experience warming with climate change (e.g., Hayhoe et al., 2004; Leung et al., 2004; Pierce et al., 2013; Polade et al., 2014), and will face significant and variable changes in hydroclimate (Dettinger, 2016; Persad et al., 2020). Annual average precipitation projections are less certain than those for temperature (Polade et al., 2017), with GCM projections from the Coupled Model Intercomparison Project (CMIP) Phase 6 showing a wide range of possible precipitation changes for CA. Li et al. (2022) narrowed these wide-ranging projections to estimate that CA precipitation will increase by 10-34% and 7-32% by the end of the 21st century, for northern and central-southern CA,



respectively. Therein, climate change is expected to increase the proportion of annual precipitation delivered via atmospheric river (AR) storms, and extreme precipitation events from ARs are expected to become more common (Gershunov et al., 2019).

Simultaneously, warmer temperatures, which correspond to increased evaporative demand, can lead to landscape drying and less runoff generation during dry periods (Underwood et al., 2018; Albano et al., 2022).

Effects of climate change on hydrology have already been observed in CA, including warmer spring temperatures leading to earlier spring runoff (Vicuna and Dracup, 2007). Similarly, snowline elevations in the Southern Cascade and Sierra Nevada mountains are already increasing and are expected to continue rising while snow accumulation decreases (Shulgina et al.,

2023). At the same time, climate and land use change are expected to increase the likelihood of extreme flooding, which increases the probability of hydrologic failure at major dams in CA by 2100 (Mallakpour et al., 2019). These amplifications of temperature and precipitation variability impact water management, highlighting the need to develop tools to support water management adaptation (Mallakpour et al., 2019).

In particular, the prevalence of intermittent rivers and ephemeral streams (IRES) is expected to increase with climate change,

and IRES remain vulnerable to anthropogenic threats (Acuña et al., 2014; Chiu et al., 2017; Gutiérrez-Jurado et al., 2019). IRES are defined as flowing waters that stop flowing or go dry at some point along their course (Datry et al., 2017) and they represent over 50% of the world's river network and global discharge (Acuña et al., 2014; Gutiérrez-Jurado et al., 2019). Flow initiation mechanisms for IRES include saturation-excess and infiltration-excess overland flow; interflow from saturated and unsaturated soils; and groundwater flow (Gutiérrez-Jurado et al., 2019). IRES flows support a variety of ecosystem processes

and biodiversity by providing water to downstream river systems, nourishing riparian vegetation, and providing habitat for fisheries (Acuña et al., 2014). IRES and the downstream ecosystems they support are uniquely vulnerable to climate change. For example, Moidu et al. (2021) investigated the effect of climate variability on end-of-season (September or October) IRES wetting conditions for 25 streams in the lower Russian River watershed. They find that antecedent precipitation at seasonal to multiyear time scales strongly predicts end-of-season flow state alongside static landscape attributes such as geology, soil

type (Gutiérrez-Jurado et al., 2019), and degree of weathering. Monitoring in Critical Zone Observatories (CZO) elsewhere shows that the effects of climate change on IRES can be localized, as exhibited by decreasing precipitation in two observatory catchments located in France and Mali leading to different outcomes: a decrease in flow and an increase in flow, respectively (Fovet et al., 2021). Furthermore, water-borne disease pathogens can be sensitive to river intermittency, and might be expected to change with climate change and human alterations to IRES catchments (Bertassello et al., 2021).

Problematically, observing IRES systems has historically been limited and challenging, especially because many IRES are in remote areas that may be difficult to access (Magand et al., 2020). In addition to inaccessibility, developing an in-situ monitoring network for stage and discharge on IRES is difficult because nascent gage networks may have less expertise, support, or funding compared to established national programs that generally focus on perennial streams (Vlah et al., 2023). There are also many environmental challenges to monitoring IRES, including turbulent flow, sediment, and debris flow following storms (Vlah

et al., 2023). Estimating discharge from stage using a rating curve approach can be especially difficult for IRES (Vlah et al., 2023), which are often located in less accessible areas with less-developed channels. While comprehensive monitoring of IRES across a wide variety of watershed sizes and climates is important (Fovet et al., 2021), it is rarely achieved.



In the absence of comprehensive in-situ monitoring, low-cost methods for detection of IRES states (e.g., wet or dry conditions) using satellite remote sensing can be useful. For example, Tulbure et al. (2022) used Landsat-8 and Sentinel-2 to detect ephemeral floods in Australia's dryland Murray-Darling Basin and Fei et al. (2022) used Sentinel-1 and Digital Elevation Models to map alpine IRES in the Tibetan Plateau. While publicly-available satellite remote sensing can help map IRES in some instances, cloud cover and vegetation often obscure streams from view and the generally narrow water surface of IRES can be difficult to resolve.

Another low-cost method for determining IRES states is using simple sensors that record time series data to indicate the stream state. For example, a measure of high electrical conductivity indicates that a stream is wet and low (or zero) conductivity indicates that a stream is dry (Chapin et al., 2014). Observations of water occurrence from conductivity sensors have also been combined with time-lapse imagery and stage measurements to gain a broader understanding of the spatial and temporal characteristics of the stream network (Kaplan et al., 2019). Alternatively, stream temperature measurements have been used in combination with statistical models to determine the presence of water in several IRES in the northwest Great Basin desert, USA (Arismendi et al., 2017).

Citizen science approaches can also be used to obtain IRES data. For example, CrowdWater is a mobile phone application that enables installation of virtual staff (stream stage measurement) plates on streams to measure relative stage over time (Seibert et al., 2019). Users can determine the status of temporary streams using the following categories: "dry streambed", "wet streambed", "isolated pools", "standing water", "trickling water", or "flowing water" (CrowdWater, 2023). As of November 2023, CrowdWater had over 18,900 virtual staff plate contributions mainly on perennial streams, and over 19,000 temporary stream contributions. Similarly, Truchy et al. (2023) developed an app called DRYRivERs that asks users to identify the state of a river or stream as flowing, disconnected pools, or dry. Nevertheless, data acquired from citizen science approaches has its limits (e.g., variable participation, data quality) and is not always usable for science or management applications.

Because in-situ measurement of IRES is frequently not possible, modeling approaches are useful. Durighetto and Botter (2021) used a precipitation-based empirical model calibrated on field survey data to create a time-lapse visualization of IRES streamflow state in the Rio Valfredda catchment in the Italian Alps. Forghanparast and Mohammadi (2022) used a long short-term memory (LSTM) model to predict streamflow in IRES in the Texas headwaters of the Colorado River. Similarly, many contemporary studies use deep learning algorithms trained with hydrologic observations to predict streamflow in IRES and perennial streams (Kratzert et al., 2019; Le et al., 2021; Feng et al., 2022).

In-situ image-based (i.e., field camera) approaches provide another method to observe and quantify water level in IRES. Image-based approaches can provide visually verifiable image data without disrupting the stream channel, and can be used with or without water level measurement equipment. Takagi et al. (1998) identified water levels from images by finding the interface between a slanted metal strip and its refracted reflection. More recently, field studies have identified water levels from field camera images based on the color contrast of the water-air interface using methods similar to Otsu (1979-1) that segment grayscale images based on their gray-level image histogram. Leduc et al. (2018) used a time-lapse camera and methods accounting for image quality issues to identify the stage and width of a stream in Alberta, Canada, finding that their image-based estimates generally agreed with daily stage measurements from a pressure transducer. Zhang et al. (2019) found that





near-infrared imagery has greater contrast at the water-air interface even during inclement weather, making the water line easier to identify on processed images of the staff gage. Noto et al. (2022) used low-cost stage cameras and reference poles to estimate 30-minute stage with minimal error at five IRES test sites in the Montecalvello catchment of Italy. Similarly, Birgand et al. (2022) used time-lapse images of a tidal creek with a dedicated, high-contrast target background to accurately measure stage using open-source software for water level measurement (Chapman et al., 2022). While they are successful in some cases, these methods require specific equipment that make them less suitable for widespread use, and it remains that the image segmentation methods required by these approaches (Otsu, 1979-1; Leduc et al., 2018; Zhang et al., 2019; Noto et al., 2022) can struggle to function when lighting conditions are poor.

Finally, machine learning and deep learning models have been successfully applied to field camera images to identify stage, velocity, and/or discharge primarily on perennial streams. For example, Tosi et al. (2020) calculated streamflow velocities on the Brenta and Tevere rivers in Italy using a machine learning algorithm that tracks features across consecutive site images. For stream sites with reliable cellular data coverage, a cloud-based computer vision stream gauging system can use short videos from a stereo camera to adaptively learn to estimate stage, surface velocity, and discharge (Hutley et al., 2023). Gupta et al. (2022) trained a deep convolutional neural network (CNN) model to recognize relative measures of streamflow using U.S. Geological Survey (USGS) timelapse camera photos from six non-IRES monitoring locations. They found that the CNN model performs almost as well as traditionally estimated streamflow reliant on manual discharge data. Windheuser et al. (2023) used deep neural network models trained on both images and time series data, including precipitation and USGS gage height, to predict flood stage for two rivers in Georgia (U.S.), showing the strength of combining machine learning with other data sources. Using image processing and deep learning for streamflow prediction is an emerging field that has mainly focused on perennial streams. Gupta et al. (2022) and Noto et al. (2022) highlight the need to create algorithms focused on IRES.

Here, we explore the use of image classification for categorizing water levels in IRES. This research is motivated by data limitations common to IRES networks, the increasing availability of low-cost field camera equipment and imagery, and scientific interest in improving IRES monitoring due to their unique ecological value and sensitivity to climate change. We first introduce our study area: a headwater stream in the upper Russian River watershed, CA. Then, we describe methods for using a combination of machine learning and field camera imagery to classify water levels on IRES. Lastly, we evaluate the performance of the image-trained machine learning model and demonstrate the value of the approach for monitoring and understanding IRES in California and elsewhere.

## 2 Methods

### 2.1 Study site and data

The Center for Western Weather and Water Extremes (CW3E) at the Scripps Institution of Oceanography installed a hydrologic and meteorological sensor network during the fall and summer of 2017 in the upper Russian River watershed, CA (fig. A1; Sumargo et al., 2021; Ralph et al., 2022). This network supports a reservoir operations strategy called Forecast Informed Reservoir Operations (FIRO) at Lake Mendocino, an impounded reservoir used for drinking water, flood control, hydroelectric



generation, and recreation (Jasperse et al., 2020). The upper Russian River watershed is a rain-dominated basin characterized by a variable Mediterranean climate with warm, dry summers and cool, wet winters where atmospheric rivers can result in heavy rainfall and flooding (Sumargo et al., 2021). The mountainous portion of the watershed that drains into Lake Mendocino is composed of Mesozoic Franciscan Formation bedrock and is a lightly-managed combination of shrub/scrub, forest (deciduous,

evergreen, and mixed), and herbaceous land cover (USGS, 2023); in contrast, the alluvial valleys are largely cultivated or developed (Cardwell, 1965; USGS, 2023).

We focus our image classification exercise on imagery obtained at one of six continuous streamflow monitoring sites in the upper Russian River watershed (fig. A1), located in the 7.05 km$^2$ Perry Creek sub-watershed (fig. 1). We also use a nearby surface meteorology site, Deerwood (DRW in fig. 1). The Perry Creek watershed is primarily composed of Franciscan

sandstone with steep slopes that are prone to landslides (Delattre and Rubin, 2020). Perry Creek is an ephemeral stream at the stream site (PEC), which is located in a narrow, steep gorge ~20 m downstream from a ~5 m tall waterfall (fig. A2) and just upstream of Lake Mendocino. The PEC site was chosen for this study because it had potentially erroneous measurements and noisy stage measurements relative to nearby sites, making the field camera imagery useful for quality control of stage data.

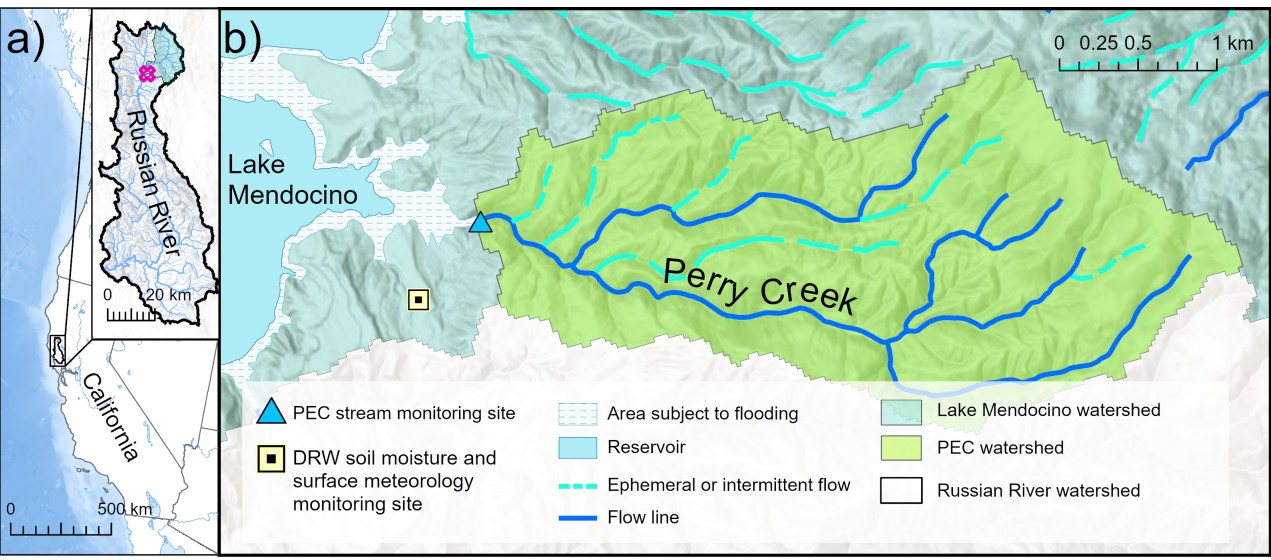

**Figure 1.** The Perry Creek (PEC) study site in the upper Russian River's Lake Mendocino watershed, California, U.S. a) The location of the Russian River watershed (inset); the pink polygon shows the location of the study area shown in b. b) Perry Creek watershed and the Center for Western Weather and Water Extremes (CW3E) monitoring sites: the stream monitoring site at Perry Creek (PEC), the soil moisture and surface meteorology site at Deerwood (DRW), and associated hydrologic features (legend). Image data sources: CW3E and National Hydrography Dataset Plus (NHDPlus) High Resolution (Moore et al., 2019; ESRI).





The PEC site is equipped with a staff plate and a stilling well containing a Solinst Levelogger (18 August 2017-11 October 2023) or HOBO MX2001-04-SS-S pressure transducer (11 October 2023-current) that record water levels at 15-minute intervals (fig. 2). There is missing data due to the inability to perform site maintenance because of the COVID-19 pandemic from 31 August 2020-24 February 2021. Atmospheric pressure from the DRW meteorological monitoring site was used to barometrically compensate stage at PEC measured by the Solinst Levelogger until the installation of a Solinst Barologger at PEC in January 2022 (Appendix A1; figs. A3, A4). While DRW is not in the PEC watershed, it is less than 1 km from PEC and has continuous 2-minute observations for the entire study period. We also use DRW's observations of precipitation, relative humidity, temperature, and soil moisture (5 cm, 10 cm, 15 cm, 20 cm, 50 cm, and 100 cm) for August 2017-November 2023. In October 2023, telemetry was installed at PEC, which includes a HOBO MX2001-04-SS-S pressure transducer and a HOBO MicroRX data logger, providing near real-time stage measurements. Prior to the installation of a HOBO telemetry system, stage data was downloaded directly from Solinst Leveloggers. The stilling well and staff plate configuration were not altered at the time of HOBO telemetry installation.

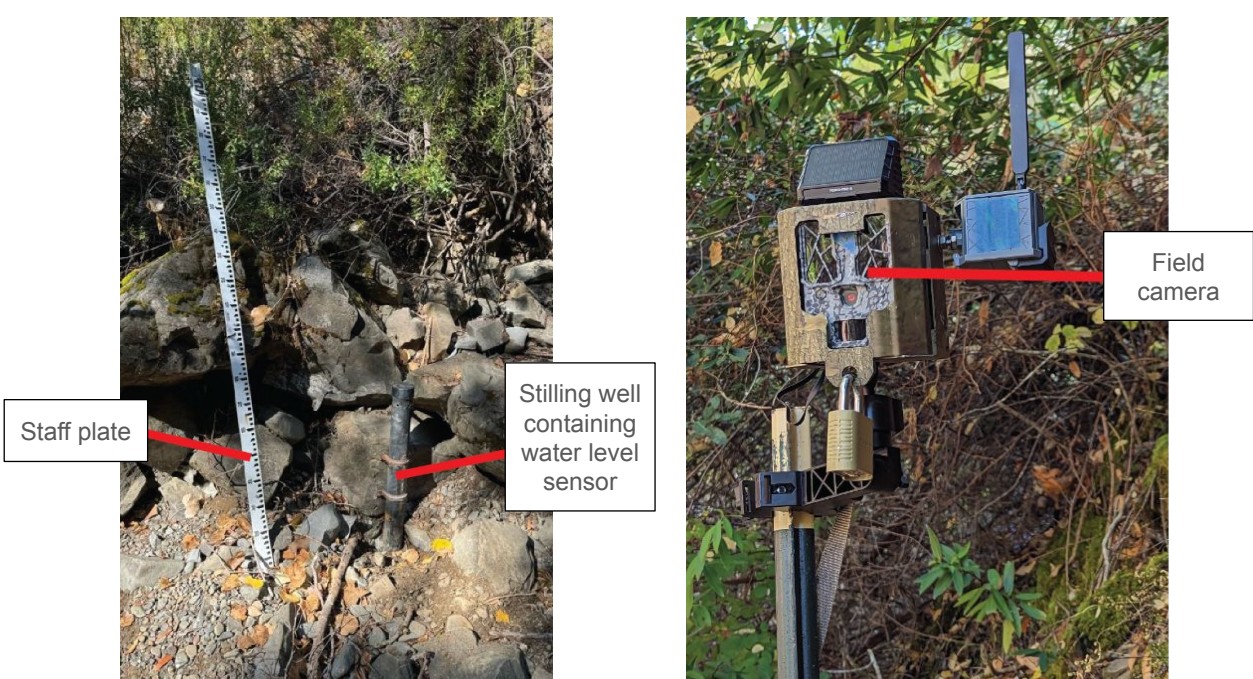

**Figure 2.** Perry Creek (PEC) streamgage station equipment. The staff plate, stilling well, and water level sensor (left) are located at the side of the streambed; the field camera (right) is located across the stream channel, facing the staff plate.



The PEC site is also equipped with a trail field camera: a Wingscapes TimelapseCam Pro (6 November 2017-23 May 2018), Spypoint Link S Dark (27 January 2022-10 October 2023), or Force-S-Pro (11 October 2023-May 2024) which take photos of Perry Creek at hourly intervals (30-minute intervals for 11 October 2023-May 2024; fig. 2). The Wingscapes field camera only took photos from 5 am-6 pm PST daily, while the others took photos at all hours. Field cameras can malfunction due to environmental damage; at PEC, the cameras were offline during June 2018-January 2022, June 2022-July 2022, and March 2023-September 2023. The Windscapes TimelapseCam Pro trail camera was prone to having its timestamp drift between field visits by as much as an hour, but we determined that these images were still likely representative of the approximate stage since flow states within an hour are sufficiently correlated. Each field camera image captures the staff plate, streambed, and surrounding environmental features such as vegetation and rocks (fig. A3). During occasional site maintenance, a handful of photos included people or were not focused on the streambed; we did not remove any of these from the dataset. Nighttime images are poorly illuminated, with the camera's flash overexposing nearby vegetation and slightly illuminating the staff plate, making it difficult – if not impossible – to discern streambed conditions at night. Field camera images were downloaded manually during routine site maintenance once or twice each year. Site servicing also included clearing brush to prevent the camera from having an obstructed view of the stream.





**Figure 3.** Examples of imagery preparation steps for example field camera images. a) Spypoint Link S Dark image for Perry Creek (PEC) from 1 December 2022, during a 'no water' period; the image is cropped to center on the staff plate and set to a size of 1,000 x 1,200 pixels. b) The four categories of labels for field camera images, showing each water level condition: 'No Water', 'Low Water', 'High Water', and 'Obstructed'. c) Examples of the grayscale conversion of b). d) The corresponding visualization of the histogram of oriented gradients (HOG) transformations of c).



Stream channel surveys and manual discharge measurements were collected from late 2017 through March 2024 at PEC.
Manual discharge measurements were performed periodically, with a focus on capturing the full range of flow conditions.
Manual discharge measurements were performed using a handheld flow meter (Pygmy, AA, or Hach MF Pro) and top-setting
wading rod at discrete intervals along a measuring tape placed perpendicular to the streamflow (Turnipseed and Sauer, 2010).
These discharge measurements were taken near the staff plate, with the staff plate level noted for data validation. As of water

year 2024, there were 12 manual discharge measurements at PEC, which is not sufficient to create a stable rating curve for
converting stage to discharge. Nevertheless, these measurements are still useful for beginning to constrain the magnitude and
range of discharge values at PEC.

Finally, we use the NOAA National Water Model (NWM) Retrospective Version 2.1 dataset, a retrospective simulation from
February 1979-December 2020 using version 2.1 of the NWM. This version uses land surface modeling based on the NOAH-

MP Land Surface Model and meteorological inputs from the Office of Water Prediction's Analysis of Record for Calibration
(NOAA, 2024a). We compare hourly NWM discharge from a segment that overlaps the location of PEC (fig. A5; reach ID:
8268225) to PEC imagery, stage, and manual discharge data. We chose the NWM for this comparison because it is a publicly-
available dataset that is accessible to water managers and available for streams across the entire U.S. There is also precedent
for using the NWM in combination with machine learning in the Russian River watershed as in Han and Morrison (2022) who

used a LSTM to correct hourly NWM discharge predictions.

In summary, the following stage, discharge, and imagery data were available at PEC from 18 August 2017-30 November
2023: continuous 15-minute or 5-minute stage observations for 91.3% of the study period (with occasional missing data during
site maintenance and prolonged missing data from 31 August 2020-24 February 2021), continuous hourly NWM-modeled
discharge from August 2017-November 2020, 11 manual discharge measurements mainly from 2018, and 15,821 field camera

images. Available field camera images documented the periods of November 2017-May 2018; February 2022-May 2022;
August 2022-February 2023; and October 2023-November 2023.

## 2.2    Classifying water levels in field camera images

We used a supervised machine learning image classification approach to identify when there was no water, low water levels,
high water levels, or an obstructed view at PEC using field camera images from August 2017-November 2023. The image

classification approach involves multiple steps: image preparation; machine learning model selection, training, and evaluation;
the development of measures to assess classification confidence; and a comparison of resulting classifications with observed
stage, manual discharge measurements, modeled discharge, and hydrologic observations.

### 2.2.1    Image preparation

We only used images taken between 9 am and 4 pm Pacific Standard Time (UTC-08:00) because low-light images are more

difficult to classify, even with a flash. For ease of classification, we cropped the images to focus on the bottom of the staff
plate and the streambed, each to the same size of 1,000 × 1,200 pixels (fig. 3a). This cropping largely removed the effect of
seasonal changes in vegetation that dominates the broader field of view and can make classification more difficult. Since the



field cameras were reinstalled during some field visits at slightly different locations along the bank opposite the staff plate, there were four slightly different viewing angles of the streambed in the images. Cropping the images helped minimize the

effect of these different angles. After cropping, we extracted the date the image was taken and the array of RGB pixel values from each image. The image preparation steps were performed using Python scripts with the sci-kit learn (Pedregosa et al., 2011c) and sci-kit image packages (Van der Walt et al., 2014).

We then labeled a subset of images for supervised learning. Of the 4,177 total PEC images taken from 9 am to 4 pm during August 2017-November 2023, we manually labeled 537 images with four distinct categories: 'no water', 'low water' level,

'high water' level, and 'obstructed' (fig. 3b). We selected more than half of these 537 images using random selection and selected the remaining images manually using visual inspection to attain a representative sample across the four categories. This resulted in the purposeful selection of a labeled data set within which there were different numbers of images across categories, i.e., samples were "unbalanced" across categories. A small subset of images show stream states from similar time periods (i.e., dates and times are not regularly sampled), but because illumination and stream stage can vary by the hour even

within the same day, we determined that the irregular time sampling should not bias model evaluation given the sample size.

Our objective was to prioritize the classification of water presence or absence – a defining feature of IRES, by exploiting the visual differences in water surface texture and color for the 'low' and 'high' water categories. Firstly, we manually labeled images that consisted primarily of branches and leaves blocking the view of the streambed as 'obstructed' even if we could discern the stream state. Images with some foliage present, but where the majority of the streambed was visible, were not

labeled as 'obstructed', and were assigned a water level label. We then assigned the 'low water' category to images with any presence of water, for which the water surface was characterized by low turbidity (i.e., clear water such that the staff plate is often reflected in the water) and the presence of little to no riffles. We assigned the 'high water' category to images according to the presence of riffles, or even rapids, often with higher turbidity indicated by the water color being a light brown compared to the typically clear water. We assigned the 'no water' category to images that lacked pooling or flowing water in the streambed.

To maintain independence of categories, each of the 537 images were assigned only one category even if they were visually similar to multiple categories (e.g. relatively clear water with some ripples).

We processed the images to minimize the effects of different amounts of sunlight by converting the color images to black and white (fig. 3c), increasing their contrast and flattening them using the histogram of oriented gradients (HOG) transformation (Lowe, 2004; Dalal and Triggs, 2005; fig. 3d), and scaling their pixel values using the Python scikit-learn package (Pedregosa

et al., 2011c). These transformations decreased color differences due to the time of day and season while making the water-air interface easier to distinguish.

### 2.2.2 Machine learning model

Within an individual model training and testing run, we randomly split the 537 labeled images into 70% training data (375 images) and 30% testing data (162 images). Figure 4 shows an example of this train-test split and demonstrates the unbalanced

proportion of labeled images within different categories. We used the 375 training images to fit a logistic regression model using the Python scikit-learn package (Pedregosa et al., 2011c). The logistic regression model is a multinomial log-linear



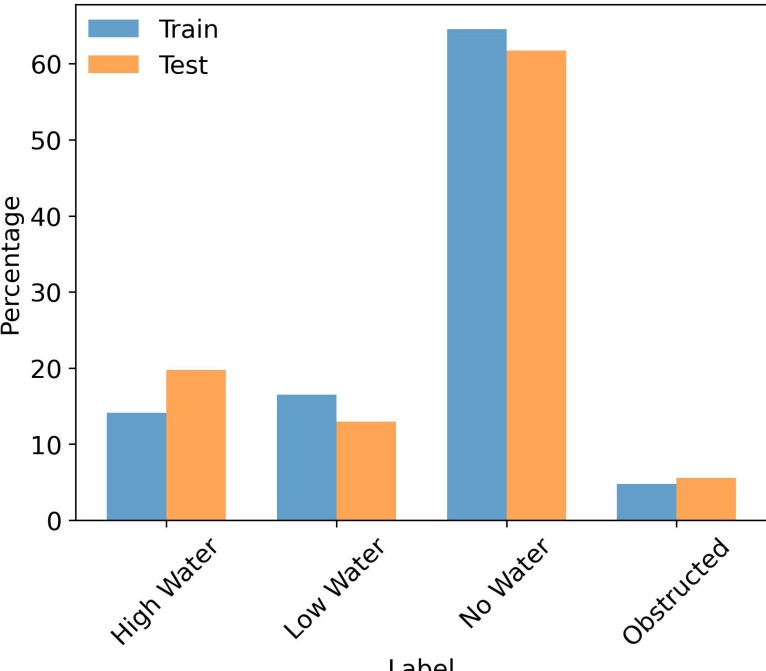

**Figure 4.** Percentage (vertical axis) of labeled Perry Creek field camera images within each of the final model run's training (70%) and testing (30%) set split (colors) according to the label category (horizontal axis).

regression model typically used for categorical classification. We chose a multinomial classification model instead of a binomial classification model (i.e., for the presence or absence of water) in order to account for the 'obstructed' view images, and to classify different flow states with distinct visual differences: 'low water' versus 'high water'. In this model, the classification

of 'high water', 'low water', 'no water', or 'obstructed' is determined by a linear combination of variables that represent the value and orientation of the pixels in each image.



We selected the logistic regression model as the classification model in part due to its estimation of a probability for each categorical classification, where the estimated probability for a category is a function of the pixel values from the prepared image (see Image Preparation). We used a standard scikit-learn package implementation of a multinomial logistic regression model with L2 regularization (for mathematical details, see Pedregosa et al., 2011a). For model fitting, we used a cross-validation (CV) routine with six stratified "K" folds, meaning the training data were internally split into K (six) different training and validation sets that preserved the percentage of images within each labeled category.

We used 20 randomly-selected training and testing splits from the labeled image dataset to fit the model for six model "configurations". Each configuration combined one of three different category weights and two different solvers, resulting in a total of 120 model training and testing runs. The three category weights were: no category weights; balanced category weights (due to our unbalanced data); and manually determined weights that emphasized the water level categories. For manual weights, the 'high water' and 'low water' categories were given weights of 3.5, 'no water' was given a weight of 1, and 'obstructed' was given a weight of 3. The two solvers were lbfgs and newton-cg; a third solver, saga, was unable to converge at the selected maximum number of 300 iterations (and up to 1,000 iterations). All remaining model parameters were set to the default scikit-learn package values (see Pedregosa et al., 2011b).

### 2.2.3 Model performance

Each of the $n$ test-set image classifications falls into one of four categories: 'no water', 'low water', 'high water', or 'obstructed'. For each category, the classification outcome is recorded as true positive (TP), false negative (FN), false positive (FP), or true negative (TN). Consider the 'no water' category as an example. If a test-set image labeled as 'no water' is correctly classified as 'no water,' it is recorded as a TP; if misclassified as 'low water,' 'high water,' or 'obstructed,' it is a FN. In contrast, if a different image labeled as anything other than 'no water' is classified as 'no water,' it is an FP; if classified correctly as any category other than 'no water,' it is a TN. We used a confusion matrix to assess these outcomes.

To evaluate the performance of different model configurations, we computed key statistical metrics for the test set results across the 20 model runs within each of the six model configurations. Specifically, we calculated the mean, standard deviation, maximum, and minimum prediction accuracy (Eq. 1). To assess accuracy across categories, we computed the mean balanced accuracy (Eq. 2), defined as the average of recall (Eq. 3) for each classification category $l = 1, ..., L$. This approach ensures that model performance is not dominated by the majority category — "no water". Next, we identified the best-performing model configuration based on the highest mean prediction accuracy and mean balanced accuracy across the six model configurations. Once selected, we used this model configuration to run the model with a fixed random seed for the train-test split and cross-validation routine, ensuring static parameterization. We refer to this as the "final model." Then, this final model was used to classify the unlabeled images. Although this final model produces identical results for repeated runs on a given operating system, slight variability may occur across different operating systems due to differences in floating-point precision and parallel processing. To evaluate final model performance, we used the test set to calculate the prediction accuracy (Eq. 1), balanced accuracy (Eq. 2), recall (Eq. 3), precision (Eq. 4), and F1 score (Eq. 5) (Pedregosa et al., 2011c).





$$\text{Prediction Accuracy} = \frac{\text{TP}}{n} \tag{1}$$

$$\text{Balanced Accuracy} = \frac{\sum \text{Recall}_l}{L} \tag{2}$$

$$\text{Recall} = \frac{\text{TP}}{\text{TP} + \text{FN}} \tag{3}$$

$$\text{Precision} = \frac{\text{TP}}{\text{TP} + \text{FP}} \tag{4}$$

$$\text{F1} = \frac{2 \cdot \text{TP}}{2 \cdot \text{TP} + \text{FP} + \text{FN}} \tag{5}$$

### 2.2.4 Classification confidence

We developed confidence levels (i.e., high, moderate, and low) for classifications of 'no water' and 'any water', where 'any water' includes images classified as either 'low water' or 'high water'. To do this, we used test set images classified as 'no water' or 'any water' from the 20 model runs with the best-performing model configuration. From these data, we established confidence levels by comparing classification probability distributions to classification outcomes (TP, FN, FP, or TN) for the 'no water' and 'any water' categories. We defined "confidence levels" using probability value thresholds determined by visually and qualitatively assessing divergence in the distributions (boxplot medians and whiskers) of the estimated classification probabilities for 'no water' and 'any water' across the TP, FN, FP, and TN classification outcomes.

### 2.3 Comparing image classifications to observed and modeled hydrology

To understand how the classified imagery from the final model relates to stage and discharge times series, we compared our classified imagery to coinciding PEC stage, manual discharge measurements, and NWM discharge. This comparison is intended to help us understand to what extent classified field camera imagery can improve the often-limited and variable quality of observational and modeled stage and discharge data in IRES systems. We do this comparison graphically by overlaying time series plots of the barometrically compensated PEC stage with the classifications of concurrent, high-confidence image classifications. This time series plot allowed us to visualize discrepancies between image classifications and stage that enabled quality control of stage data (Appendix A2). The quality-controlled, barometrically compensated PEC stage data were then used for all subsequent analyses.

We compared the quality-controlled stage data to the NWM discharge and manual discharge measurements by overlaying each time series with corresponding high-confidence image classifications. We also use boxplots to visualize the ranges of




stage and NWM discharge that correspond to each category of medium- and high-confidence image classifications. Using a
scatter plot stratified by image classification category, we compared the PEC stage to NWM discharge and manual discharge
measurements to determine the magnitude of discrepancy between data sources. For example, we calculated how often the
observed stage was zero while the NWM had non-negligible flow — discharge greater than 0.028 cubic meters per second
($m^3 s^{-1}$). This threshold was chosen because NWM values greater than zero but less than 0.028 $m^3 s^{-1}$ were considered
negligible.

## 2.4 Environmental conditions

Finally, we compared our image classifications to surface meteorology and soil moisture observations from DRW to under-
stand the concurrent and antecedent hydrologic conditions associated with 'high water', 'low water', and 'no water' image
classifications. We computed the daily total precipitation along with the daily mean temperature, daily mean relative humidity,
and daily mean soil volumetric water content (VWC) at multiple depths (5 cm, 10 cm, 15 cm, 20 cm, 50 cm, and 100 cm).
We then graphically compared these daily observations to the time series of PEC stage that corresponded to high-confidence
image classifications. Using scatterplots and linear regressions of those visualized data, we evaluated the relationship between
15-minute stage observations stratified by medium- and high-confidence image classification and the following variables: the
daily rolling mean soil VWC at 5 cm and 100 cm; the rolling sum precipitation at daily and monthly time scales; 2-minute
mean temperature; and 2-minute mean relative humidity. We compare the 15-minute stage to aggregated (rolling mean) values
of soil moisture and precipitation to represent antecedent moisture conditions, but compare the same stage measurements to
non-aggregated (2-minute) temperature and relative humidity to represent concurrent weather conditions.

## 3 Results

### 3.1 Model performance

The model configuration with the lbfgs solver and balanced category (class) weights performed best across the 20 model runs,
with a mean prediction accuracy of 91% and a mean balanced accuracy of 78% (Table 1). In general, the mean balanced
accuracy was lower than the mean prediction accuracy for each model configuration due to lower recall for the 'obstructed'
category compared to the other categories, resulting from a smaller training and test set size for the 'obstructed' category.

Within the 20 runs of the best-performing model configuration (lbfgs solver and balanced category weights), the minimum
prediction accuracy was 85% and the maximum prediction accuracy was 96%, with a standard deviation of 2.7% (Table 1).
Classification outcomes across these 20 model runs are shown in parentheses in figure 5, which lists the range of counts of
correct and incorrect classifications for individual labels. For example, 'obstructed' images are misclassified as 'high water'
images a maximum of 7 times within any individual model run, but 'high water' images never get classified as 'obstructed'.
Figure 5 also illustrates the data used to compute summary statistics of model performance. The most common incorrect
classification within these 20 model runs was the classification of 'low water' images as 'no water', which occurred in 2.8% of





all classifications. In addition, 'obstructed' images were occasionally misclassified as either 'high water' or 'no water' (1.1%

and 1.8% of these classifications, respectively). Due to the stream's ephemerality, it is likely there was in fact no water at PEC in

the 'obstructed' images misclassified as 'no water'. Similarly, 2.4% of image classifications were classifications of an incorrect

water level (i.e., a 'low water' classification when the image's label was 'high water' and vice-versa). Because there is some

overlap in image features that distinguish 'low water' and 'high water' images, incorrect water level magnitude classifications

are expected, but do not detract from the utility of the model for prediction of IRES water presence or absence. The standard

deviation of prediction accuracy across these 20 model runs was low (0.027, Table 1), indicating that any individual model run

would produce similar image classifications when applied to the unlabeled images.

**Table 1.** Model configuration accuracy metrics across the six model configurations. Model configurations include the combination of solver and category weight options (rows), and accuracy metrics (columns) include the mean, standard deviation, maximum, and minimum for prediction accuracy along with mean balanced accuracy for the 20 model iterations (different random training and testing data splits) within each configuration. The first row shows the best-performing model configuration.

| Solver | Category weight | Mean accuracy | Std. dev. of accuracy | Max. accuracy | Min. accuracy | Mean balanced accuracy |
|--------|-----------------|---------------|----------------------|---------------|---------------|------------------------|
| lbfgs | Balanced | 0.91 | 0.027 | 0.96 | 0.85 | 0.78 |
| lbfgs | None | 0.90 | 0.027 | 0.96 | 0.85 | 0.70 |
| lbfgs | Custom | 0.90 | 0.027 | 0.96 | 0.85 | 0.72 |
| newton-cg | Balanced | 0.90 | 0.030 | 0.96 | 0.83 | 0.76 |
| newton-cg | None | 0.89 | 0.026 | 0.94 | 0.85 | 0.70 |
| newton-cg | Custom | 0.90 | 0.027 | 0.96 | 0.85 | 0.72 |

To run the final model, we used a static parameterization of the best-performing model configuration — lbfgs solver and

balanced category weights. To illustrate the final model's performance within and across individual label categories, we used

a confusion matrix (Figure 5) to show classification outcome counts from the final model run. Table 2 shows accuracy metrics

by image category for the final model run, showing greater than 0.80 for precision, recall, and F1 scores across all categories

except for the recall and F1 scores for the 'obstructed' category. Supplementary figure A6 shows an example of test set output

for the final model.




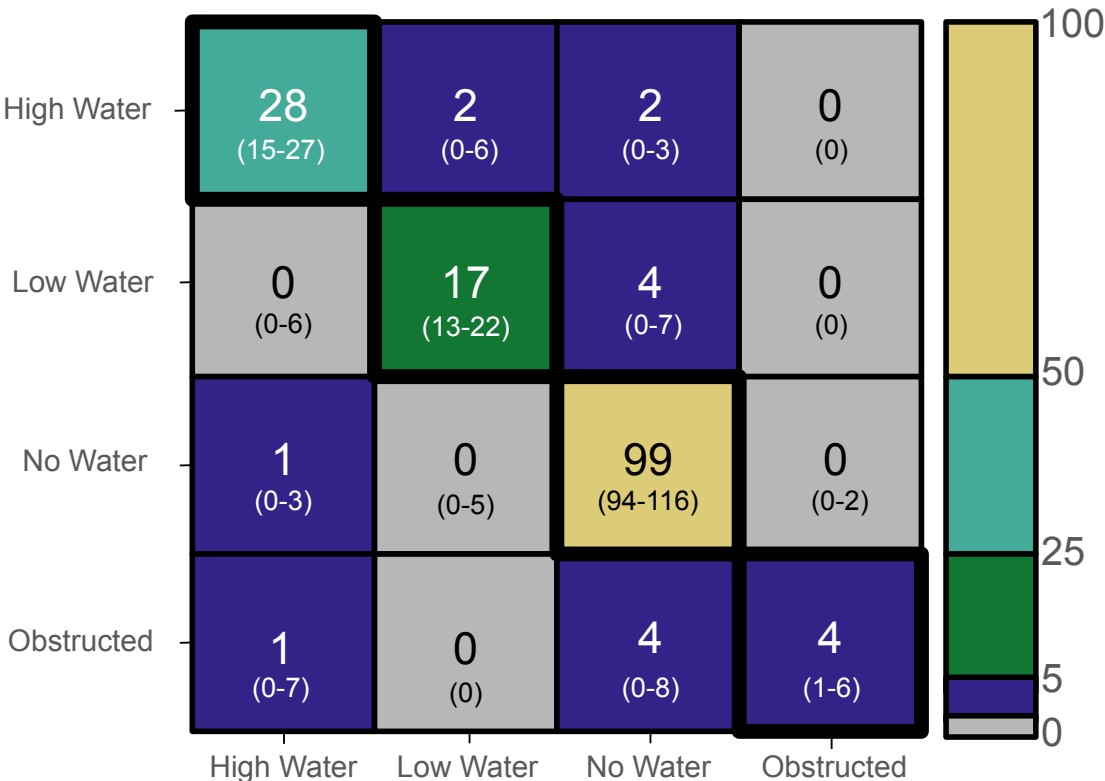

**Figure 5.** Confusion matrix showing test set classification counts from the final model run. The matrix displays classification counts from the final model run (colors, centered text) and the range of classification counts across the 20 model runs (parentheses) using the best-performing model configuration —lbfgs solver and balanced category weights.

**Table 2.** Precision, recall, and F1 scores for the final model run.

| Category | Metric | | |
|---|---|---|---|
| | Precision | Recall | F1 score |
| High water | 0.93 | 0.88 | 0.90 |
| Low water | 0.89 | 0.81 | 0.85 |
| No water | 0.91 | 0.99 | 0.95 |
| Obstructed | 1.00 | 0.44 | 0.62 |





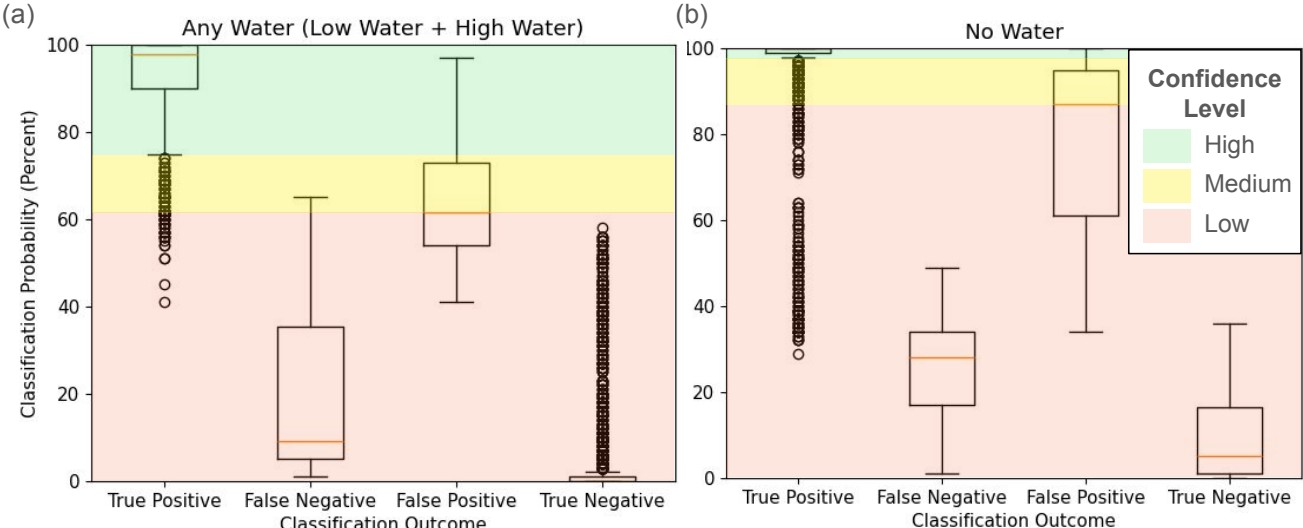

**Figure 6.** The distribution of classification probabilities from the 20 runs of the best-performing model configuration evaluated on test set images. The panels show classification probabilities (vertical axis) across true positive, false negative, false positive, and true negative classification outcomes (horizontal axis) for: a) 'any water' ('low water' or 'high water') classifications and b) 'no water' classifications. The colors indicate the probability ranges used to assign high (green), medium (yellow), and low (red) confidence to image classifications. The boxplots show the interquartile (IQR) range (box), median (orange line), the upper/lower quartile +/- 1.5 * IQR (whiskers), and outliers (points).

## 3.2 Classification confidence

We define levels of classification confidence based on assessment of the distributions (fig. 6) of classification probabilities from the 20 model runs of the best-performing model configuration across TP, FN, FP, and TP classification outcomes. To assign levels of confidence, we chose thresholds in classification probability distributions that prioritized minimizing the risk of a false classification. To do so, we compared the distribution of true positive probabilities to false positive probabilities; we did not use false negative probabilities in the assessment of confidence because the probability of false negative outcomes is inherently

much lower than the false positive probability for images with 'no water' and 'any water' labels (fig. 6). We also did not use true negative probabilities in determining classification confidence because a true negative outcome represents any label other than the one considered.



We assigned a 'high' confidence level (fig. 6, green) if the classification probability fell within the range of probabilities greater than the lower whisker for true positive classifications; this captures the full range of probabilities (excluding outliers) for true positive outcomes while not overlapping with the interquartile range for false positive outcomes for both the 'no water' and 'any water' categories. We assigned a 'medium' confidence level (fig. 6, yellow) if the classification probability fell within the range of probabilities less than or equal to the lower whisker for true positive classifications and greater than the median for false positive classifications; this captures some of the outlier probabilities for true positive outcomes, but recognizes that some probabilities in this range may be false positive classifications. Finally, we assigned a 'low' confidence level (fig. 6, red) if the classification probability was less than or equal to the median of false positive classifications. For the final model's classification of unlabeled images, we evaluated each image classification's model-estimated probability according to these threshold values to provide categorical high, medium, and low confidence assignments for each classification. We subsequently use these confidence assignments, alongside their image classifications, to evaluate correspondence between water level classifications and modeled and observed water levels.

## 3.3 Comparing image classifications to observed and modeled hydrology

The medium and high (fig. 7) confidence image classifications generally agreed with the PEC stage data. For example, a total of 94.6% of images classified as 'any water' corresponded to stage greater than zero, while 99.9% of images classified as 'no water' corresponded to a stage of zero. Low confidence image classifications did not agree nearly as well; 61.5% of low confidence classifications identified 'any water' when stage was greater than zero, and 56.7% identified 'no water' when stage was zero. The median stage for medium- and high-confidence image classifications for 'high water' (27.1 cm) is greater than the median stage for images classified as 'low water' (19.1 cm; fig. 8a). In addition, the images classified as 'high water' contain all but two occurences of stage above the 99th percentile (46.1 cm, table A1) at PEC (fig. A7a). 75.1% (63.2%) of images classified as high (low) water level corresponded to stage above (below) the 90th percentile at PEC, which is equivalent to a stage of 21.5 cm (table A1).







**Figure 7.** Original and quality controlled, barometrically compensated stage from the Perry Creek (PEC) site from December 2022-February 2023. Stage values (black lines) are colored (points) by high-confidence image classifications. a) Shows the time series before quality control and b) shows the time series after quality control.





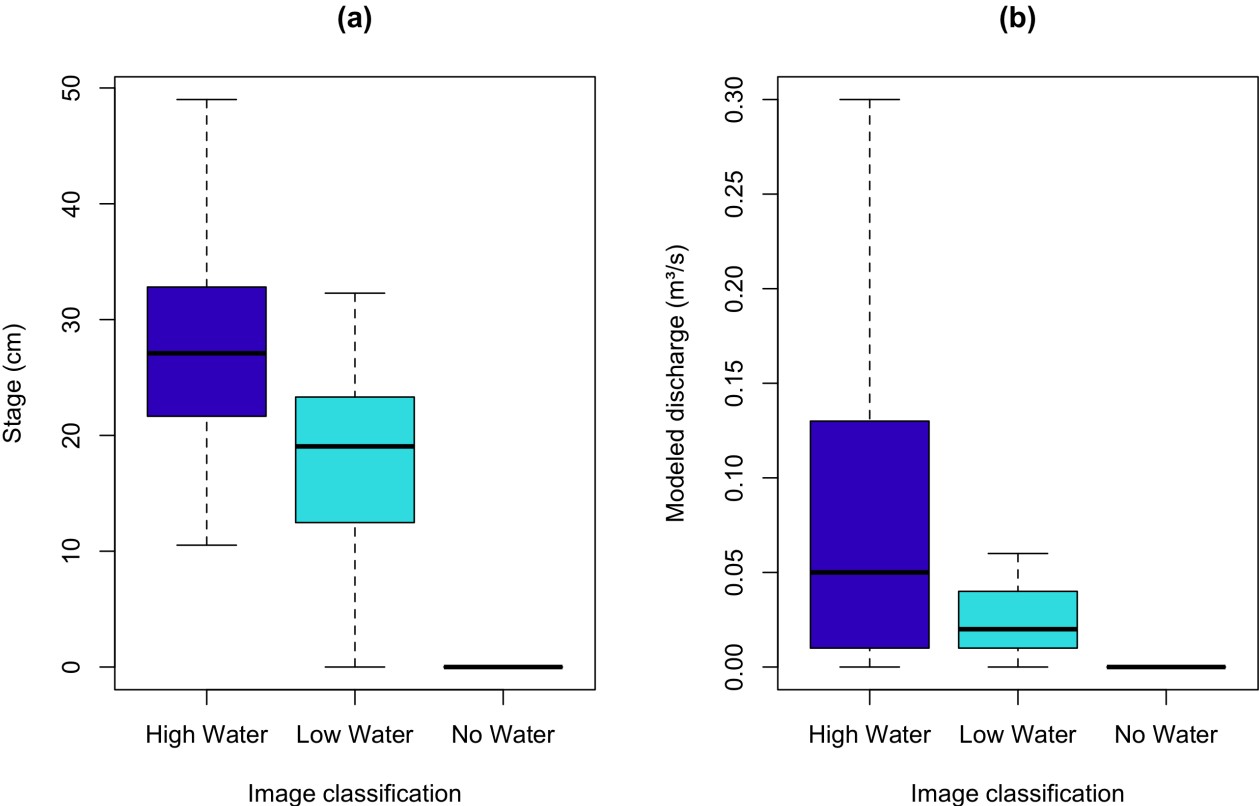

**Figure 8.** Distribution of Perry Creek (PEC) stage and modeled discharge for medium- and high-confidence water level classifications. The boxplots show all values of stage (a) and modeled discharge (b) at PEC (vertical axis) corresponding to medium- and high-confidence (only) classifications of images as high, low, and no water (horizontal axis). The boxplots show the interquartile (IQR) range (box), median (bold line), and the upper/lower quartile +/- 1.5 * IQR (whiskers); outliers were excluded from boxplots but not from the calculation of boxplot statistics (see figure A7 for boxplots with outliers).



The final corrected and quality-controlled PEC stage time series (fig. 9) is the product of manually removing erroneous stage observations determined using image classifications, as well as standard quality control (i.e. removal of stage observations taken during sensor maintenance). By comparing images classified as 'no water' and the corresponding observed stage (visually), we determined that stage measurements oscillating near zero were noise instead of short-duration flow events; we then set those measurements to zero. Similarly, we used the stage time series and the image classifications of 'high water' images to identify

(visually) stage measurements that were erroneously low, and subsequently removed those measurements. For example, many of the images from January 2023 were classified as 'high water'; however, the corresponding stage values were near zero or below zero (fig. 7a). This indicated sensor measurement error that was likely related to turbulent flow conditions or debris from high flows; we flagged and removed these data from the record (fig. 7b). Similarly, in late 2017 through early 2018, images were classified as 'high water' while the observed stage was low (fig. 9). This identified a time period during which

the stage sensor was pulled upwards in the stilling well sleeve due to debris catching on the exposed data cable, according to a subsequent site inspection. We chose not to remove the late 2017 through early 2018 stage data, but acknowledge that the stage is likely underestimated: the image classifications provided evidence that peak stage during this time may have been higher than recorded by the (compromised) stage sensor. For more details, see Appendix A2.



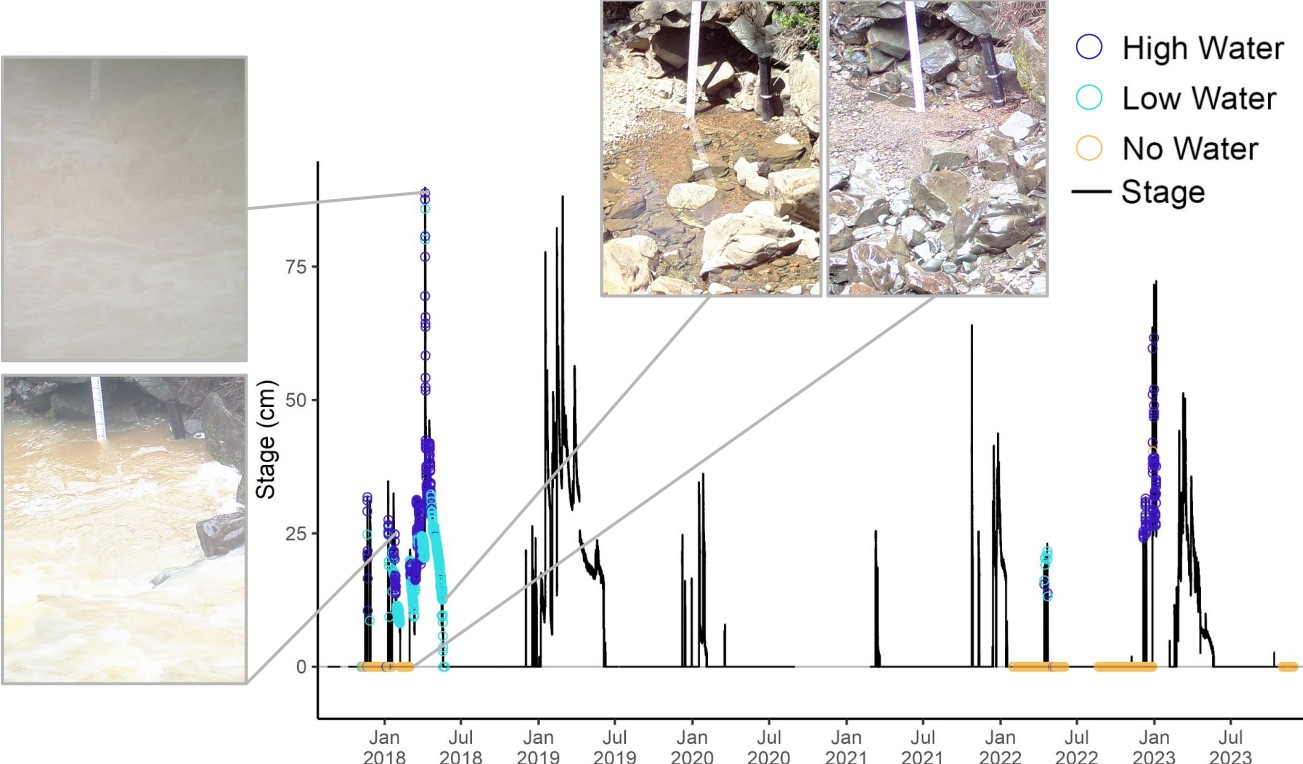

**Figure 9.** Barometrically compensated and quality controlled stage from the Perry Creek (PEC) site from August 2017 - November 2023. The time series shows PEC stage (vertical axis) across the full study period (horizontal axis) colored by concurrent high-confidence image classifications (colored points). Four example field camera images are shown. Field camera images were not available from 2019-2021.



We performed a similar comparison of classified images with modeled discharge data from August 2017-August 2020. The
median modeled discharge for images classified as 'high water' (0.05 $m^3 s^{-1}$) is greater than the median modeled discharge
for images classified as 'low water' (0.02 $m^3 s^{-1}$; fig 8b). In addition, the images classified as 'high water' contain most of the
extreme flow events, indicated by the 'high water' classification containing almost all of the outliers (fig. A7). 67.9% (90.2%)
of images classified as high (low) water corresponded to modeled flow above (below) the 90th percentile (0.04 $m^3 s^{-1}$). All
images classified as 'no water' corresponded to flow less than the 80th percentile (0.02 $m^3 s^{-1}$), but approximately 18.2% of
images classified as 'no water' have non-zero modeled discharge, albeit near-zero.

Instantaneous manual discharge measurements provide additional context (figs. 10 and 11). There are several instances in
which modeled discharge peaks correspond to PEC stage peaks (April, 2018; fig. 10). In contrast, modeled discharge is zero
while the observed stage is greater than 10 cm for a total of 459 times (1.8% of the times that modeled discharge and PEC stage
overlap temporally); at these times, the mean observed stage is 19.8 cm, with a standard deviation of 5.1 cm. Observations
of stage that exceed 10 cm while modeled discharge is zero, and for which we have image classifications, occur 83 times
(5.2% of the times where modeled discharge, PEC stage, and image classifications overlap temporally; fig. 11). For example,
from 18 January 2018 through 6 February 2018, image classifications show primarily 'high water' and manual discharge
measurements estimate flows of 0.167 $m^3 s^{-1}$ and 0.368 $m^3 s^{-1}$, while the modeled discharge estimates 0 $m^3 s^{-1}$ and 0.011
$m^3 s^{-1}$, respectively (fig. 10). In contrast, during 19-21 March 2018, modeled discharge values were within 0.031 $m^3 s^{-1}$ of
three contemporaneous manual discharge measurements (fig. 10). There are also 3,758 times (14.4% of the times that modeled
discharge and PEC stage overlap temporally) when the observed stage is zero while the modeled discharge is greater than zero;
at these times, the mean discharge is 0.014 $m^3 s^{-1}$, with a standard deviation of 0.007 $m^3 s^{-1}$.





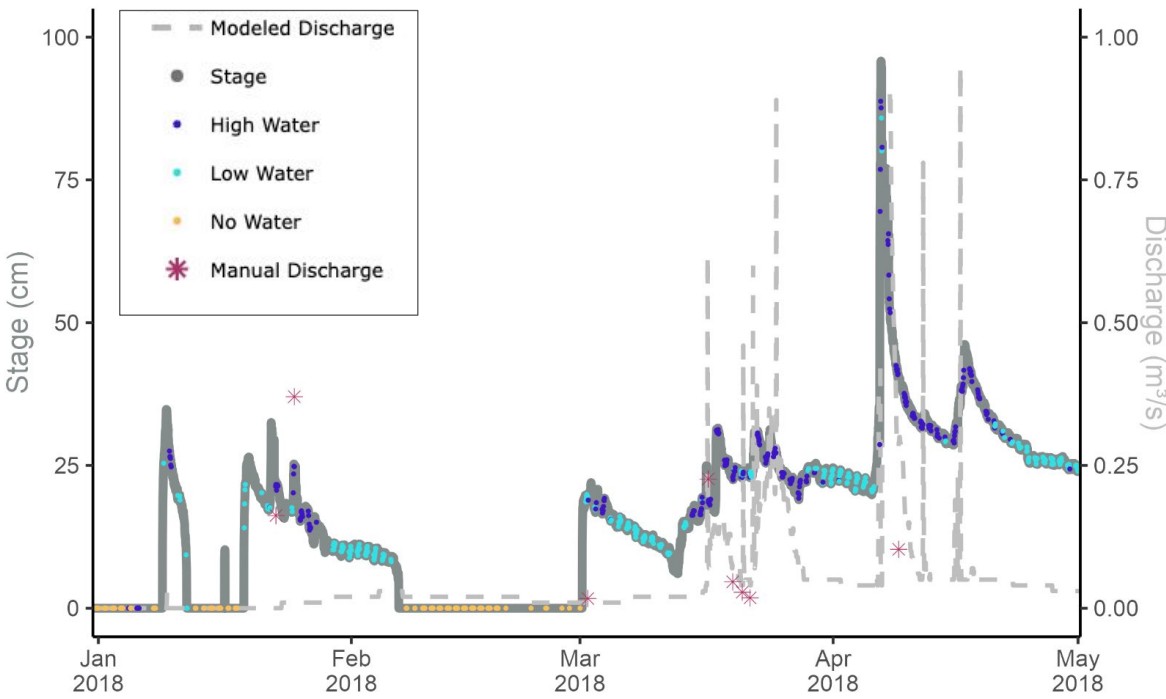

**Figure 10.** Perry Creek (PEC) observed stage, instantaneous manual discharge measurements, and modeled discharge for January 2018 through April 2018. The PEC stage is colored by the high-confidence image classifications: 'high water', 'low water', or 'no water'.



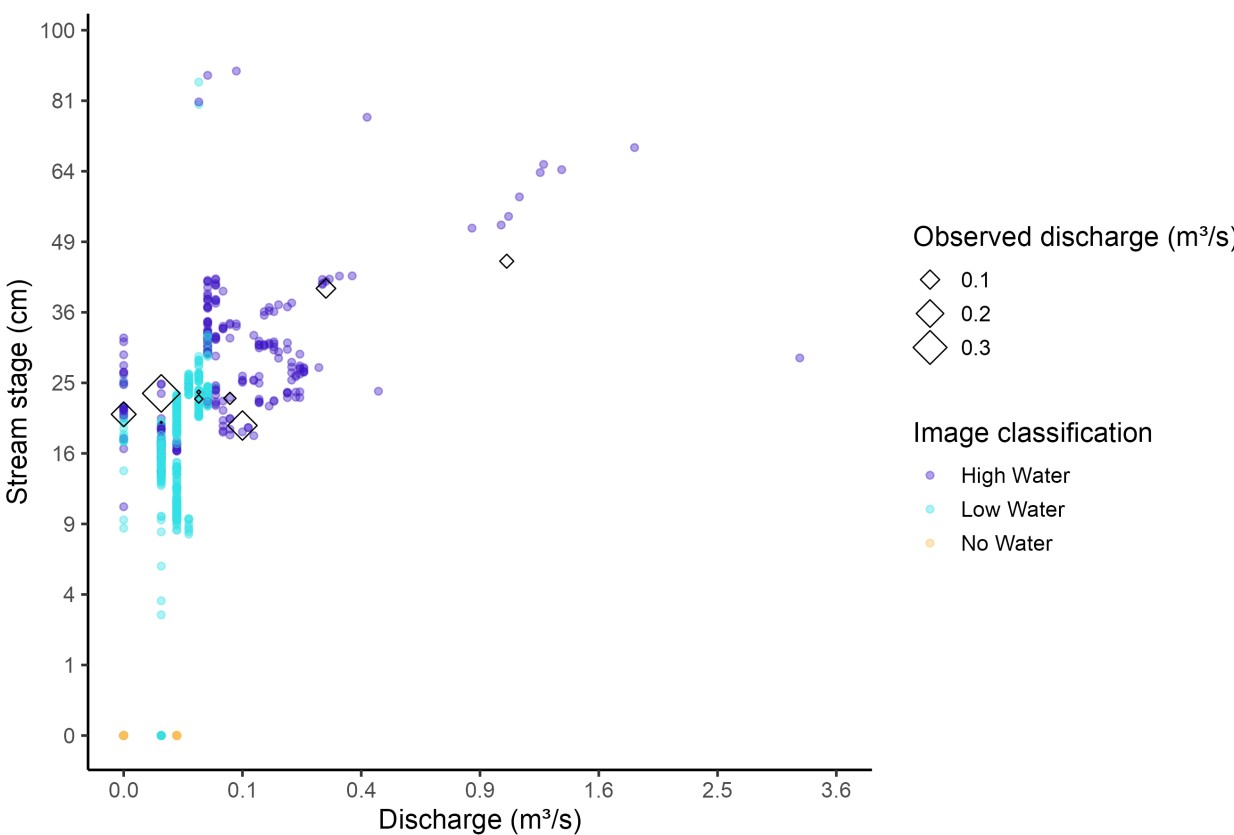

**Figure 11.** Comparison between modeled discharge, manual discharge measurements, and observed stage at Perry Creek (PEC). Points show observations of stage (cm, vertical axis) and corresponding modeled discharge ($m^3/s$, horizontal axis), according to the corresponding high-confidence image classification (color). The value of available manual discharge measurements (unfilled diamonds) corresponding to the observed stage and modeled discharge points are shown according to their magnitude ($m^3/s$, diamond size).





Finally, while we do not have a sufficient number of manual discharge measurements to translate PEC stage to discharge across the full range of flow, the manual discharge measurements nevertheless provide some understanding of possible observation-
based discharge values at PEC. The minimum observed discharge during periods with non-zero stage and discharge was 0.014 $m^3\,s^{-1}$ on 19 April 2023, and the maximum observed discharge was 0.614 $m^3\,s^{-1}$ on 6 March 2024. Photos and image classifications suggest that discharge can exceed this, but high flows remain unmeasured due to the site being unsafe to access during flows much higher than the 0.614 $m^3\,s^{-1}$ measured on 6 March 2024. We gain some additional understanding of high discharge values at PEC by comparing modeled discharge and concurrent manual discharge measurements. This comparison
(fig. A9) indicates some agreement between measured and modeled discharge at moderate flows ($< 0.283\ m^3\,s^{-1}$), but poor agreement at higher flows ($> 0.283\ m^3\,s^{-1}$). Additionally, the longer time period modeled by the NWM (2000-2020, fig. A10) suggests that our study period (2017-2020) does not capture the full range of peak flows at PEC.

### 3.4  Environmental conditions

In order to understand the hydrologic context of the image classifications and stage data, we evaluated temporal correspondence
between stage and concurrent surface meteorology and soil moisture data (figs. 12 and A8). In general, 'high water' stage has a positive relationship with daily rolling sum precipitation (slope = 0.21, $R^2 = 0.41$), 30-day rolling sum precipitation (slope = 0.16, $R^2 = 0.41$), and 5 cm daily rolling mean soil VWC (slope = 126.36, $R^2 = 0.37$). Similarly, 'low water' stage is also positively related to these meteorological variables; the relationship is similar to that of 'high water' for daily precipitation (slope = 0.47, $R^2 = 0.15$) and less robust for 30-day precipitation (slope = 0.07, $R^2 = 0.2$) and 5 cm soil VWC (slope = 29.74,
$R^2 = 0.08$). Notably, the $R^2$ value for 'high water' and 30-day precipitation indicates a stronger fit compared to the $R^2$ value for 'high water' and daily precipitation. There is also a slight positive relationship between 'high water' stage and 100 cm daily soil VWC (slope = 22.65, $R^2 = 0.04$).





**Figure 12.** Correspondence between Perry Creek (PEC) stage and concurrent surface meteorology and soil moisture data. PEC stage (cm, vertical axis) and its concurrent medium or high-confidence image classification (point color) compared to (horizontal axes): a) 2-minute mean air temperature, b) 2-minute mean relative humidity, c) daily rolling sum precipitation, d) 30-day rolling sum precipitation, and daily rolling mean soil volumetric water content at e) 5 cm depth and f) 100 cm depth. Linear regression estimates (lines, insets) summarize the relationship between stage and environmental data (a-f) for 'low water' (dotted lines) and 'high water' (solid lines) image classifications.





Stage does not have a clearly identifiable relationship with the other evaluated meteorological variables, although different stage values appear to occur within identifiable ranges of all the variables. All three image classification categories — 'high water', 'low water', and 'no water' — occur for a wide range of temperatures, but there are no instances of water occurrence at temperatures greater than 30°C (fig. 12a). For relative humidity (RH) below 24% there are only "no water" or "low water" classifications; for RH greater than 24%, all three classifications are present but with an increasing number of "high water" classifications as RH approaches 100% (fig. 12b). Overall, maximum stage increases as RH increases, although stage was at times zero across the full range of RH values. Stage is only greater than zero when daily and 30-day rolling sum precipitation exceeds 45 mm and 160 mm, respectively (fig. 12c-d), and stage is always zero when 30-day precipitation is at or near zero (fig. 12d). The daily soil VWC regularly exceeds 0.7 at a soil depth of 100 cm (fig. 12), but rarely exceeds 0.5 at the shallower soil depth of 5 cm (fig. 12e). While 'high water' stage increases on average with increasing VWC at both 5 cm (slope = 126.36) and 100 cm (slope = 22.65) depths, stage measurements and their water level classification vary substantially across much of the range of VWC values. Even so, for 5 cm VWC, stage is zero when soil VWC is less than 0.15; only 'low water' or 'no water' classifications occur at soil VWC between 0.15 - 0.27; all classifications are present for soil water contents between 0.27 - 0.39, and stage is greater than zero when soil VWC is greater than 0.39. In summary, meteorological and soil moisture patterns are consistent with expected antecedent and coincident hydrology (stage) at this site, and also reflect expected patterns in the associated image classifications.

## 4   Discussion

### 4.1   Image classification performance

We directly address the need to develop methods to observe stream conditions in IRES (Gupta et al., 2022; Noto et al., 2022) through the training and application of a simple logistic regression model for the classification of field camera imagery. While more complex machine learning model algorithms can sometimes provide more accurate image classifications, initial testing demonstrated that the logistic regression model performed better than standard implementations of other models, including image segmentation, clustering, and random forests. While convolutional neural network (CNN) models are common for image classification and desirable due to their theoretical generalization, we did not have enough training data to adequately train a CNN. Thus, our implementation prioritizes a simple, accurate, site-specific model that requires minimal manually-labeled training data. Even so, our overarching modeling workflow is transferable to other stream sites, and only requires labeling a subset of site images as demonstrated in this study.

While our image classification method performed well, the method could be improved by further considering the time-dependence of flow states, especially in cases where stage data is either missing or not available. Since individual streamflow observations are related to each other at the 30-minute or hourly time scale of the field camera images (i.e. if one image is classified as 'no water', the next image is more likely to also be classified as 'no water'), this temporal correlation of flow states might be used to flag unlikely classifications and set them to a 'low' confidence level. Precipitation and soil moisture



data, in conjunction with the image classifications, could also provide additional quality control information even in the absence of stage or discharge measurements.

Furthermore, the present application includes only three categorical flow states which loosely correspond to quickflow ('high water'), baseflow or pooling ('low water'), and a dry streambed with no flow ('no water'). This is fitting for IRES as this classification approach prioritizes the presence or absence of water. Future work could expand the number of classified

categories to include more water levels as in Seibert et al. (2019) and Gupta et al. (2022). Alternatively, models might be designed to estimate stage for images classified as 'low water' or 'high water' by building on methods for estimating stage on perennial streams, as in Leduc et al. (2018), but would need to address the problem of adverse lighting and turbulent flow conditions common to IRES. An approach more suitable for images classified as 'low water' or 'high water' in the IRES context could be the estimation of relative streamflow from images using a CNN, as in Gupta et al. (2022), but this would

require more training data.

One strength of the categorical image classification is that 'low water' conditions include times of water presence in connected or disconnected pools, a distinct phase of IRES that supports aquatic life (Magand et al., 2020). At PEC, the stage sensor is generally not exposed to water during pooling, so the stage is measured as 0 cm even when there may be water in nearby pools, meaning that the observed stage may not fully illustrate water presence. Thus, when cropping the images prior

to classification, it was important to include most of the width of the streambed so that pooling in the streambed was visible. Further work could expand the 'low water' category to separate low flow from pooling conditions. Discharge time series also do not capture the pooling phase of IRES because discharge is also zero at these times. Therefore, field camera images and image classification offer a way to observe the ecologically important 'pooling' phase of IRES.

Regardless of the potential uses of field camera imagery, image quality and the reliability of field cameras will always

be a challenge due to factors including inclement weather, shadows, and changes to IRES sites (i.e., high flows leading to debris accumulating in the channel). Pre-processing images helped to decrease the effect of variable lighting, and assigning classification confidence levels helped flag lower confidence classifications that may have been affected by image quality issues. Other studies have tackled this issue by pre-processing images with low image quality prior to classification. For example, Leduc et al. (2018) pre-processed images to account for image quality issues by removing images or adjusting

identification methods in situations of inclement weather, shadows, or when rocks emerged from the streambed. Another issue is the difficulty of taking high quality images at night, even with a flash. Zhang et al. (2019) solved this problem by using an infrared camera, but regular cameras are often a better fit for remotely monitored sites. Finally, field cameras can often malfunction due to environmental exposure including extreme precipitation, humidity, and temperatures, making field cameras difficult to maintain.

For stream imaging generally, it is advantageous to choose a protected, stable, and fixed position for the field camera such that images are all taken with a full view of the streambed and from the same angle, which saves time during image pre-processing and classification. In this study, our method was complicated by four slightly different camera viewing angles. We minimized the effect of these changing viewing angles by cropping all images to focus on the staff plate and streambed. For locations with



a time series of images with the same viewing angle, pre-processing would be less burdensome, and classification would likely
improve because differences in images would only come from event evolution and not camera location.

## 4.2   National Water Model discharge at Perry Creek

We used NWM discharge estimates at the study site to demonstrate the use of image classifications to augment modeled discharge from a highly generalized streamflow prediction model, which the NWM represents. The observed differences between modeled discharge and stage are expected, particularly for small basins like that of PEC, since uncertainty in hydrologic models generally increases log-linearly as watershed area decreases (Carpenter and Georgakakos, 2004). In our case, modeled discharge values are for a stream segment that overlaps the PEC site (fig. A5), so there is a difference in the spatial scales corresponding to stage and discharge measurements that may affect the comparison (NOAA, 2024a; Cosgrove et al., 2024). Specifically, the NWM's representation of the PEC stream segment may have some overlap with Lake Mendocino during high lake levels, which could result in the representation of backflow from the lake as increased (observed) stage but with no corresponding increase in discharge. Additionally, much of the NWM's PEC segment overlaps an area underlain by well-indurated Franciscan sandstone (Delattre and Rubin, 2020), which could induce higher discharge values along much of the Perry Creek segment relative to the location of the PEC site (see Discussion: Unique Site Features). Because the NWM is unable to resolve the detailed geology, steep slope, and groundwater-surface water interactions of this reach of Perry Creek, we would not expect the NWM to accurately reproduce flows at this particular site. While the NWM is generally relied upon to model discharge at larger scales (basin areas > 1,000 km$^2$), the modeled discharge estimates are nevertheless useful for representing the type of limited hydrologic information often available at less well-studied sites.

According to our comparisons, the NWM discharge misses or underestimates some short-duration, moderately-sized quickflow events (fig. 10). Additionally, at low stage and discharge levels, manual measurements of PEC discharge can be greater than those of modeled discharge (fig. A9), and modeled low discharge appears stratified (fig. 11). This is not surprising, as the NWM's physical process representation of low flow is not tailored to this site nor IRES systems in general. Low flows have been shown to be sensitive to the spatial scale of analysis and site-specific catchment characteristics, especially geology and soils (Chagas et al., 2024), which are not well represented in generalized models like the NWM. Conversely, at high stage levels, the few available manual measurements of discharge are less than corresponding modeled discharge (fig. 11). Overall, our comparison makes clear that image classification in IRES benefits not only the quality of the in-stream observational record, but also stands to improve hydrologic model representations of IRES. Given the relatively uncertain but important contribution of IRES systems to downstream watershed flow regimes, better quantification of low flow components of inflows to heavily managed systems, like the Lake Mendocino watershed, would support water storage and delivery planning efforts (e.g., Jasperse et al., 2020).

## 4.3   Environmental conditions

DRW surface meteorology and soil moisture observations help establish the hydrologic context for the image classifications. These observations show the expected seasonal cycle of a Mediterranean climate with cool, wet winters and warm, dry sum-





mers, along with significant interannual variability in precipitation (fig. A8). The ranges of meteorological and soil VWC values observed for different flow states (e.g., there only being 'no water' classifications for temperatures > 30°C) may provide helpful upper and lower bounds within which different flow conditions can be expected to occur in IRES. Nevertheless, a

longer-term record of these data would be required to understand those bounds.

The comparison of soil VWC at 5 cm and 100 cm shows that high shallow soil moisture is a stronger predictor of water presence at PEC compared to deep soil moisture (fig. 12e-f). The existence of 'no water' image classifications across the full range of values for 100 cm VWC shows that deep soil moisture can be decoupled from the presence of water at PEC (fig. 12f). In contrast, the shallow 5 cm VWC provides insight into the soil moisture conditions present during high water

levels; 'high water' image classifications are predominant when 5 cm VWC is greater than 0.39 (fig. 12e). With respect to low soil moisture conditions, stage was zero when 5 cm VWC was less than 0.15 (fig. 12e), indicating that there may be a soil moisture threshold at which flow cannot occur at PEC. Notably, there is considerable spatial variation in soil moisture observations, and the soil type at DRW is different from PEC (Soil Survey Staff, 2024). This suggests that if soil moisture observations existed in the PEC watershed, the relationship between such observations and PEC stage could be different than

our findings for DRW soil moisture and PEC stage. The different soil type at DRW is likely related to the different geology at DRW—the Ukiah Formation, an early Quaternary to Late Neogene continental basin deposit including conglomerate, silty sandstone, and clayey siltstone (Delattre and Rubin, 2020). In contrast, the PEC watershed's steep slopes, landslide deposits, and fractured Franciscan Formation likely results in shallower and more unstable soils compared to soil in the rolling hills surrounding DRW. Confirmation of our understanding of the role of soils and geology in modulating the stage at PEC would

require additional research that is beyond the scope of this work. However, it remains clear that soil moisture observations can help relate antecedent soil moisture conditions to flow occurrence at PEC.

### 4.4    Unique site features

PEC is a unique site due to its proximity to the reservoir-impounded Lake Mendocino and its location just downstream of a geologic contact. Its geologic features, in particular, may generate an observational setting that is distinct from other, even

nearby, locations. While the hydrograph at PEC is often characterized by discrete quickflow events (e.g. WY 2020 in fig. 9), some wet seasons may also have baseflow for part or all of the wet season (e.g. WY 2019 in fig. 9). Just upstream of PEC, mossy cliffs surround a gorgeous waterfall that flows over dark gray, fine-grained, massive, well-indurated Franciscan sandstone (fig. A2; Delattre and Rubin, 2020). The streambed at PEC is composed primarily of angular to subangular pebble to boulder-sized Franciscan sandstone (fig. A11) and there is often woody debris that shows evidence of high flows. The stream gradient

of Perry Creek is very high upstream of PEC with a series of pools and drops; it then flattens just upstream of PEC, at the last visible outcrop of sandstone bedrock. Due to these features, PEC is prone to flashy, turbulent, sediment-laden flows, which field camera images may capture but that may not be reliably represented in stream monitoring records due to noisy and/or biased stage measurements resulting from equipment malfunctions at high flows (Appendix A2).

Another feature of the PEC site, which may influence surface flow patterns, is subsurface flow. We hypothesize that sub-

surface flows may at times bypass the PEC surface site due to the Lake Mendocino and PEC water tables being connected.



Geologic maps and visible surface characteristics indicate that the PEC site may be located in an area along Perry Creek where surface flow transitions to subsurface flow. For example, in January 2022, the waterfall upstream of the PEC site was flowing (fig. A2) and there was a small pool of water in the sandstone outcrop just upstream of PEC; however, there was no water in the streambed at PEC, though the rocky streambed was damp. Subsequently, during a site visit in June 2024, the waterfall upstream
of PEC was again flowing, and the PEC site was dry. In addition, gray, poorly indurated shale, which likely has substantial water holding capacity, was found in the steep slope just north of PEC. On the south side of PEC, there is a steep, mossy, poorly-consolidated landslide deposit that contains a mix of sandstone boulders, shale, and mud. This landslide deposit begins just upstream of PEC and may partially overlay the shale. This, combined with highly transmissive Holocene alluvial deposits along Perry Creek's outfall to Lake Mendocino (Delattre and Rubin, 2020), may lead to groundwater recharge and subsurface
flow to Lake Mendocino from an area in the streambed just upstream of PEC. This could cause decreased surface water flow at PEC. Additionally, the pattern of groundwater flow between PEC and Lake Mendocino may be further complicated by preferential flow pathways through fractures in the Franciscan Complex sandstone, analysis of which is beyond the scope of the present study.

High lake levels at Lake Mendocino may also affect PEC surface water dynamics. PEC borders an area designated as
"subject to flooding" due to its proximity to Lake Mendocino (fig. 1; Moore et al., 2019). Another reason to suspect that lake levels influence stream state at PEC is the difficulty of constructing a rating curve at PEC — a range of values for discharge have been observed at almost identical stage values. It is likely that there are several stage-discharge relationships at this site based on a variety of factors including channel geometry as stage increases, backwater influences, and instability of flow. Constraining these relationships would require more manual discharge measurements at the full range of flows, along with
analysis of cross-sectional surveys to understand how the channel geometry affects flow as stage increases from low water levels to bankfull conditions. These efforts are outside the scope of this work, and it remains that estimating an accurate stage-discharge relationship at extreme high flows utilizing conventional discharge measurement techniques may be impossible due to site access limitations.

The above unique site features indicate that Perry Creek may at times have an outsized, yet unaccounted for, contribution
to Lake Mendocino inflows. The primary streamgage used for understanding inflows to Lake Mendocino is USGS streamgage 11461500 located on the East Fork of the Russian River (EFR in fig. A1), which monitors flows from a drainage network separate from that of Perry Creek. The PEC and EFR watersheds account for about 3% and 89% of the total area draining into Lake Mendocino, respectively. Despite its smaller size, the relatively steep relief and underlying geology of the Perry Creek watershed may result in runoff ratios that are greater than those experienced by the larger Lake Mendocino watershed
for similarly-sized precipitation events. This could potentially result in Perry Creek inflows constituting a fraction of total Lake Mendocino inflow that is larger than its watershed size would suggest, specifically during more extreme storm events. While arrival at this hypothesis was made possible through our comparison of classified field camera imagery and site monitoring, confirmatory analysis is beyond the scope of the present study. Nevertheless, investigation of this hypothesis would be valuable for understanding the relevance of IRES for decision support (e.g., reservoir inflow model design) in managed systems like
that of Lake Mendocino.



## 4.5 Extensibility and potential applications

The classification model (see Data and Code Availability statement) is structured to ingest identically-sized images (with the streambed in the field of view) from any individual site's field camera imagery collection, as long as a subset of images are labeled and sorted into folders that are named for each classification category. Currently, the classification portion of the model
code will work to classify any number of categories specified by the user, but the classification confidence portion is written specifically for the four categories from this work ('no water', 'low water', 'high water', and 'obstructed'). The code could be expanded to provide classification confidence values for additional and/or alternative categories by any user familiar with the Python programming language. Thus, the method as currently structured allows a user to create a time series of categorical flow states, sorted by confidence level, for any IRES site with similar timelapse field camera imagery (e.g., images from USGS Flow
Photo Explorer; USGS, 2024) with or without stage sensors. For sites without stage sensors, this method offers a cost-effective way to estimate categorical flow states in IRES. For sites with a stage sensor, image classification can support validation and quality control of stage data.

Although there have been efforts to use field camera imagery to quantify stream stage and flow, especially on perennial streams (Leduc et al., 2018; Gupta et al., 2022), these efforts have struggled to translate to IRES. While it is impossible to
have field cameras at all IRES, expansion of field camera networks with image processing supported by machine learning has the potential to greatly increase monitoring of IRES at low cost, which could inform modeling of both gaged and ungaged reaches. Sites with field cameras would also benefit from the installation of machine-readable high-contrast target backgrounds to improve measurement of stage (Chapman et al., 2022). The rapidly developing field of multimodal artificial intelligence models might also help accelerate and automate image labeling, and be employed to perform image classification (Liu et al.,
2023). Ideally, future hydrologic image classification work would focus on combining and simplifying existing approaches with a focus on usability across stream types, including IRES.

## 4.6 IRES observations in a changing climate

Due to the significant but relatively poorly understood contribution of IRES to downstream water systems – natural and man-aged – better understanding and managing of IRES networks is an important component of climate change resilience and
adaptation in arid regions like the western U.S. The plethora of ungaged IRES make field cameras and image classification a desirable low-cost option for improving our understanding of the role of IRES in climate-changed freshwater systems. For example, finding ways to keep water in the cooler, often-forested headwaters longer by increasing groundwater storage and maintaining IRES flow for longer periods of time benefits freshwater ecosystems, including valuable fisheries (Dettinger et al., 2023). Strategies for keeping water in headwater streams longer include 'natural' options such as wetland restoration, beaver
dam analogues, and forest thinning, as well as 'managed' options such as Forecast Informed Reservoir Operations (FIRO; Dettinger et al., 2023), which uses weather and reservoir inflow forecasts to support more flexible reservoir operations (AMS, 2020). The functionality of any such strategy for climate change adaptation in IRES systems cannot be evaluated without comprehensive monitoring.



A specific example of this is provided by the FIRO program operated at Lake Mendocino (fig. A1). Streamflow observations
from EFR currently inform reservoir inflow models used in Lake Mendocino's FIRO program. Climate change is expected to
lead to drying in IRES, which could have unknown effects on lands adjacent to the reservoir and reservoir storage, especially
if IRES tributaries like Perry Creek might contribute an outsized amount of unmonitored flow, as conditions at PEC suggest.
Thus, as the FIRO program expands, field cameras and image classification may help integrate understanding of the presence
and/or magnitude of IRES contributions at low cost.

Finally, formally recognizing IRES as being part of a river system can incentivize monitoring and protect them from degra-
dation caused by climate change and human activities like mining and urban development (Acuña et al., 2014). Thus, the 2023
exclusion of ephemeral streams from federal protection under the U.S. Clean Water Act highlights a direct threat to IRES
monitoring and protection. Under the assumption that IRES flow contributions to larger downstream freshwater networks are
environmentally and socially important, low-cost and computationally efficient IRES monitoring approaches like the one pre-
sented here may prove critical for understanding the impacts and effectiveness of water management efforts today and into the
future.

*Code and data availability.* Code that implements the final model and associated data from this study (images, PEC stage data, PEC manual
discharge data, DRW soil and surface meteorology data, and NWM discharge data) are publicly available and citable on the CUAHSI
HydroShare platform at: http://www.hydroshare.org/resource/926f16b0d54242879777b19fa805ef79.



## Appendix A: Appendix A

### A1 Appendix A1: Barometrically compensating stage data

Prior to the installation of barometric loggers in late 2021 or early 2022 at each CW3E stream site (fig. A1), 2-minute atmospheric pressure from nearby surface meteorology stations was used to barometrically compensate pressure measurements from the stage sensors. Each stream site was matched with a nearby surface meteorology site that was closest in elevation; thus, the Perry Creek stream site (PEC) was matched with the Deerwood (DRW) surface meteorology site. After the barometric loggers were installed at the stream sites, we were able to account for the difference in atmospheric pressure between the barometric logger location and the surface meteorology location by calculating the mean pressure difference for April 2022 through March 2023 (figs. A3, A4).

The mean annual atmospheric pressure difference between DRW and PEC was 5.00 hPa. The minimum difference calculated at 15-minute intervals was 3.30 hPa and the maximum difference was 7.10 hPa for April 2022 through March 2023. This means that the stage would have been on average 5.00 cm higher if it had been calculated with the atmospheric pressure at DRW instead of the atmospheric pressure of the barometric logger at PEC. Prior to installation of the barometric logger at PEC, 5.00 hPa was subtracted from the surface meteorology pressure values to account for the location discrepancy such that the stage formula for PEC was:

$$
\begin{aligned}
\text{PEC stage (cm)} = \text{PEC stage sensor (cm)} &- \left( \text{DRW atmospheric pressure (hPa)} \times 1.01972 \, \frac{\text{cm}}{\text{hPa}} \right) \\
&- \left( \text{mean} \left( \text{PEC atmospheric pressure (hPa)} - \text{DRW atmospheric pressure (hPa)} \right) \times 1.01972 \, \frac{\text{cm}}{\text{hPa}} \right)
\end{aligned}
\tag{A1}
$$

In comparison, the stage for PEC after the installation of the barometric logger at PEC is simply:

$$
\text{PEC stage (cm)} = \text{PEC stage sensor (cm)} - \left( \text{PEC atmospheric pressure (hPa)} \times 1.01972 \, \frac{\text{cm}}{\text{hPa}} \right)
\tag{A2}
$$

### A2 Appendix A2: Quality control of stage data

In addition to barometric compensation of pressure transducer data, quality control was performed on these stage data to remove anomalous negative stage values, account for noise in these data, and remove values that disagreed with the field camera observations. We performed these corrections by looking at the raw stage values in tandem with the image classifications and field camera images (subset of time series shown in fig. A7). First, all values less than -8 cm were set to null because these values were clearly more negative than the range of noise in the stage data. Then, the mean (-2.79 cm) and standard deviation (0.92 cm) were calculated for all negative values until October 12, 2023, when the baseline stage returned to near zero following the installation of the HOBO Water Level Data Logger. Water depth cannot cannot be negative (we assumed that zero water depth equaled zero stage for this study) and field camera images generally showed dry conditions—zero level—when the stage was below zero. To account for this, an offset of 2.79 cm was added to all data prior to October 12, 2023. In addition, since the





field camera images showed that values below 2 standard deviations above zero (or 1.84 cm) were generally dry conditions, all values less than 1.84 cm were set to zero to remove noise.

Then, we looked at the rest of the time series in comparison to the image classifications to ensure that the rest of the stage data were plausible. Data prior to 7 October 2017 appeared to be higher amplitude noise oscillating near zero, so these data were set to zero. For data prior to 1 March 2018, the amplitude of the noise was greater than the rest of the time series when the stream was dry as shown in the image classifications, so all values less than three standard deviations above zero were set to zero. We also removed a few anomalous near-zero data points resulting from maintenance visits where the pressure transducer

was removed from the stilling well during times of flow. From 5 January 2023 at 15:30 UTC until 4 February 2023, the image classifications show high flow while the stage sensor data shows low or negative flow, so these data were removed. While the stage sensor appears to start functioning on 4 February 2023, the quality of data following this malfunction is unclear as the field camera was not functioning due to water damage from February through 7 October 2023. Some possible reasons for the stage sensor malfunction are a buildup of sediment within the stilling well and turbulent flow confounding the stage sensor

readings. From comparing PEC to other sites and precipitation data, we find that it is possible that this malfunction occurred during what may have been the highest stage values of the PEC time series.

At PEC in October 2023, the original Solinst pressure transducer was demobilized and replaced with the HOBO MicroRX datalogger and MX2001 pressure transducer. During the installation of the HOBO pressure transducer, an attempt was made to prevent malfunctioning of the logger as in winter 2023 by encasing the data cable in flexible plastic conduit anchored to rock

and removing sediment from the stilling well. Specifically, 0.5 inch flexible plastic conduit was anchored to large boulders along the stream channel to house the data cable connecting the pressure transducer to the datalogger, which was installed on a bench above the primary stream channel. Enclosing the data cable in conduit (McMaster-Carr, 2024) anchored to rock prevents debris-laden flows from catching on the data cable and exerting forces which could pull the pressure transducer upwards in the stilling well (which may have occurred in winter 2023). In addition, compacted sediment was cleared from the stilling well and

holes were drilled in the sides of the stilling well to allow sediment to flow through the stilling well more easily.



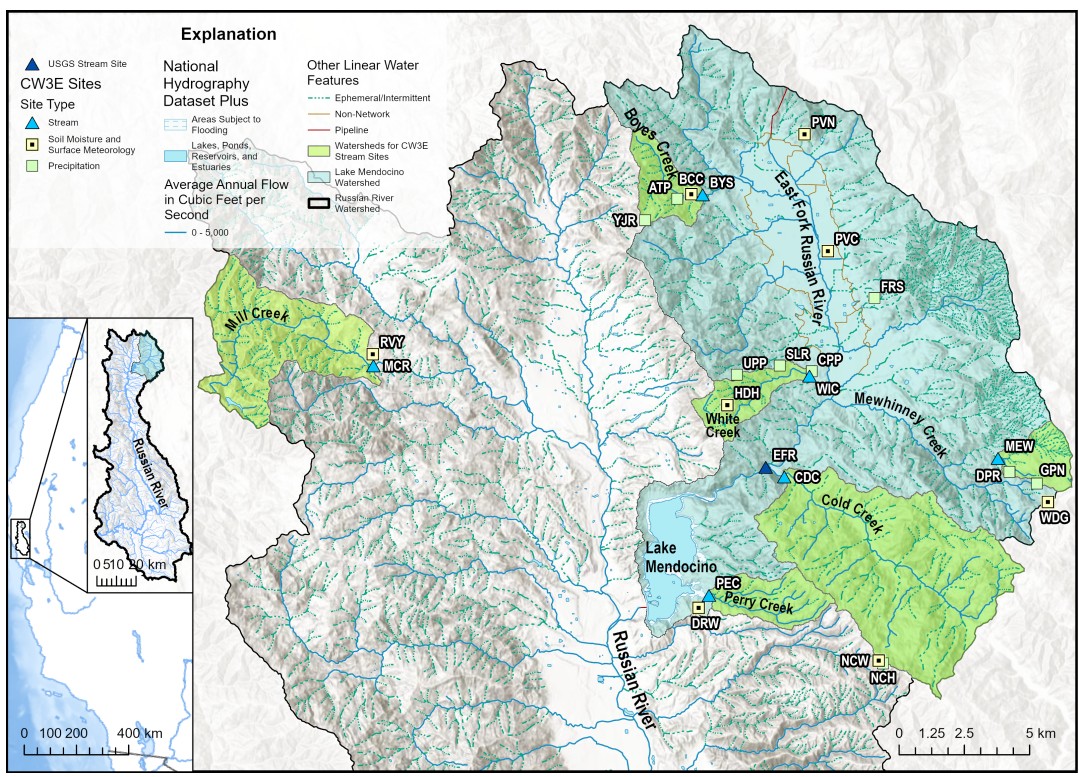

**Figure A1.** Location of the U.S. Geological Survey streamgage 11461500 (EFR) and Center for Western Weather and Water Extremes (CW3E) stream, precipitation, and surface meteorology sites in the upper Russian River watershed, California (ESRI).



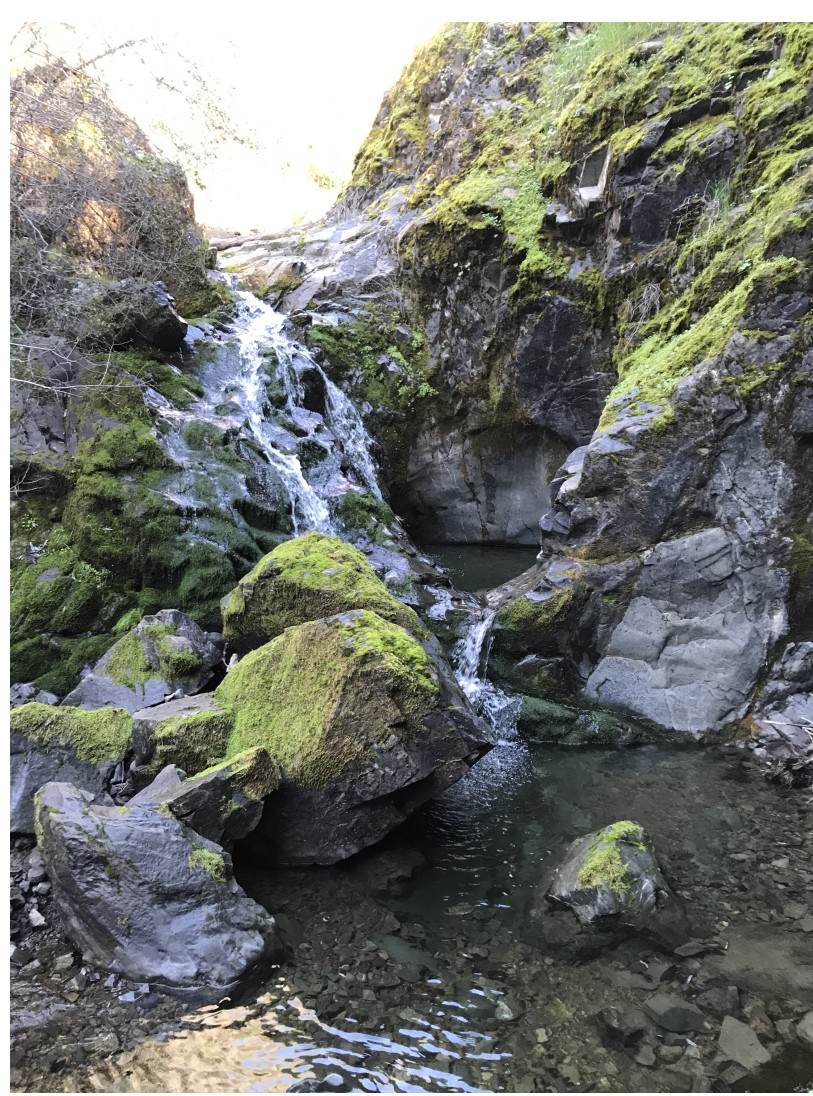

**Figure A2.** Waterfall flowing over mossy Franciscan sandstone just upstream of the Perry Creek (PEC) stream site from January 2022.



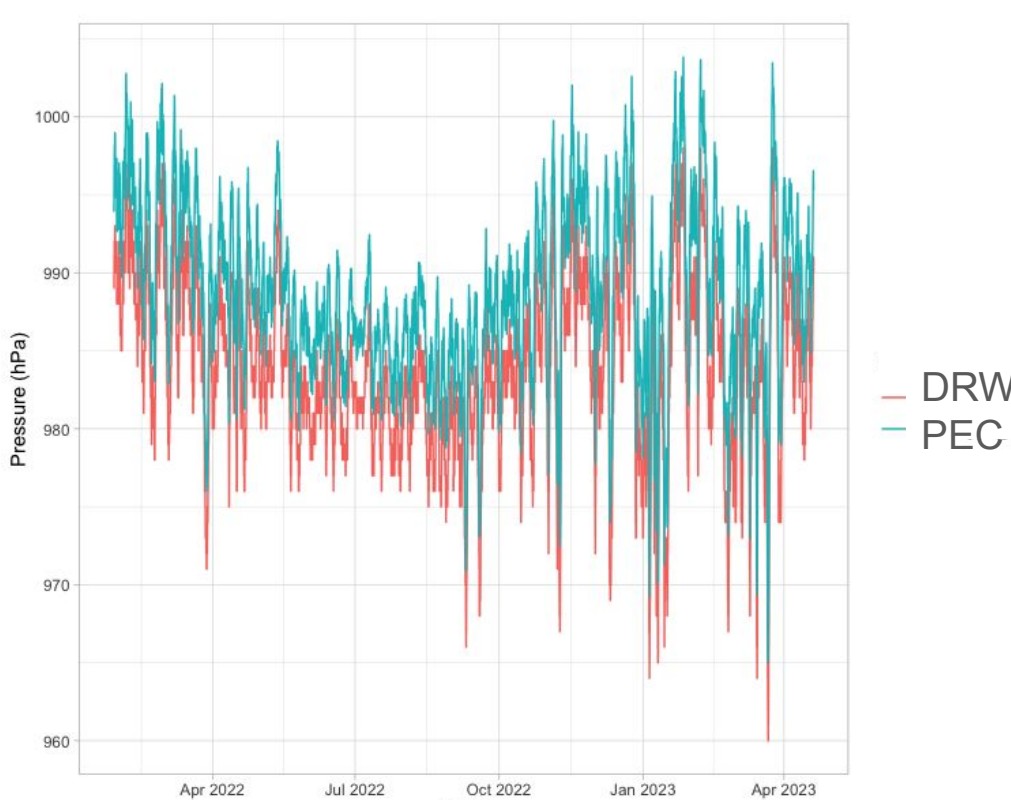

**Figure A3.** Perry Creek (PEC) barologger atmospheric pressure and Deerwood (DRW) surface meteorology site atmospheric pressure.





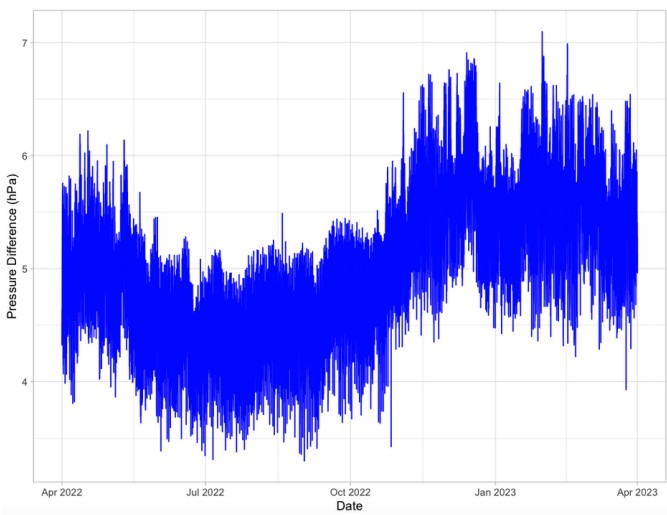

**Figure A4.** Difference between Perry Creek (PEC) barologger atmospheric pressure and Deerwood (DRW) atmospheric pressure from April 2022 through March 2023.





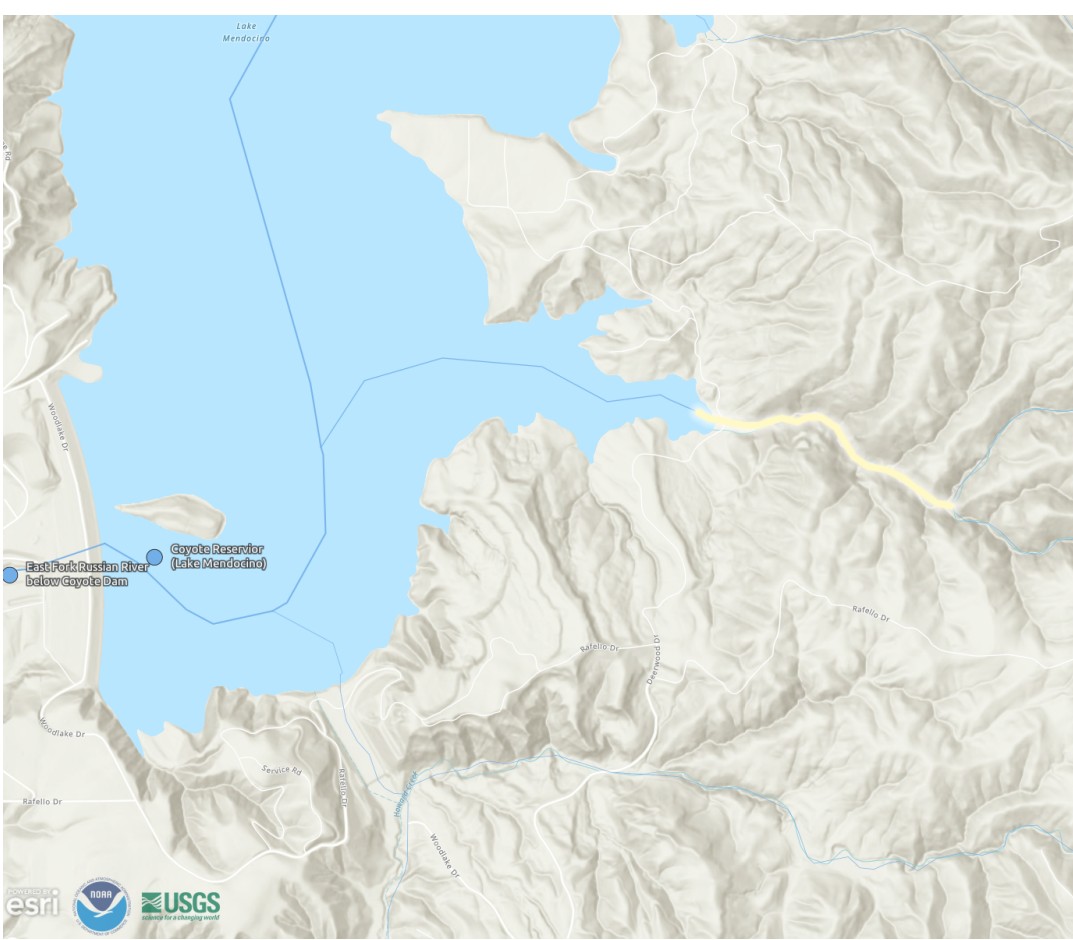

**Figure A5.** The National Water Model segment of Perry Creek that overlaps with the Perry Creek (PEC) monitoring site. The stream segment is highlighted in light yellow, and the PEC site (not shown) is located just upstream of Lake Mendocino along the stream segment in this image. Image source: (NOAA, 2024b)



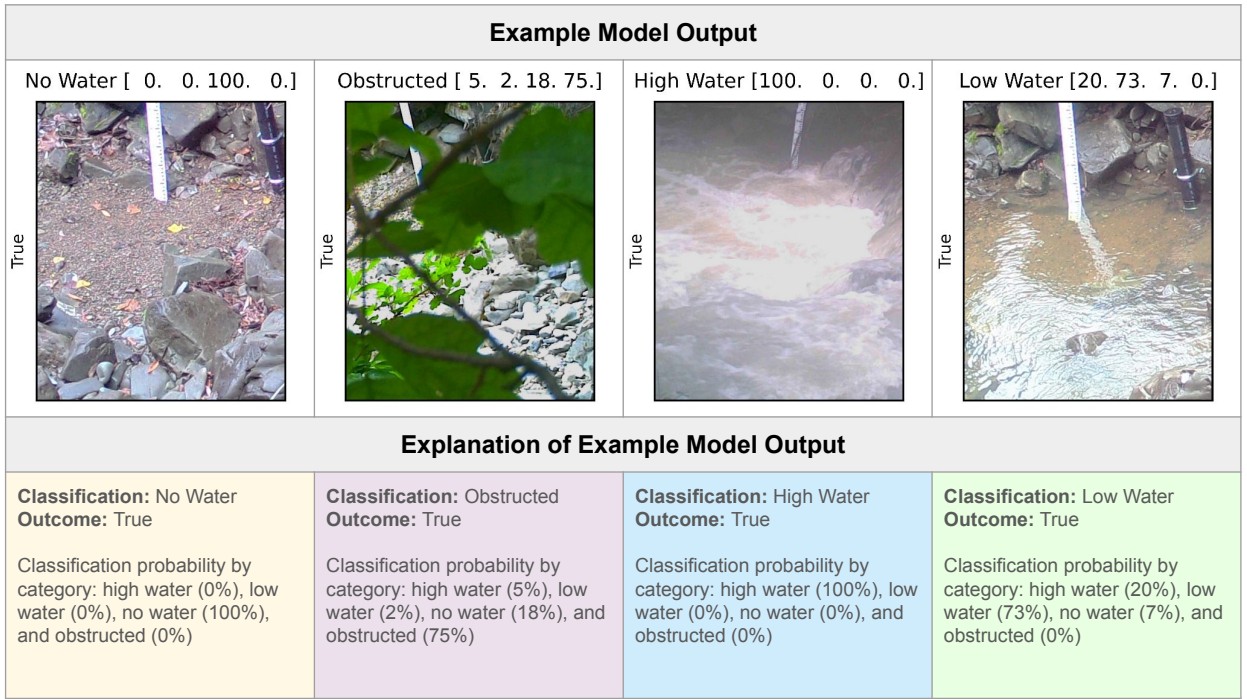

**Figure A6.** A sample of labeled test set images and corresponding classification outputs from the final model run. The images are annotated (text above images) with the classification (high water, low water, no water, or obstructed category labels) and the classification probabilities for each category (%). The simple outcome ("True" for correct classifications and "False" for incorrect classifications) is to the left of each image. A detailed explanation of the classification output is below each corresponding image.





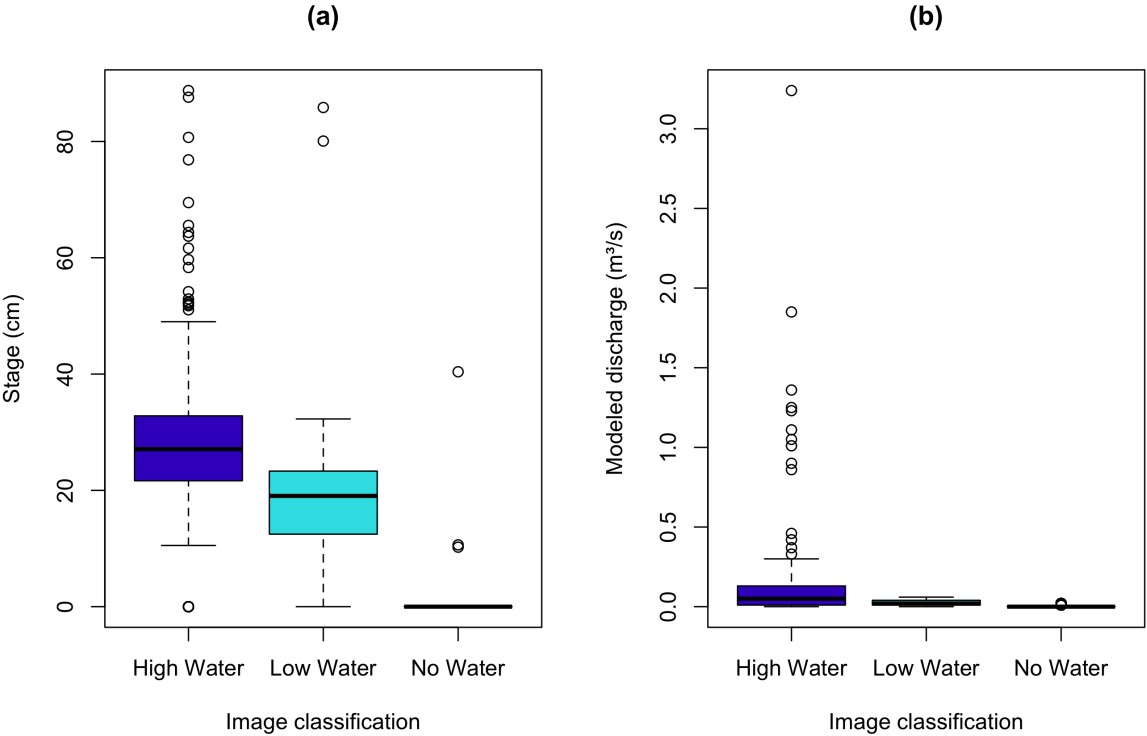

**Figure A7.** Distribution of Perry Creek (PEC) stage and modeled discharge for medium- and high-confidence image classifications. The boxplots show all values for stage (a) and modeled discharge (b) at PEC (vertical axis) corresponding to medium- and high-confidence (only) classifications of images as high, low, and no water (horizontal axis). The boxplots show the interquartile (IQR) range (box), median (bold line), the upper/lower quartile +/- 1.5 * IQR (whiskers), and outliers (points).



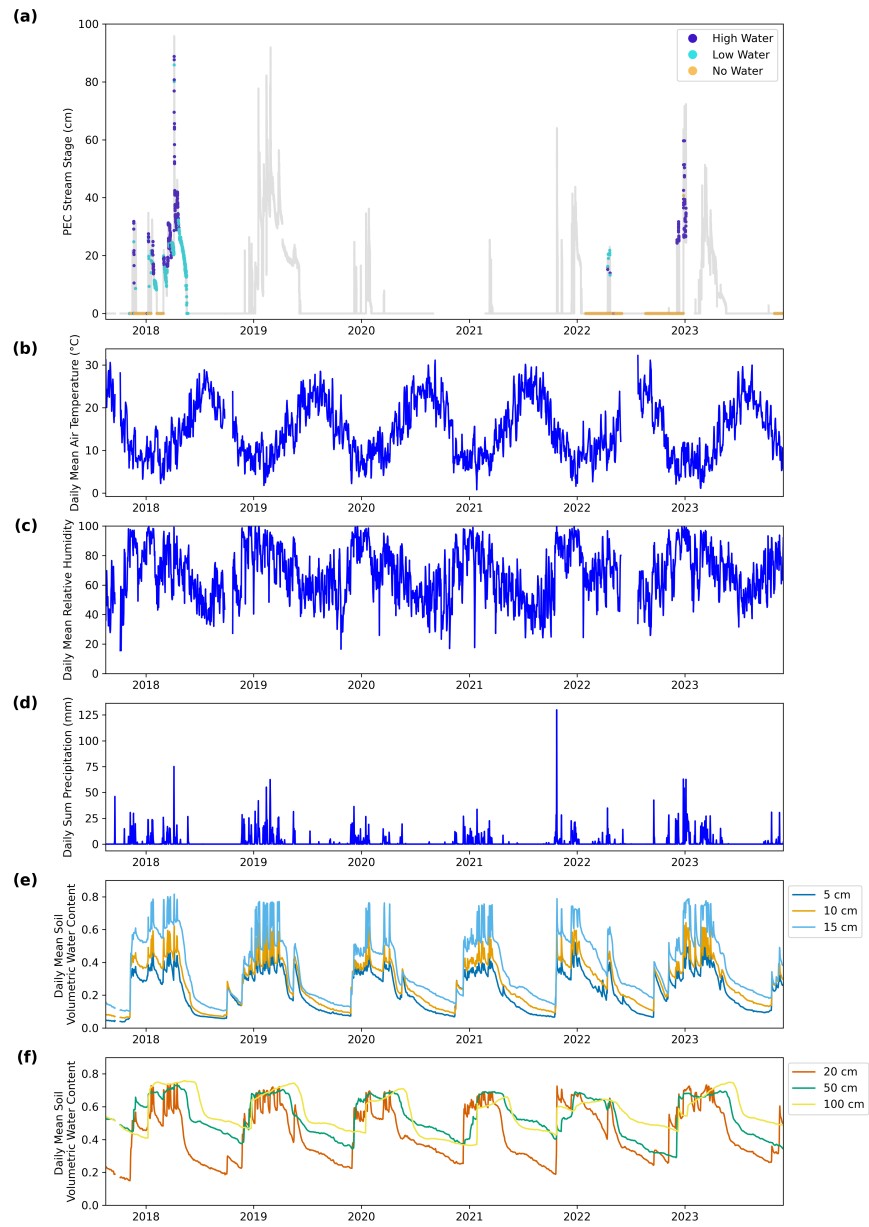

**Figure A8.** Perry Creek (PEC) stage and Deerwood (DRW) site meteorology and soil moisture, August 2017 - November 2023. Time series include: (a) 15-minute barometrically compensated and quality controlled PEC stage colored by concurrent medium or high-confidence image classifications; and daily DRW (b) mean air temperature, (c) mean relative humidity, (d) total precipitation, (e) mean soil volumetric water content (at depths of 5 cm, 10 cm, and 15 cm), and (f) mean soil volumetric water content (at depths of 20 cm, 50 cm, and 100 cm).




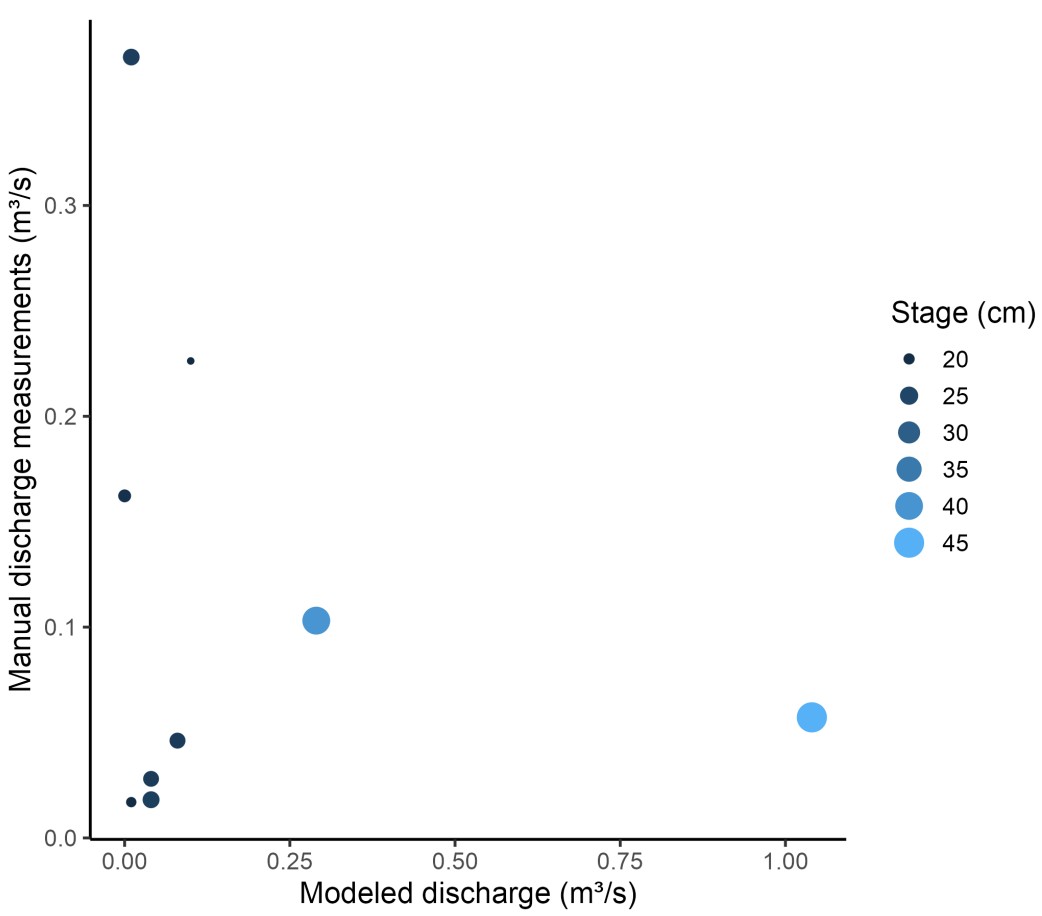

**Figure A9.** National Water Model (NWM) discharge and manual discharge measurements at Perry Creek. Points show available concurrent observations of measured discharge (vertical axis) at the Perry Creek (PEC) site and modeled discharge (horizontal axis) at a stream segment overlapping PEC, which are colored by observed PEC stage.





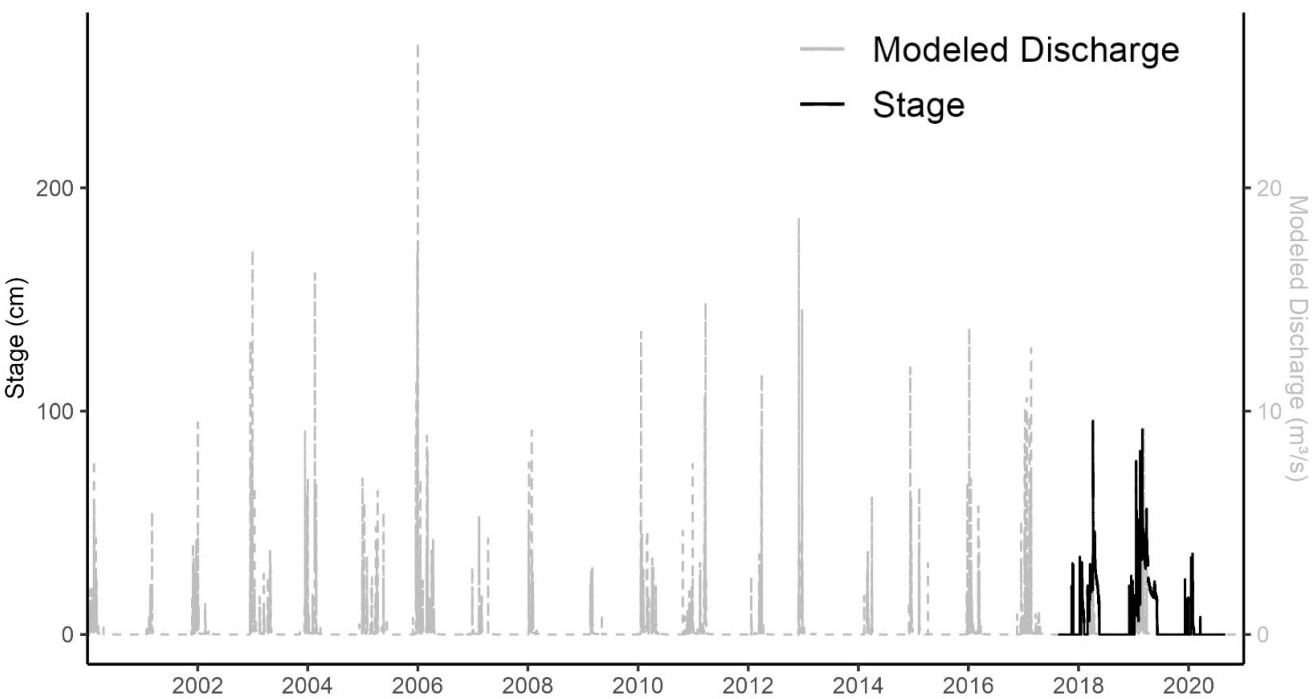

**Figure A10.** Time series of National Water Model (NWM) discharge and stage measurements at Perry Creek. Lines show partially-overlapping observations of observed stage at the Perry Creek (PEC) site (left vertical axis, solid black line) between 2017-2020, and modeled discharge (right vertical axis, dotted gray line) at a stream segment overlapping PEC between 2000-2020.




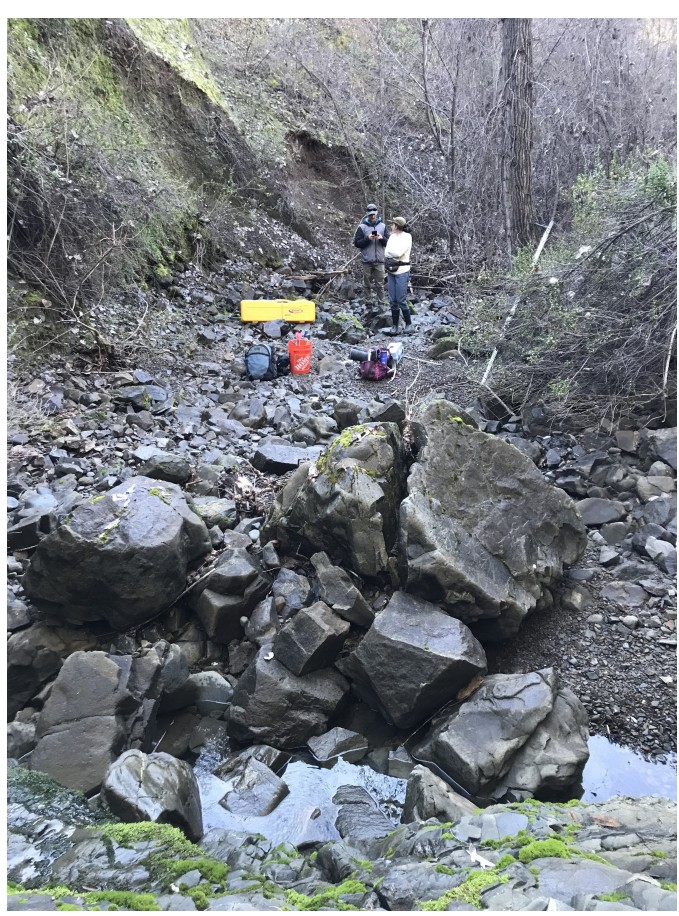

**Figure A11.** Photo from a Perry Creek (PEC) site visit in January, 2022. The photo shows a small pool of water upstream of the PEC site in front of the last visible outcrop of Franciscan sandstone. The rocky, dry streambed is visible in the middle, with the white staff plate on the right. The steep hillside on the left is a landslide deposit. This photo was taken looking downstream.



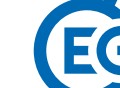

| PEC stage, August 2017 through November 2023 | | | | | | | | | | |
|---|---|---|---|---|---|---|---|---|---|---|
| **Stage percentile** | **60%** | **65%** | **70%** | **75%** | **80%** | **85%** | **90%** | **95%** | **99%** | **100%** |
| PEC stage, cm | 0.0 | 0.0 | 0.0 | 0.0 | 5.1 | 16.7 | 21.5 | 31.3 | 46.1 | 95.8 |
| 'No water' classifications > stage percentile | 0.1% | 0.1% | 0.1% | 0.1% | 0.1% | 0.0% | 0.0% | 0.0% | 0.0% | - |
| 'Low water' classifications > stage percentile | 95.2% | 95.2% | 95.2% | 95.2% | 94.8% | 58.8% | 36.8% | 1.2% | 0.4% | - |
| 'High water' classifications > stage percentile | 93.8% | 93.8% | 93.8% | 93.8% | 93.8% | 86.7% | 75.1% | 32.3% | 5.6% | - |
| **NWM discharge, August 2017 through August 2020** | | | | | | | | | | |
| **Discharge percentile** | **60%** | **65%** | **70%** | **75%** | **80%** | **85%** | **90%** | **95%** | **99%** | **100%** |
| NWM discharge, m$^3$/s | 0.0 | 0.01 | 0.01 | 0.01 | 0.02 | 0.03 | 0.04 | 0.10 | 0.75 | 9.22 |
| 'No water' classifications > discharge percentile | 23.0% | 18.2% | 18.2% | 18.2% | 0.0% | 0.0% | 0.0% | 0.0% | 0.0% | - |
| 'Low water' classifications > discharge percentile | 91.6% | 71.4% | 71.4% | 71.4% | 36.3% | 25.3% | 9.8% | 0.0% | 0.0% | - |
| 'High water' classifications > discharge percentile | 86.4% | 73.5% | 73.5% | 73.5% | 69.3% | 69.0% | 67.9% | 30.0% | 3.5% | - |
| **NWM discharge, January 2000 through December 2020** | | | | | | | | | | |
| **Discharge percentile** | **60%** | **65%** | **70%** | **75%** | **80%** | **85%** | **90%** | **95%** | **99%** | **100%** |
| NWM discharge, m$^3$/s | 0.01 | 0.01 | 0.02 | 0.03 | 0.04 | 0.05 | 0.11 | 0.37 | 1.65 | 26.52 |
| 'No water' classifications > discharge percentile | 18.2% | 18.2% | 0.0% | 0.0% | 0.0% | 0.0% | 0.0% | 0.0% | 0.0% | - |
| 'Low water' classifications > discharge percentile | 71.4% | 71.4% | 36.3% | 25.3% | 9.8% | 0.2% | 0.0% | 0.0% | 0.0% | - |
| 'High water' classifications > discharge percentile | 73.5% | 73.5% | 69.3% | 69.0% | 67.9% | 47.4% | 29.3% | 4.2% | 0.7% | - |

**Table A1.** Percentages of images classified as no, low, and high water that correspond to stage or discharge greater than: percentiles of observed stage at Perry Creek (PEC) from August, 2017 through November, 2023 (top); National Water Model (NWM) discharge at a stream segment overlapping PEC from August, 2017 through August, 2020 (middle); and NWM discharge from January, 2000 through December, 2020 (bottom). All percentages were calculated using the same medium- and high-confidence image classifications from August, 2017 through November, 2023. For NWM data, only image classifications from 2017-18 are used because there is no NWM data after 2020, and there are no images available during 2019 and 2020.



*Author contributions.* S.E.O. and M.C.L conceptualized the research and methodology, and M.C.L. supervised the research. S.E.O. worked with M.C.L. to perform the analysis. G.M. managed the stream sites and described them in the manuscript. S.E.O., G.M., and A.J. processed and visualized the stream data. S.E.O. and M.C.L drafted the manuscript, and S.E.O. and M.C.L. edited the manuscript with contributions from all authors. M.C.L. and F.M.R. managed funding acquisition.

*Competing interests.* The authors declare that they have no conflict of interest.

*Acknowledgements.* S.E.O. thanks the CW3E field team, especially Sarah Burnett and Adolfo Lopez along with support from Duncan Watson-Parris, Konstantine Georgakakos, Ming Pan, Adrian Borsa, Patrick Mulrooney, Agniv Sengupta, and Peter Yao. We also thank CW3E's Russian River cooperators: stakeholders, landowners, and FIRO partners including the U.S. Army Corps of Engineers, Sonoma County Water Agency, and the California Department of Water Resources. This work was funded by a combination of the following: a
Hellman Fellows Program (faculty) award and NSF Award #2205239 (M.C.L., S.E.O.); and the U.S. Army Corps of Engineers Engineer Research and Development Center FIRO program (Award USACE W912HZ-24-2-0001; S.E.O., G.M., A.J., F.M.R., and CW3E field team work on the Russian River streamgage network).



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
