# Peer review of "Imagery classification of stream stage to support ephemeral stream monitoring"

_EGUsphere, 2025_

## Referee Comment (RC2)

**Overall Impression:**

This manuscript presents a timely, and valuable study that addresses a critical challenge in hydrology: monitoring intermittent rivers and ephemeral streams (IRES). The application of a relatively simple logistic regression model to classify flow states from field camera imagery is both pragmatic and innovative. The methodology is clearly described, the results are robust and convincingly presented, and the discussion thoughtfully places the work in the broader context of IRES monitoring, climate change, and water management. The integration of image classifications for quality control of stage data is a particularly strong and practical contribution. The manuscript is generally well-written and structured. I believe it represents a significant contribution to the field and is a strong candidate for publication after revisions.

**Abstract**

The abstract effectively summarizes the study's motivation, methods, key findings, and implications. It clearly states the problem (monitoring challenges in IRES), the proposed solution (image classification), the location, and the broader significance of the work. However, I am including some comments and questions that can improve the abstract.

1. The abstract mentions the model was used for quality control of the stage time series. Could you briefly hint at the nature of the discrepancies/uncertainties/errors found (e.g., sensor drift, noise during high flow) to immediately highlight the practical utility of the method?

2. The term "Imagey" in the title appears to be a typo for "Image-based." Was this intentional?

3. The abstract focuses on categorical classification. Did the model's probability output itself provide any additional, continuous-like insight beyond the three discrete categories?

**Introduction**

The introduction provides a comprehensive and compelling background, effectively building the case for the importance of IRES and the difficulties in monitoring them. The literature review is extensive and covers relevant areas, including remote sensing, citizen science, and various modeling approaches.

1. While you cover technological methods, could you briefly mention the organizational/funding challenges of maintaining sensor networks in remote IRES to further justify the need for low-cost methods?

2. You mention that deep learning has been mainly applied to perennial streams. Could you elaborate on one or two key reasons why these methods are particularly challenging to directly transfer to IRES (e.g., more dynamic channel geometry, greater debris, longer dry periods)?

3. The introduction effectively sets up the use of machine learning with imagery. Would it be valuable to more explicitly state the core hypothesis: that visual features in daytime imagery are sufficient to reliably classify IRES flow states for monitoring purposes?

4. Have you considered citing studies that discuss the hydrological significance of the "pooling" phase in IRES, which your method can detect but stage sensors cannot? For example, Stubbington, R., et al. (2017). The biota of intermittent rivers and ephemeral streams: aquatic and terrestrial assemblages. In *Intermittent Rivers and Ephemeral Streams* (pp. 217-245). Academic Press. could strengthen this point.

5. The transition from the broad introduction to the specific objectives of the study is clear, but could the final paragraph be slightly more structured to explicitly list the primary aims of the paper?

**Methods**

The methods section is exceptionally detailed and reproducible, a major strength of the manuscript. The description of the study site, data sources, image preparation, model training, and validation is thorough. The handling of unbalanced classes and the development of a confidence metric are particularly sophisticated and commendable.

1. You limited image analysis to 9 am–4 pm PST to avoid low-light issues. Was any consideration given to using the camera's flash-illuminated nighttime images for a simple binary "water"/"no water" classification, given that this is a defining feature of IRES?

2. For the image cropping to 1000x1200 pixels, was this specific size determined empirically? Did you experiment with different crop sizes or aspect ratios to optimize feature recognition?

3. The manual weighting scheme (3.5 for water categories, 3 for obstructed, 1 for 'no water') is interesting. Could you provide a sentence on the rationale behind these specific weight values?

4. The confidence level assignment is well-explained but based on qualitative assessment of probability distributions. Were any quantitative metrics (e.g., maximizing Youden's J index) explored to define the probability thresholds more objectively?

5. You use soil moisture data from a nearby site (DRW) with probably different geology. How might this spatial disconnect influence the interpretation of the relationships between soil moisture and stage at PEC? In addition, a section of uncertainties would be great, since I can see some sources of uncertainties in your work/modelling, for example: image quality and environmental variability (the classification model's performance is inherently tied to the quality and consistency of the input imagery), sensor data reliability and spatial mismatch (the "ground truth" data used for validation and comparison are themselves sources of uncertainty), limited training data and site specificity (the model was trained on a relatively small, manually labeled dataset (537 images) from a single site. this raises uncertainty about its performance when transferred to other IRES with different channel morphology, substrate, vegetation, and water clarity. the model's features (e.g., learned from the specific staff plate and rocks at pec) may be overly tailored to this unique location). Do not get me wrong, I still think there is a lot of value in publishing this paper, however, it is good to show the uncertainties and potential bias of the approach.

**Results**

The results are clearly presented, with appropriate use of tables, figures, and statistics. The model performance metrics are convincing, and the comparison between image classifications, observed stage, and modeled discharge is effective in demonstrating the value of the approach.

1. The confusion matrix shows that 'obstructed' images were most often misclassified as 'no water'. Given the ephemerality of the stream, do you think this misclassification might be functionally acceptable in many cases, as it likely reflects a true dry state?

2. Figure 7/9 and the text describe how image classifications identified sensor malfunctions. How many erroneous data points would have been missed without this image-based quality control? A rough percentage or count would powerfully quantify this benefit.

3. In Figure 10, the results show a notable discrepancy where the NWM reported zero discharge during periods of observed high water (e.g., Jan-Feb 2018). What is your leading hypothesis for this systematic underestimation by the NWM in this specific catchment?

4. The relationship between stage and soil moisture at 5 cm is stronger than at 100 cm. Does this suggest that flow at PEC is primarily driven by shallow subsurface flow or saturation-excess overland flow rather than deeper groundwater contributions? If so, that should be discussed, showing how the changes over time may impact the local hydrology of the watershed and river.

5. You mention that high flows remain unmeasured due to safety. Could the image classifications be calibrated against the NWM output or other hydraulic models to provide a rough estimate of discharge during these extreme events?

**Discussion**

The discussion successfully interprets the results, acknowledges limitations, and explores the wider implications of the work. The sections on unique site features and extensibility are particularly thoughtful and elevate the manuscript beyond a simple methods paper.

1. You rightly note that temporal correlation of flow states could be used for further quality control. Could a simple Hidden Markov Model be a natural next step to incorporate this temporal dependency?

2. How does the performance of your logistic regression model (91% accuracy) compare, in your view, to the potential trade-offs of using a more complex but data-hungry model like a CNN for this specific task? That could be a good paragraph in the discussion, showing the drawbacks and positives sides of using a parsimonious but effective model.

3. In the biggening (abstract and objectives) you state the method is "transferable." Could you specify the primary condition for transferability (e.g., the presence of a staff plate or a consistent field of view of the streambed)? Maybe a bit of discussion on the costs of this equipment set up would also help the reader to have an idea of how much it would cost. Perhaps that could be included in the methods?!

4. The discussion on the potential for subsurface flow bypassing the PEC site is fascinating. Could this hypothesis be further supported by comparing the water level in Lake Mendocino with dry/wet periods at PEC?

5. You mention that your code is transferable. What is the minimum number of manually labeled images you would estimate is necessary to achieve reasonable performance at a new, similar site?

6. In the context of climate change, how might your method help in detecting shifts in the timing of flow initiation and cessation in IRES, which is a key impact of warming temperatures?

7. Have you considered referencing studies that have successfully implemented low-cost, image-based methods in data-scarce regions? For instance,

   - Noto, S., Tauro, F., Petroselli, A., Apollonio, C., Botter, G., & Grimaldi, S. (2022). Low-cost stage-camera system for continuous water-level monitoring in ephemeral streams. *Hydrological Sciences Journal*, *67*(9), 1439–1448. https://doi.org/10.1080/02626667.2022.2079415
   - Rodrigues, R. M., Braga, B. B., & Costa, C. A. G. (2025). Efficiency in river discharge measurement: combining Chiu's method with particle image velocimetry techniques. *RBRH*, *30*, e31.

These papers could broaden the perspective on transferability and cost-effectiveness.

**Conclusion**

The conclusion effectively summarizes the main findings and their significance. It compellingly argues for the role of this low-cost method in improving IRES monitoring and, consequently, water management in a changing climate.

1. Could the conclusion more explicitly state the single most important recommendation for water managers seeking to implement this method?

2. Beyond FIRO, can you speculate on one other specific water management decision (e.g., environmental flow allocations, drought contingency planning) that would benefit from the categorical flow data your method provides?

3. While the method is low-cost, the conclusion could acknowledge the ongoing costs and challenges of maintaining field cameras in harsh environments as a consideration for long-term deployment.

4. What do you see as the next critical technological or methodological advancement needed to make IRES monitoring truly scalable across vast river networks?

I congratulate the authors on an excellent piece of work. I hope that with these revisions, the manuscript will be an even stronger contribution to the literature.

---

## Author Comment (AC1)

**Comment from Referee #1 (responses are in blue):**

I find this article to be generally well-written, well-structured, and applies a transferrable methodology to classify stream conditions in ephemeral streams in a single study watershed. The authors support their claims and provide adequate figures to support their argument.

**Thank you for your thoughtful review of this article.**

I did find that some of the discussion sections strayed beyond the scope of the study described in the introduction section to discuss other features of the watershed and ephemeral streams more broadly. The paper would be strengthened by focusing on its central contribution.

These are valid points regarding the discussion sections, and we address them below.

I did find that a limitation of the study was that it focused on a subset of images from a single site. The methodology was demonstrated and its performance evaluated against predictions from the National Water Model, but statements about its transferability to other locations or are undercut by the limited nature of the data.

We appreciate this feedback and propose addressing it through the incorporation of an appendix that demonstrates transferability. The primary purpose of this work was to provide a proof of concept for this method using the Perry Creek site and its unique geographic context. While analysis of other sites in different locations and contexts would be beyond the scope of this paper, we can nevertheless provide a parsimonious set of examples to demonstrate the method's transferability.

We tested the model code (which is posted on the HydroShare repository associated with this paper) on two example sites from the USGS Flow Photo Explorer to produce time series of categorical flow states. These sites are Dry Brook Upper in Massachusetts and USGS streamgage 10247170 on Troy Creek in Nevada. Below, we have included a draft of the proposed new appendix text and figures, which would be referenced in a revised Discussion section, where extensibility and the USGS Flow Photo Explorer in particular are discussed.

**Proposed appendix on transferability to other sites:**

Although the main goal of this work was to demonstrate proof-of-concept at the PEC site, we also tested our model on two additional U.S. sites from the USGS Flow Photo Explorer: Dry Brook Upper in Massachusetts and USGS streamgage 10247170 on Troy Creek in Nevada (USGS, 2024). We selected these sites because they are both IRES with thousands of photos available. After labeling only 105 and 111 photos, respectively, the model achieved 84.4% accuracy at Dry Brook Upper and 76.5% at Troy Creek. The resulting time series of categorical flow states from model predictions (for all confidence levels) are shown in figure A12. This exercise was performed with fewer labeled photos compared to the PEC case, no photo cropping, and no changes to the model code (aside from updating the formatting of dates).

Based on this preliminary transferability analysis, we find that about 100 labeled images – with all categories represented in both training and testing sets – appear sufficient to transfer this method to other sites with consistent imagery. Notably, all photos used in this exercise were taken at noon, which likely enhanced model performance due to minimal variation in sun angle. While additional labeled photos would likely improve performance at any site, those with unbalanced categories or dramatic changes in illumination would benefit most.

Current deep learning models in the USGS Flow Photo Explorer (USGS, 2024) estimate relative flow states but cannot distinguish dry streambeds (Gupta et al., 2022; Goodling et al., 2025). Our method could complement the existing relative streamflow method, for example by being included in a conditional two-step approach: detect water presence first with our simple model; if present, estimate relative discharge using a CNN. This would preserve the simplicity and high accuracy of our model while enabling conditional estimation of streamflow when relevant. This approach would be well-suited to watersheds managed for both water supply and fishery health since both streamflow volume and stream connectivity would affect watershed

management. With hundreds of thousands of photos available on the USGS Flow Photo Explorer (USGS, 2024) and the likelihood of increased IRES prevalence with climate change, this screening for IRES-specific states would be especially valuable. Instructions for applying our methodology to USGS Flow Photo Explorer images are available on HydroShare (see Data and Code Availability statement).

I have minor comments regarding clarity and a few considerations not in the original text but overall find the article a suitable contribution to HESS:

Thank you, I will address these comments below.

1. Page 7, line 155: What defines "environmental damage"? Tampering? Batteries dying?

"Environmental damage" refers to various issues that can affect a field camera, such as the camera breaking due to water damage and the batteries dying due to the solar panel not receiving enough sunlight to charge fully. We propose editing the relevant text to: "... environmental damage, such as water damage or dirty solar panels not generating enough power."

2. **Page 9, Lines 173-180**: The National Water Model (NWM) is trained/calibrated to gage flows, how close is the closest calibration site? In figure A1 looks like it is on the East Fork of Russian River, so not on the stream you are monitoring. Worth pointing out in this section.

Yes, the nearest calibration site is the East Fork Russian River gage (EFR), which is not part of the Perry Creek watershed, but is part of the Lake Mendocino watershed, and is indicated in Figure A1. We propose adding the following sentence to Section 2.1: "Although the NWM was not calibrated using data from the Perry Creek watershed, it was calibrated using data from the USGS East Fork Russian River streamgage (EFR in Fig. A1; Cosgrove et al., 2024), also located within the Lake Mendocino watershed."

3. **Figure 5:** You need axis labels indicating which axis is predicted and which is observed.

Thank you for this suggestion. We will revise the figure to include the labels "Predicted Category" on the horizontal axis and "Observed Category" on the vertical axis.

4. **Page 9, line 197:** Indeed, cropping vegetation may be helpful here – if there is a mediterranean climate, vegetation dynamics and streamflow ephemerality are both highly seasonal the model could learn more from the banks (which could make up more of the image) than the channel where intended.

This is an interesting insight, although its consideration – and the potential for bank vegetation to provide useful information for flow prediction (rather than reduce model performance, as in our case) – is beyond the scope of the present analysis. Nevertheless, we did not crop the images used in the new appendix application of photos from the USGS Flow Photo Explorer (see above), and we propose incorporating

the following point into a revised Discussion section as follows: "Due to there being multiple field camera angles at PEC, we cropped the images to focus on the stream channel and staff plate. However, images do not necessarily need to be cropped, and bank vegetation could potentially help the model predict flow states. For sites with consistent camera types and viewing angles, a useful exercise could be to find the optimal image resolution and cropping extent for feature recognition. In such an exercise, the cost of increased computing power for higher resolution images should be balanced with model performance."

5. **Page 10, line 204:** You only labelled 12.8% of the total images you had available – this is acceptable but is a relatively small dataset for training or reporting performance (your testing set is 3.9% of your total image dataset) that will represent the population. This is a limitation of the study, since as you note the lighting can be very different at different times of day/year. Ideally you have a big enough testing set to represent performance at each class during different lighting (and vegetation and channel) conditions.

Yes, we agree that the number of labeled photos is relatively small. Our intention, in part, was to demonstrate that the model can still perform well even with a limited number of classified photos; this is a situation that is common in data-limited settings. We believe this point is conveyed sufficiently throughout the paper and in our demonstration of strong model performance despite the limited number of labeled photos.

6. Page 10, line 203-210: Random sampling was used in the selection of images for training/testing, which is acceptable, but this means the performance is only representative of historical conditions coincident with the label dataset. The performance reported in this paper is not representative of model prediction on new unseen imagery. This point is worth noting to make sure the reader knows what the model performance represents.

This is correct. Because the present study is limited to imagery from the study period, even "out-of-sample" testing data fall within the study period domain. Broadly, this means that model performance is only representative within the study period. However, to the extent that variation within the study period reflects variation outside of it, model performance during the study period may reasonably be considered indicative of performance under true out-of-period conditions. We thank the reviewer for raising this important point and propose incorporating the following language into Section 2.2.1 (and/or elsewhere, as appropriate): "Because this study is limited to imagery from the study period, our analysis and modeling strictly reflect that period. However, if the variation in imagery and corresponding flow during the study period captures the seasonal and inter-annual variability typical of other years, then the selected images may be considered broadly representative. In our case, the study period includes the full range from wet to dry years and thus arguably captures this variability."

7. **Page 12**, **lines 247-250**: Why were these manual weights selected?

Thank you for noting the omission of our reasoning behind the selection of manual weights. We propose adding the following explanation to this section: "We assigned manual weights to emphasize water presence categories ('high water' and 'low water') over 'no water,' and gave the 'obstructed' category a weight higher than 'no water' -- reflecting its smaller sample size -- but lower than the water categories, given its lesser importance."

8. **Page 14, line 298:** Is there a reference for the 0.028 m3 s-1 threshold for NWM flow? How sensitive are your results to this selection? The selection of the threshold appears arbitrary at the moment.

Thank you for noting this error. We ultimately did not use this threshold. As such, we propose replacing the line with the following description of what our analysis did: "For example, we calculated how often the observed stage at PEC was zero while the NWM predicted flow."

9. Figure 7: Why are there negative stage values? And why are there purple high water dots in panel 7 when stage is reported negative? Is that supposed to be a diagnostic tool for quality assurance of the stage data, which leads to the record in panel b? The paragraph in the main text where Figure 7 is mentioned does not walk the reader through this. Also in Figure 7 are the stage observations without any dots times where there was no imagery or times where the imagery classification was deemed not high confidence? Consider adding shading to indicate "no imagery available" and another color of dot to indicate "no high confidence prediction" or something similar so the absence of data is clear.

Thank you for your comments, which indicate that the placement and discussion of this image were not clear in the manuscript. To address this, we propose moving Figure 7 to the position of Figure 8, so that the relevant methods are discussed before the figure is presented. Otherwise, answers to most of the reviewer's questions are already provided in Section 3.3 and Appendix 2. For remaining questions and omissions, we propose making clarifications in both the figure caption and the surrounding text. Specifically, we propose revising the figure 7 caption to read: "Original and quality-controlled, barometrically compensated stage from the Perry Creek (PEC) site from December 2022-February 2023. No imagery was available after 1 February 2023. Stage values (black lines) are colored (points) by high-confidence image classifications (only). a) Shows the time series before quality control, and b) shows the time series after quality control." In addition, we propose emphasizing in Section 3.3 that observed stage data can be "prone to sensor error". Finally, we propose adding text describing the negative stage values in more detail in Section 3.3: "Noisy data and stage measurements less than zero were an issue before installing the HOBO MX2001-04-SS-S pressure transducer and HOBO MicroRX data logger in October 2023; thus, the image classifications were useful in validating when the observed stage should be zero."

10. Page 28, line 446-448 and Page 29, line 464-465: Is there a citation or the claim of not having enough imagery to train a CNN? The Gupta et al. and Noto et al. studies you cite have about as much imagery as you do. You report ~4,700 images, which is more than at 2 of Gupta et al. 's sites.

We agree that our language misstates the point and is overly general. Gupta et al. found that using a reduced number of *labeled* image pairs (500–1,000) resulted in worse performance. In our study, we used 537 labeled images, even though the total number of available images -- both in our case and in Gupta et al.'s -- was higher. We intentionally limited the number of labeled images to evaluate model accuracy under more constrained conditions. Accordingly, we agree that the discussion of CNNs is not particularly helpful, as we did not explicitly evaluate CNN performance or its relationship to training size in our study. Therefore, we propose removing the references to CNNs and their sample size requirements.

11. **Section 4.4:** This section largely diverges from the central contribution of the study (a methodology to classify images) and into a lot of site-specific information that is largely conjecture about processes and reads as redundant to the prior section (4.3). This section could be eliminated.

Thank you for your input. Upon review, we agree with your recommendation. We propose moving this section to an appendix and referencing it in the main text where appropriate.

12. **Page 33**, **line 604**: Is there a citation for the claim that "these efforts have struggled to translate to IRES"? Neither study cited included nonperennial streams.

We agree, this should be reworded to: "these efforts did not focus on IRES".

13. **Section 4.6:** This section is only loosely connected with the central contribution of the paper (image classification model) and is material that could be included in the introductory material. This section could be eliminated.

We agree that the material in this section is better suited for partial incorporation into the introduction, as well as inclusion in a new "Conclusion" section (in response to your comment below).

14. **Conclusion section is missing:** It is traditional to summarize the paper's contribution in a conclusion section, one is missing here.

We propose including a new Conclusion section that summarizes the overall contribution, and incorporates salient points from Section 4.6 (in response to your comment above):

"This work demonstrates that a simple machine learning algorithm can classify timelapse field camera images to identify no, low, or high water levels in IRES, providing a

low-cost, transferable method for monitoring water occurrence in these sparsely observed systems. Given the prevalence of ungaged IRES, field cameras and image classification offer a practical approach to improving understanding of their role in climate-impacted freshwater systems. For example, the FIRO program at Lake Mendocino (Fig. A1) currently uses streamflow observations from EFR to inform reservoir inflow models. As climate change is expected to increase drying in IRES, unmonitored contributions from tributaries such as Perry Creek (Appendix 3) could affect reservoir storage. Thus, as the FIRO program expands, field cameras and image classification may offer a cost-effective approach to integrating information on the presence and magnitude of IRES contributions.

This approach can also support monitoring of critical habitats, including tributaries where salmon passage depends on streamflow connectivity threatened by drought and water diversions (see e.g., Scott River, 2025). Installing cameras at tributary confluences could inform targeted habitat restoration. More broadly, formally recognizing IRES as integral to river systems can incentivize monitoring and protect them from degradation due to climate change, mining, and urban development (Acuña et al., 2014). The 2023 exclusion of ephemeral streams from U.S. Clean Water Act protections highlights the vulnerability of IRES and the importance of cost-effective monitoring approaches like ours for understanding the impacts and effectiveness of water management efforts.

We conclude by offering practical recommendations for implementing our method. Cameras should be installed along IRES reaches that are important for monitoring water management objectives (e.g., fish passage, drought contingency planning). Installations should be in stable locations with clear views of the streambed and minimal vegetation interference. Consistent camera types and viewing angles should be used, as they improve the robustness of time series and the effectiveness of classification. Long-term maintenance budgets are also recommended to support sustained monitoring. Finally, this approach can be integrated with complementary methods (Gupta et al., 2022; Goodling et al., 2025) and deployed through accessible platforms such as the USGS Flow Photo Explorer (USGS, 2024) and the CrowdWater mobile application (SPOTTERON GmbH, 2025)."

---

## Author Comment (AC2)

**Comments from Referee #2 (responses are in blue):**

**Overall Impression:**

This manuscript presents a timely, and valuable study that addresses a critical challenge in hydrology: monitoring intermittent rivers and ephemeral streams (IRES). The application of a relatively simple logistic regression model to classify flow states from field camera imagery is both pragmatic and innovative. The methodology is clearly described, the results are robust and convincingly presented, and the discussion thoughtfully places the work in the broader context of IRES monitoring, climate change, and water management. The integration of image classifications for quality control of stage data is a particularly strong and practical contribution. The manuscript is generally well-written and structured. I believe it represents a significant contribution to the field and is a strong candidate for publication after revisions.

Thank you for your thoughtful review.

**Abstract**

The abstract effectively summarizes the study's motivation, methods, key findings, and implications. It clearly states the problem (monitoring challenges in IRES), the proposed solution (image classification), the location, and the broader significance of the work. However, I am including some comments and questions that can improve the abstract.

1. The abstract mentions the model was used for quality control of the stage time series. Could you briefly hint at the nature of the discrepancies/uncertainties/errors found (e.g., sensor drift, noise during high flow) to immediately highlight the practical utility of the Method?

Yes, we propose adding the following to the abstract: "We then used image classifications to perform quality control on the continuous stage time series, which allowed us to identify when the stream was dry and when the sensor malfunctioned."

- 2. The term "Imagey" in the title appears to be a typo for "Image-based." Was this Intentional?
- "Imagery" was intentional, but upon review, we agree that "Image-based" more correctly emphasizes that we are using images as an input for classification, so we propose using "Image-based" instead. Thank you.
- 3. The abstract focuses on categorical classification. Did the model's probability output itself provide any additional, continuous-like insight beyond the three discrete categories? Although we did not explore the probability results beyond what was relevant to our categorical classifications (i.e., Figure 6), we did observe some overlap between 'low water' and 'high water' image classifications, and a range of corresponding probabilities assigned to these classes. For example, some photos labeled as "high water" were classified as "low water" and

vice versa. Nevertheless, we ultimately evaluated only the probability distributions for 'any water' (the combination of low and high water; Figure 6).

**Introduction**

The introduction provides a comprehensive and compelling background, effectively building the case for the importance of IRES and the difficulties in monitoring them. The literature review is extensive and covers relevant areas, including remote sensing, citizen science, and various modeling approaches.

- 1. While you cover technological methods, could you briefly mention the organizational/funding challenges of maintaining sensor networks in remote IRES to further justify the need for low-cost methods?
- Yes, we mention this in line 60: "In addition to inaccessibility, developing an in-situ monitoring network for stage and discharge on IRES is difficult because nascent gage networks may have less expertise, support, or funding compared to established national programs that generally focus on perennial streams (Vlah et al., 2023)."
- 2. You mention that deep learning has been mainly applied to perennial streams. Could you elaborate on one or two key reasons why these methods are particularly challenging to directly transfer to IRES (e.g., more dynamic channel geometry, greater debris, longer dry periods)?

We mention deep learning approaches in the Introduction (I. 101-111) and the Discussion (I. 463-465), and we propose adding the following to the Discussion: "Current deep learning models in the USGS Flow Photo Explorer (USGS, 2024) estimate relative flow states but cannot distinguish dry streambeds (Gupta et al., 2022; Goodling et al., 2025), potentially due to the dynamic channel morphology, shifting debris and vegetation, and ambiguous flow states of IRES – all of which can make training deep learning models challenging."

- 3. The introduction effectively sets up the use of machine learning with imagery. Would it be valuable to more explicitly state the core hypothesis: that visual features in daytime imagery are sufficient to reliably classify IRES flow states for monitoring purposes? Thank you for this suggestion. We propose replacing "Here, we explore the use of image classification for categorizing water levels in IRES" with "Here, we explore whether visual features in daytime imagery are sufficient to reliably classify IRES flow states for monitoring purposes."
- 4. Have you considered citing studies that discuss the hydrological significance of the "pooling" phase in IRES, which your method can detect but stage sensors cannot? For example, Stubbington, R., et al. (2017). The biota of intermittent rivers and ephemeral streams: aquatic and terrestrial assemblages. In Intermittent Rivers and Ephemeral Streams (pp. 217-245). Academic Press. could strengthen this point.

Thank you for highlighting the importance of the pooling phase in relation to biota in IRES. We discuss this later on in the Discussion section (4.1), and also propose including your suggested reference in that discussion..

5. The transition from the broad introduction to the specific objectives of the study is clear, but could the final paragraph be slightly more structured to explicitly list the primary aims of the paper?

Yes, we propose adding the hypothesis from your point #3 to this paragraph to make it more explicit.

**Methods**

The methods section is exceptionally detailed and reproducible, a major strength of the manuscript. The description of the study site, data sources, image preparation, model training, and validation is thorough. The handling of unbalanced classes and the development of a confidence metric are particularly sophisticated and commendable.

1. You limited image analysis to 9 am—4 pm PST to avoid low-light issues. Was any consideration given to using the camera's flash-illuminated nighttime images for a simple binary "water"/"no water" classification, given that this is a defining feature of IRES?

Yes, we address this directly in Section 2.2.1 and again in the Discussion section.

2. For the image cropping to 1000x1200 pixels, was this specific size determined empirically? Did you experiment with different crop sizes or aspect ratios to optimize feature recognition?

We describe the method for preparing images in Section 2.2.1. We propose adding the following to that section: "To apply our method, all images were required to have the same dimensions. We selected a resolution of  $1,000 \times 1,200$  pixels (fig. 3a) because it was low enough to ensure that each image focused on the staff plate and streambed. Thus, the image size was determined by resolution constraints rather than through empirical or experimental testing." In addition, we propose adding the following to the Discussion: "For sites with consistent camera types and viewing angles, a useful exercise could be to find the optimal image resolution and cropping extent for feature recognition. In such an exercise, the cost of increased computing power for higher resolution images should be balanced with model performance."

3. The manual weighting scheme (3.5 for water categories, 3 for obstructed, 1 for 'no water') is interesting. Could you provide a sentence on the rationale behind these specific weight values?

Thank you for noting the omission of our reasoning behind the selection of manual weights. We propose adding the following explanation to this section: "We assigned manual weights to emphasize water presence categories ('high water' and 'low water') over 'no water,' and gave

the 'obstructed' category a weight higher than 'no water' -- reflecting its smaller sample size -- but lower than the water categories, given its lesser importance."

- 4. The confidence level assignment is well-explained but based on qualitative assessment of probability distributions. Were any quantitative metrics (e.g., maximizing Youden's J index) explored to define the probability thresholds more objectively? This is an interesting question, and we agree that we could have experimented with methods further in this case. However, because our primary objective was avoiding false positive classifications, the distribution (boxplot) assessment approach met our needs in this case. We agree, however, that a more objective strategy might be applicable in a generalized version of this study, and propose adding the following in the discussion: "In addition, a more objective strategy for evaluating classification confidence for other sites could be developed."
- 5. You use soil moisture data from a nearby site (DRW) with probably different geology. How might this spatial disconnect influence the interpretation of the relationships between soil moisture and stage at PEC?

  We discuss the differences in the DRW vs. PEC site, including soil features, in Section 4.3, I. 432-441, and propose adding the following to that section: "Specifically, the soil hydrologic group at DRW is Group B, which indicates moderate infiltration rates, while the PEC watershed contains Groups C and D, which indicate slow or very slow infiltration rates, respectively (SSURGO, 2024)."

In addition, a section of uncertainties would be great, since I can see some sources of uncertainties in your work/modelling, for example: image quality and environmental variability (the classification model's performance is inherently tied to the quality and consistency of the input imagery), sensor data reliability and spatial mismatch (the "ground truth" data used for validation and comparison are themselves sources of uncertainty), limited training data and site specificity (the model was trained on a relatively small, manually labeled dataset (537 images) from a single site. This raises uncertainty about its performance when transferred to other IRES with different channel morphology, substrate, vegetation, and water clarity. the model's features (e.g., learned from the specific staff plate and rocks at pec) may be overly tailored to this unique location). Do not get me wrong, I still think there is a lot of value in publishing this paper, however, it is good to show the uncertainties and potential bias of the approach.

The issue of transferability to other sites was also raised by Reviewer #1, and we appreciate you also bringing in the related concept of uncertainty, and how that relates to transferability.

The primary purpose of this work was to provide a proof of concept for this method using the Perry Creek site and its unique geographic context. Accordingly, we used a model that outputs prediction probabilities, allowing uncertainty in the predictions to be explicitly represented. We expect that both the distribution of probabilities and the assignment of confidence levels would likely vary across sites, making classification uncertainty primarily a within-site issue. Exploring this variation would require cross-site comparisons, which are beyond the scope of the present

study. We can nevertheless provide a parsimonious set of examples to demonstrate the method's transferability.

We tested the model code (which is posted on the HydroShare repository associated with this paper) on two example sites from the USGS Flow Photo Explorer to produce time series of categorical flow states. These sites are Dry Brook Upper in Massachusetts and USGS streamgage 10247170 on Troy Creek in Nevada. Below, we have included a draft of a proposed new appendix text and figures, which would be referenced in a revised Discussion section, where extensibility and the USGS Flow Photo Explorer in particular are discussed.

**Proposed appendix on transferability to other sites:**

Although the main goal of this work was to demonstrate proof-of-concept at the PEC site, we also tested our model on two additional U.S. sites from the USGS Flow Photo Explorer: Dry Brook Upper in Massachusetts and USGS streamgage 10247170 on Troy Creek in Nevada (USGS, 2024). We selected these sites because they are both IRES with thousands of photos available. After labeling only 105 and 111 photos, respectively, the model achieved 84.4% accuracy at Dry Brook Upper and 76.5% at Troy Creek. The resulting time series of categorical flow states from model predictions (for all confidence levels) are shown in figure A12. This exercise was performed with fewer labeled photos compared to the PEC case, no photo cropping, and no changes to the model code (aside from updating the formatting of dates).

Based on this preliminary transferability analysis, we find that about 100 labeled images — with all categories represented in both training and testing sets — appear sufficient to transfer this method to other sites with consistent imagery. Notably, all photos used in this exercise were taken at noon, which likely enhanced model performance due to minimal variation in sun angle. While additional labeled photos would likely improve performance at any site, those with unbalanced categories or dramatic changes in illumination would benefit most.

Current deep learning models in the USGS Flow Photo Explorer (USGS, 2024) estimate relative flow states but cannot distinguish dry streambeds (Gupta et al., 2022; Goodling et al., 2025). Our method could complement the existing relative streamflow method, for example by being included in a conditional two-step approach: detect water presence first with our simple model; if present, estimate relative discharge using a CNN. This would preserve the simplicity and high accuracy of our model while enabling conditional estimation of streamflow when relevant. This approach would be well-suited to watersheds managed for both water supply and fishery health since both streamflow volume and stream connectivity would affect watershed management. With hundreds of thousands of photos available on the USGS Flow Photo Explorer (USGS, 2024) and the likelihood of increased IRES prevalence with climate change, this screening for IRES-specific states would be especially valuable. Instructions for applying our methodology to USGS Flow Photo Explorer images are available on HydroShare (see Data and Code Availability statement).

**Results**

The results are clearly presented, with appropriate use of tables, figures, and statistics. The model performance metrics are convincing, and the comparison between image classifications, observed stage, and modeled discharge is effective in demonstrating the value of the approach.

- 1. The confusion matrix shows that 'obstructed' images were most often misclassified as 'no water'. Given the ephemerality of the stream, do you think this misclassification might be functionally acceptable in many cases, as it likely reflects a true dry state? Yes, this is a good point. We mention this in the Results section at I. 325-326 with: "In addition, 'obstructed' images were occasionally misclassified as either 'high water' or 'no water' (1.1% and 1.8% of these classifications, respectively). Due to the stream's ephemerality, it is likely there was in fact no water at PEC in the 'obstructed' images misclassified as 'no water'."
- 2. Figure 7/9 and the text describe how image classifications identified sensor malfunctions. How many erroneous data points would have been missed without this image-based quality control? A rough percentage or count would powerfully quantify this benefit.

We describe our quality control of data (i.e., removal of erroneous stage data) at I. 370-383 and Appendix 2; we also propose revising Section 3.3 to improve clarity based on suggestions from Reviewer #1. Therein, we did not specifically calculate how many erroneous data points would have been removed using some other method, e.g., simple visual inspection or use of a value threshold vs. image-based quality control, but instead note that the image-based approach could either replace other methods or augment them. Even so, we propose adding to Section 3.3 that "we flagged and removed almost a month of these data from the record".

3. In Figure 10, the results show a notable discrepancy where the NWM reported zero discharge during periods of observed high water (e.g., Jan-Feb 2018). What is your leading hypothesis for this systematic underestimation by the NWM in this specific Catchment?

In Section 4.2, we discuss reasons for the discrepancies between the NWM, observed discharge, and image classifications that are shown in Figure 10. To respond further to your question, we propose adding to Section 4.2: "Many 'high-water' image classifications occur during January - February 2018, when the NWM shows little discharge. This is illustrated by two manual discharge measurements from January 2018 (Figure 10), which record substantially higher flows at PEC than those simulated by the NWM. We hypothesize that the NWM struggles to represent early wet season flow processes in the steep slopes and low-infiltration soils of the PEC watershed. Later in the season, when soils in the PEC watershed are likely more saturated, the NWM discharge aligns more closely with PEC stage data and manual measurements, suggesting that the model performs better under saturated conditions."

4. The relationship between stage and soil moisture at 5 cm is stronger than at 100 cm. Does this suggest that flow at PEC is primarily driven by shallow subsurface flow or saturation-excess overland flow rather than deeper groundwater contributions? If so,

that should be discussed, showing how the changes over time may impact the local hydrology of the watershed and river.

Yes, we discuss the relationship between stage and soil moisture in Section 4.3, I. 526-532, and propose adding to that discussion: "This, combined with our understanding of the geology of the PEC watershed, suggests that runoff generation at PEC is primarily driven by shallow subsurface flow and saturation-excess overland flow."

5. You mention that high flows remain unmeasured due to safety. Could the image classifications be calibrated against the NWM output or other hydraulic models to provide a rough estimate of discharge during these extreme events?

We interpret this question as asking whether the images and NWM (or another model) could be used to estimate (continuous) discharge during extreme events, given the lack of observed discharge data. This would be beyond the scope of our study. Our model was not trained to predict continuous flow, as some previous studies have done (see references to Gupta et al., 2022 in the Introduction and Discussion sections). Instead, our model predicts categorical flow states, which we then compared to NWM discharge values and the limited available discharge observations. Because we have very few observed discharge measurements (as described in Section 3.3 I. 402-405), we cannot calibrate or validate hydrologic model estimates of discharge. Consequently, linking image classifications to potentially uncertain modeled discharge estimates would not provide meaningful results in this case.

**Discussion**

The discussion successfully interprets the results, acknowledges limitations, and explores the wider implications of the work. The sections on unique site features and extensibility are particularly thoughtful and elevate the manuscript beyond a simple methods paper.

- 1. You rightly note that temporal correlation of flow states could be used for further quality control. Could a simple Hidden Markov Model be a natural next step to incorporate this temporal dependency?
- Thank you for this question. There are a number of different methods that might be used to incorporate temporal dependency. Because evaluation of the suitability of methods for this is beyond the scope of this paper, we declined to list potential methods to avoid confusion.
- 2. How does the performance of your logistic regression model (91% accuracy) compare, in your view, to the potential trade-offs of using a more complex but data-hungry model like a CNN for this specific task? That could be a good paragraph in the discussion, showing the drawbacks and positives sides of using a parsimonious but effective model. In Section 4.1, I. 447, we describe that our method "prioritizes a simple, accurate, site-specific model that requires minimal manually-labeled training data." We also note in Section 4.1, I. 463-465, that a CNN may be more suitable for images classified as 'low water' or 'high water'. In accordance with this suggestion, as well as prior feedback from both you and Reviewer #1,

we propose including a discussion of potential applications of our model, in conjunction with CNN models, within a new appendix focused on model transferability (see above).

3. In the biggening (abstract and objectives) you state the method is "transferable." Could

- you specify the primary condition for transferability (e.g., the presence of a staff plate or a consistent field of view of the streambed)? Maybe a bit of discussion on the costs of this equipment set up would also help the reader to have an idea of how much it would cost. Perhaps that could be included in the methods?! In accordance with suggestions made by you and Reviewer #1, we proposed editing relevant sections of the Discussion and adding a new appendix that specifically addresses transferability (see above). Furthermore, with respect to the cost of setup, we propose adding a brief text appendix that details our setup costs, and propose referencing that appendix in a revised Section 4.5, I. 605-606 where we mention the low cost of our method: "The (2025) cost of field cameras similar to those used in this study range from €100 - €300. The mounting accessories and telemetry equipment add about €100, though costs may vary depending on specific hardware choices. The telemetry system enables near real-time image access but requires an annual renewal fee of about €70. Total installation costs can vary considerably depending on site accessibility and labor expenses. Sites typically require biannual servicing to maintain a consistent power supply, clear vegetation that could obstruct the camera's view of the stream, and to perform routine maintenance."
- 4. The discussion on the potential for subsurface flow bypassing the PEC site is fascinating. Could this hypothesis be further supported by comparing the water level in Lake Mendocino with dry/wet periods at PEC?

  We agree that this hypothesis could certainly be explored further, and observations at PEC and other sites in the same watershed suggest that exploration may be worthwhile. However, that exploration is outside the scope of this work.
- 5. You mention that your code is transferable. What is the minimum number of manually labeled images you would estimate is necessary to achieve reasonable performance at a new, similar site?
- Based on suggestions made by you and Reviewer #1, we proposed adding a new appendix that specifically addresses transferability, including discussion of performance with a minimal number of manually labeled images (see above).
- 6. In the context of climate change, how might your method help in detecting shifts in the timing of flow initiation and cessation in IRES, which is a key impact of warming Temperatures?

As described in Section 4.1, I. 457-459, this method is well-suited for IRES since it focuses on water presence or absence. Therefore, it is well-suited to identify the timing of flow initiation and cessation in IRES. We propose making this point more explicit by adding the following to this same Discussion section: "Our method could be expanded to detect IRES-relevant states including wet streambeds, isolated pools, standing water, trickling water, snow, or ice." We discuss the context of climate change in the Introduction, and describe how our method

supports IRES monitoring under climate change in the proposed new conclusion and transferability appendix.

- 7. Have you considered referencing studies that have successfully implemented low-cost, image-based methods in data-scarce regions? For instance,
- Noto, S., Tauro, F., Petroselli, A., Apollonio, C., Botter, G., & Grimaldi, S. (2022). Low-cost stage-camera system for continuous water-level monitoring in ephemeral streams. Hydrological Sciences Journal, 67(9), 1439–1448. <a href="https://doi.org/10.1080/02626667.2022.2079415">https://doi.org/10.1080/02626667.2022.2079415</a>
  Yes, we cite this work.
- Rodrigues, R. M., Braga, B. B., & Costa, C. A. G. (2025). Efficiency in river discharge measurement: combining Chiu's method with particle image velocimetry techniques. RBRH, 30, e31.

We have not cited this work, but we propose referencing it in the introduction section discussing image-based approaches to monitoring IRES.

These papers could broaden the perspective on transferability and cost-effectiveness.

**Conclusion**

The conclusion effectively summarizes the main findings and their significance. It compellingly argues for the role of this low-cost method in improving IRES monitoring and, consequently, water management in a changing climate.

1. Could the conclusion more explicitly state the single most important recommendation for water managers seeking to implement this method?

In response to this and feedback from Reviewer #1, we propose including a new Conclusion section that summarizes the overall contribution (see below), and incorporates discussion of practical recommendations:

"This work demonstrates that a simple machine learning algorithm can classify timelapse field camera images to identify no, low, or high water levels in IRES, providing a low-cost, transferable method for monitoring water occurrence in these sparsely observed systems. Given the prevalence of ungaged IRES, field cameras and image classification offer a practical approach to improving understanding of their role in climate-impacted freshwater systems. For example, the FIRO program at Lake Mendocino (Fig. A1) currently uses streamflow observations from EFR to inform reservoir inflow models. As climate change is expected to increase drying in IRES, unmonitored contributions from tributaries such as Perry Creek (Appendix 3) could affect reservoir storage. Thus, as the FIRO program expands, field cameras and image classification may offer a cost-effective approach to integrating information on the presence and magnitude of IRES contributions.

This approach can also support monitoring of critical habitats, including tributaries where salmon passage depends on streamflow connectivity threatened by drought and water diversions (see e.g., Scott River, 2025). Installing cameras at tributary confluences could inform targeted habitat restoration. More broadly, formally recognizing IRES as integral to river systems can incentivize monitoring and protect them from degradation due to climate change, mining, and urban development (Acuña et al., 2014). The 2023 exclusion of ephemeral streams from U.S. Clean Water Act protections highlights the vulnerability of IRES and the importance of cost-effective monitoring approaches like ours for understanding the impacts and effectiveness of water management efforts.

We conclude by offering practical recommendations for implementing our method. Cameras should be installed along IRES reaches that are important for monitoring water management objectives (e.g., fish passage, drought contingency planning). Installations should be in stable locations with clear views of the streambed and minimal vegetation interference. Consistent camera types and viewing angles should be used, as they improve the robustness of time series and the effectiveness of classification. Long-term maintenance budgets are also recommended to support sustained monitoring. Finally, this approach can be integrated with complementary methods (Gupta et al., 2022; Goodling et al., 2025) and deployed through accessible platforms such as the USGS Flow Photo Explorer (USGS, 2024) and the CrowdWater mobile application (SPOTTERON GmbH, 2025)."

- 2. Beyond FIRO, can you speculate on one other specific water management decision (e.g., environmental flow allocations, drought contingency planning) that would benefit from the categorical flow data your method provides?

  Yes, our proposed new conclusion section discusses monitoring for salmon migration (see above).
- 3. While the method is low-cost, the conclusion could acknowledge the ongoing costs and challenges of maintaining field cameras in harsh environments as a consideration for long-term deployment.

Yes, our proposed new conclusion section recommends including a budget for site maintenance in all long-term monitoring programs.

4. What do you see as the next critical technological or methodological advancement needed to make IRES monitoring truly scalable across vast river networks? While the broader topic of scaling IRES monitoring approaches like ours is beyond the scope of our paper, we propose addressing this briefly at the end of our new conclusion section, and expanding on this in the new appendix (see above).

I congratulate the authors on an excellent piece of work. I hope that with these revisions, the manuscript will be an even stronger contribution to the literature.

Thank you for your thorough review. The revisions based on your feedback will undoubtedly strengthen the manuscript.

---

## Author Response (AR1)

**Point-by-Point Reply to Comments**

The reviewers and editor requested moderate-to-major revisions, particularly regarding transferability, novelty, and methodology. In preparing this revised submission, we aimed to fully address these concerns as well as all reviewer comments. Our point-by-point response is organized around the two reviewer reports and is consistent with our previously submitted responses. The revised manuscript reflects substantial changes throughout. In particular, we have significantly revised and restructured the Results, Discussion, and Conclusion sections to improve clarity and coherence. In other parts of the paper, including the Introduction and Methods, we clarified methodological details and strengthened descriptions. We also corrected various minor issues, including typographical and/or grammatical errors, updated affiliations, and improved overall readability.

To address the concerns about transferability, we further emphasize the purpose and value of our single-site focus as a necessary and informative demonstration, and we discuss in greater depth when and where site-specific features of our study do or do not influence extensibility. We then add a detailed new appendix that provides supporting analyses showing the application of our model code at two additional external sites with publicly available image data; this new material is referenced in the revised Discussion section.

Below, we respond individually to each reviewer's comment and describe how we incorporated the suggested revisions into the present revised draft.

**Responses to Referee Comments**

For the two referee comments, we include their comments and show our responses in blue.

**Referee #1**

I find this article to be generally well-written, well-structured, and applies a transferrable methodology to classify stream conditions in ephemeral streams in a single study watershed. The authors support their claims and provide adequate figures to support their argument.

Thank you for your thoughtful review of this article.

I did find that some of the discussion sections strayed beyond the scope of the study described in the introduction section to discuss other features of the watershed and ephemeral streams more broadly. The paper would be strengthened by focusing on its central contribution.

These are valid points regarding the discussion sections, and we address them below.

I did find that a limitation of the study was that it focused on a subset of images from a single site. The methodology was demonstrated and its performance evaluated against predictions from the National Water Model, but statements about its transferability to other locations or are undercut by the limited nature of the data.

We appreciate this feedback and have addressed it through the incorporation of a new Appendix A4 "Transferability to other sites" that demonstrates transferability. The primary purpose of this work was to provide a proof of concept for this method using the Perry Creek site and its unique geographic context. While analysis of other sites in different locations and contexts would be beyond the scope of this paper, the new appendix nevertheless provides a parsimonious set of examples to demonstrate the method's transferability. In this appendix, we tested the model code (which is posted on the HydroShare repository associated with this paper) on two example sites from the USGS Flow Photo Explorer to produce time series of categorical flow states. These sites are Dry Brook Upper in Massachusetts and USGS streamgage 10247170 on Troy Creek in Nevada. This new appendix is referenced in the revised Discussion section at l. 608, and also references a new Appendix Figure 12, which presents categorical flow state time series from model predictions for both sites.

I have minor comments regarding clarity and a few considerations not in the original text but overall find the article a suitable contribution to HESS:

Thank you, I will address these comments below.

1. **Page 7, line 155:** What defines "environmental damage"? Tampering? Batteries dying?

   "Environmental damage" referred to various issues that can affect a field camera, such as the camera breaking due to water damage and the batteries dying due to the solar panel not receiving enough sunlight to charge fully. To be clearer, we changed "environmental damage" to "environmental exposure", and edited the relevant text (l. 170) to: "In general, field equipment is susceptible to failure, including problems due to environmental exposure, such as water damage or reduced power generation from dirty solar panels."

2. **Page 9, Lines 173-180**: The National Water Model (NWM) is trained/calibrated to gage flows, how close is the closest calibration site? In figure A1 looks like it is on the East Fork of Russian River, so not on the stream you are monitoring. Worth pointing out in this section.

   Yes, the nearest calibration site is the East Fork Russian River gage (EFR), which is not part of the Perry Creek watershed, but is part of the Lake Mendocino watershed, and is indicated in Figure A1. We added the following sentence to Section 2.1 (l. 194): "Although the NWM was not calibrated using data from the Perry Creek watershed, it was calibrated using data from the USGS East Fork Russian River streamgage (EFR in Fig. A1; Cosgrove et al., 2024), also located within the Lake Mendocino watershed."

3. **Figure 5:** You need axis labels indicating which axis is predicted and which is observed.

Thank you for this suggestion. We revised Figure 5 to include the labels "Predicted Category" on the horizontal axis and "Observed Category" on the vertical axis.

4. **Page 9, line 197:** Indeed, cropping vegetation may be helpful here – if there is a mediterranean climate, vegetation dynamics and streamflow ephemerality are both highly seasonal the model could learn more from the banks (which could make up more of the image) than the channel where intended.

   This is an interesting insight, although its consideration – and the potential for bank vegetation to provide useful information for flow prediction (rather than reduce model performance, as in our case) – is beyond the scope of the present analysis. Nevertheless, we did not crop the images used in Appendix A4, and we incorporated the following point into a revised Discussion section (l. 594-599): "Due to there being multiple field camera angles at PEC, we cropped the images to focus on the stream channel and staff plate. However, images do not necessarily need to be cropped, and bank vegetation could potentially help the model predict flow states. For sites with consistent camera types and viewing angles, a useful exercise could be to find the optimal image resolution and cropping extent for feature recognition. In such an exercise, the cost of increased computing power for higher resolution images should be balanced with model performance."

5. **Page 10, line 204:** You only labelled 12.8% of the total images you had available – this is acceptable but is a relatively small dataset for training or reporting performance (your testing set is 3.9% of your total image dataset) that will represent the population. This is a limitation of the study, since as you note the lighting can be very different at different times of day/year. Ideally you have a big enough testing set to represent performance at each class during different lighting (and vegetation and channel) conditions.

   Yes, we agree that the number of labeled photos is relatively small. Our intention, in part, was to demonstrate that the model can still perform well even with a limited number of classified photos; this is a situation that is common in data-limited settings. We believe this point is conveyed sufficiently throughout the paper and in our demonstration of strong model performance despite the limited number of labeled photos.

6. **Page 10, line 203-210:** Random sampling was used in the selection of images for training/testing, which is acceptable, but this means the performance is only representative of historical conditions coincident with the label dataset. The performance reported in this paper is not representative of model prediction on new unseen imagery. This point is worth noting to make sure the reader knows what the model performance represents.

   This is correct. Because the present study is limited to imagery from the study period, even "out-of-sample" testing data fall within the study period domain. Broadly, this means that model performance is only representative within the study period. However, to the extent that variation within the study period reflects variation outside of it, model

performance during the study period may reasonably be considered indicative of performance under true out-of-period conditions. We thank the reviewer for raising this important point and incorporated the following language into Section 2.2.1 (l. 232-235): "Because this study is limited to imagery from the study period, our analysis and modeling strictly reflect that period. However, if the variation in imagery and corresponding flow during the study period captures the seasonal and inter-annual variability typical of other years, then the selected images may be considered broadly representative. In our case, the study period includes the full range from wet to dry years and thus arguably captures this variability."

7. **Page 12, lines 247-250:** Why were these manual weights selected?

Thank you for noting the omission of our reasoning behind the selection of manual weights. We added the following explanation to this section (l. 271-273): "We assigned manual weights to emphasize water presence categories ('high water' and 'low water') over 'no water,' and gave the 'obstructed' category a weight higher than 'no water' (reflecting its smaller sample size) but lower than the water categories, given its lesser importance."

8. **Page 14, line 298:** Is there a reference for the 0.028 $m^3$ $s^{-1}$ threshold for NWM flow? How sensitive are your results to this selection? The selection of the threshold appears arbitrary at the moment.

Thank you for noting this error. We ultimately did not use this threshold. As such, we replaced the line with the following description of what our analysis did (l. 321-322): "For example, we calculated how often the observed stage at PEC was zero while the NWM predicted flow."

9. **Figure 7:** Why are there negative stage values? And why are there purple high water dots in panel 7 when stage is reported negative? Is that supposed to be a diagnostic tool for quality assurance of the stage data, which leads to the record in panel b? The paragraph in the main text where Figure 7 is mentioned does not walk the reader through this. Also in Figure 7 are the stage observations without any dots times where there was no imagery or times where the imagery classification was deemed not high confidence? Consider adding shading to indicate "no imagery available" and another color of dot to indicate "no high confidence prediction" or something similar so the absence of data is clear.

Thank you for your comments, which indicate that the placement and discussion of this image were not clear in the manuscript. To address this, we moved Figure 7 to the position of Figure 8, so that the relevant methods are discussed before the figure is presented. Otherwise, answers to most of the reviewer's questions are already provided in Section 3.3 and Appendix 2. For remaining questions and omissions, we made clarifications in both the figure caption and the surrounding text. Specifically, we revised the Figure 8 caption to read: "Stage from the Perry Creek (PEC) site from December

2022-February 2023. No imagery was available after 1 February 2023. Stage values (black lines) are colored (points) by high-confidence image classifications (only). a) Shows the time series before quality control, and b) shows the time series after quality control." In addition, we emphasized in Section 3.3 (l. 405) that observed stage data can be "prone to sensor error". Finally, we added text describing the negative stage values in more detail in Section 3.3 (l. 411-413): "Noisy data and stage measurements less than zero were an issue before installing the HOBO MX2001-04-SS-S pressure transducer and HOBO MicroRX data logger in October 2023; thus, the image classifications were useful in validating when the observed stage should be zero."

10. **Page 28, line 446-448 and Page 29, line 464-465:** Is there a citation or the claim of not having enough imagery to train a CNN? The Gupta et al. and Noto et al. studies you cite have about as much imagery as you do. You report ~4,700 images, which is more than at 2 of Gupta et al. 's sites.

    We agree that our language misstates the point and is overly general. Gupta et al. found that using a reduced number of *labeled* image pairs (500–1,000) resulted in worse performance. In our study, we used 537 labeled images, even though the total number of available images -- both in our case and in Gupta et al.'s -- was higher. We intentionally limited the number of labeled images to evaluate model accuracy under more constrained conditions. Accordingly, we agree that the discussion of CNNs is not particularly helpful, as we did not explicitly evaluate CNN performance or its relationship to training size in our study. Therefore, we removed the references to CNNs and their sample size requirements.

11. **Section 4.4:** This section largely diverges from the central contribution of the study (a methodology to classify images) and into a lot of site-specific information that is largely conjecture about processes and reads as redundant to the prior section (4.3). This section could be eliminated.

    Thank you for your input. Upon review, we agree with your recommendation. We moved this section to an appendix and referenced it in the Discussion (l. 540, 590) and Conclusion (l. 615).

12. **Page 33, line 604:** Is there a citation for the claim that "these efforts have struggled to translate to IRES"? Neither study cited included nonperennial streams.

    We removed this statement, but we do say in l. 128 that "Gupta et al. (2022) and Noto et al. (2022) highlight the need to create algorithms focused on IRES."

13. **Section 4.6:** This section is only loosely connected with the central contribution of the paper (image classification model) and is material that could be included in the introductory material. This section could be eliminated.

We agree that the material in this section is better suited for partial incorporation into the Introduction, as well as inclusion in a new Conclusion section (in response to your comment below).

14. **Conclusion section is missing:** It is traditional to summarize the paper's contribution in a conclusion section, one is missing here.

We include a new Conclusion section that summarizes the overall contribution and incorporates salient points from the original Section 4.6 (in response to your comment above).

**Referee #2**

Overall Impression:
This manuscript presents a timely, and valuable study that addresses a critical challenge in hydrology: monitoring intermittent rivers and ephemeral streams (IRES). The application of a relatively simple logistic regression model to classify flow states from field camera imagery is both pragmatic and innovative. The methodology is clearly described, the results are robust and convincingly presented, and the discussion thoughtfully places the work in the broader context of IRES monitoring, climate change, and water management. The integration of image classifications for quality control of stage data is a particularly strong and practical contribution. The manuscript is generally well-written and structured. I believe it represents a significant contribution to the field and is a strong candidate for publication after revisions.

Thank you for your thoughtful review.

**Abstract**

The abstract effectively summarizes the study's motivation, methods, key findings, and implications. It clearly states the problem (monitoring challenges in IRES), the proposed solution (image classification), the location, and the broader significance of the work. However, I am including some comments and questions that can improve the abstract.

1. The abstract mentions the model was used for quality control of the stage time series. Could you briefly hint at the nature of the discrepancies/uncertainties/errors found (e.g., sensor drift, noise during high flow) to immediately highlight the practical utility of the Method?

Yes, we added the following to the abstract (l. 9-10): "We then used image classifications to perform quality control on the continuous stage time series, which allowed us to identify when the stream was dry and when the sensor malfunctioned."

2. The term "Imagey" in the title appears to be a typo for "Image-based." Was this Intentional?

"Imagery" was intentional, but upon review, we agree that "Image-based" more correctly emphasizes that we are using images as an input for classification, so we updated the title to "Image-based". Thank you.

3. The abstract focuses on categorical classification. Did the model's probability output itself provide any additional, continuous-like insight beyond the three discrete categories?

Although we did not explore the probability results beyond what was relevant to our categorical classifications (i.e., Figure 6), we did observe some overlap between 'low water' and 'high water' image classifications, and a range of corresponding probabilities assigned to these classes. For example, some photos labeled as "high water" were classified as "low water" and vice versa. Nevertheless, we ultimately evaluated only the probability distributions for 'any water' (the combination of low and high water; Figure 6).

**Introduction**

The introduction provides a comprehensive and compelling background, effectively building the case for the importance of IRES and the difficulties in monitoring them. The literature review is extensive and covers relevant areas, including remote sensing, citizen science, and various modeling approaches.

1. While you cover technological methods, could you briefly mention the organizational/funding challenges of maintaining sensor networks in remote IRES to further justify the need for low-cost methods?

Yes, we mention this in line 64: "In addition to inaccessibility, developing an in-situ monitoring network for stage and discharge on IRES is difficult because nascent gage networks may have less expertise, support, or funding compared to established national programs that generally focus on perennial streams (Vlah et al., 2023)."

2. You mention that deep learning has been mainly applied to perennial streams. Could you elaborate on one or two key reasons why these methods are particularly challenging to directly transfer to IRES (e.g., more dynamic channel geometry, greater debris, longer dry periods)?

We mention deep learning approaches in the Introduction (l. 117-128) and added this to Appendix A4 (l. 750-753): "Current deep learning models in the USGS Flow Photo Explorer (USGS, 2024) estimate relative flow states but cannot distinguish dry streambeds (Gupta et al., 2022; Goodling et al., 2025), potentially due to the dynamic channel morphology, shifting debris and vegetation, and ambiguous flow states of IRES – all of which can make training deep learning models challenging."

3. The introduction effectively sets up the use of machine learning with imagery. Would it be valuable to more explicitly state the core hypothesis: that visual features in daytime imagery are sufficient to reliably classify IRES flow states for monitoring purposes?

Thank you for this suggestion.  We replaced "Here, we explore the use of image classification for categorizing water levels in IRES" with "Here, we explore whether visual features in daytime imagery are sufficient to reliably classify IRES flow states for monitoring purposes" (l. 129-130).

4. Have you considered citing studies that discuss the hydrological significance of the "pooling" phase in IRES, which your method can detect but stage sensors cannot? For example, Stubbington, R., et al. (2017). The biota of intermittent rivers and ephemeral streams: aquatic and terrestrial assemblages. In Intermittent Rivers and Ephemeral Streams (pp. 217-245). Academic Press. could strengthen this point.

Thank you for highlighting the importance of the pooling phase in relation to biota in IRES. We discuss this later on in the Discussion section (4.1), and added your reference in that discussion (l. 505).

5. The transition from the broad introduction to the specific objectives of the study is clear, but could the final paragraph be slightly more structured to explicitly list the primary aims of the paper?

Yes, we add the hypothesis from your point #3 to this paragraph to make it more explicit (l. 129-130): "Here, we explore whether visual features in daytime imagery are sufficient to reliably classify IRES flow states for monitoring purposes".

**Methods**

The methods section is exceptionally detailed and reproducible, a major strength of the manuscript. The description of the study site, data sources, image preparation, model training, and validation is thorough. The handling of unbalanced classes and the development of a confidence metric are particularly sophisticated and commendable.

1. You limited image analysis to 9 am–4 pm PST to avoid low-light issues. Was any consideration given to using the camera's flash-illuminated nighttime images for a simple binary "water"/"no water" classification, given that this is a defining feature of IRES?

Yes, we address this directly in l. 178-180: "Nighttime images are poorly illuminated, with the camera's flash overexposing nearby vegetation and slightly illuminating the staff plate, making it difficult – if not impossible – to discern streambed conditions at night."

2. For the image cropping to 1000x1200 pixels, was this specific size determined empirically? Did you experiment with different crop sizes or aspect ratios to optimize feature recognition?

We describe the method for preparing images in Section 2.2.1. We added this to that section (l. 215-218): "To apply our method, all images were required to have the same dimensions. We

selected a resolution of 1,000 x 1,200 pixels (fig. 3a) because it was low enough to ensure that each image focused on the staff plate and streambed. Thus, the image size was determined by resolution constraints rather than through empirical or experimental testing." In addition, we added the following to the Discussion (l. 596-599): "For sites with consistent camera types and viewing angles, a useful exercise could be to find the optimal image resolution and cropping extent for feature recognition. In such an exercise, the cost of increased computing power for higher resolution images should be balanced with model performance."

3. The manual weighting scheme (3.5 for water categories, 3 for obstructed, 1 for 'no water') is interesting. Could you provide a sentence on the rationale behind these specific weight values?

Thank you for noting the omission of our reasoning behind the selection of manual weights. We added the following explanation to this section (l. 271-273): "We assigned manual weights to emphasize water presence categories ('high water' and 'low water') over 'no water,' and gave the 'obstructed' category a weight higher than 'no water' (reflecting its smaller sample size) but lower than the water categories, given its lesser importance."

4. The confidence level assignment is well-explained but based on qualitative assessment of probability distributions. Were any quantitative metrics (e.g., maximizing Youden's J index) explored to define the probability thresholds more objectively?

This is an interesting question, and we agree that we could have experimented with methods further in this case. However, because our primary objective was avoiding false positive classifications, the distribution (boxplot) assessment approach met our needs in this case. We agree, however, that a more objective strategy might be applicable in a generalized version of this study, and added the following in the Discussion (l. 603-604): "In addition, a more objective strategy for evaluating classification confidence for other sites could be developed."

5. You use soil moisture data from a nearby site (DRW) with probably different geology. How might this spatial disconnect influence the interpretation of the relationships between soil moisture and stage at PEC?

We discuss the differences in the DRW vs. PEC site, including soil features, in Section 4.3. We added the following to that section (l. 579-581): "Specifically, the soil hydrologic group at DRW is Group B, which indicates moderate infiltration rates, while the PEC watershed contains Groups C and D, which indicate slow or very slow infiltration rates, respectively (SSURGO, 2024)."

In addition, a section of uncertainties would be great, since I can see some sources of uncertainties in your work/modelling, for example: image quality and environmental variability (the classification model's performance is inherently tied to the quality and consistency of the input imagery), sensor data reliability and spatial mismatch (the "ground truth" data used for validation and comparison are themselves sources of uncertainty), limited training data and site specificity (the model was trained on a relatively small, manually labeled dataset (537

images) from a single site. This raises uncertainty about its performance when transferred to other IRES with different channel morphology, substrate, vegetation, and water clarity. the model's features (e.g., learned from the specific staff plate and rocks at pec) may be overly tailored to this unique location). Do not get me wrong, I still think there is a lot of value in publishing this paper, however, it is good to show the uncertainties and potential bias of the approach.

The issue of transferability to other sites was also raised by Reviewer #1, and we appreciate you also bringing in the related concept of uncertainty, and how that relates to transferability.

As noted in the response to referee #1, we have addressed this feedback through the incorporation of a new Appendix A4 "Transferability to other sites" that demonstrates transferability. The primary purpose of this work was to provide a proof of concept for this method using the Perry Creek site and its unique geographic context. While analysis of other sites in different locations and contexts would be beyond the scope of this paper, the new appendix nevertheless provides a parsimonious set of examples to demonstrate the method's transferability. In this appendix, we tested the model code (which is posted on the HydroShare repository associated with this paper) on two example sites from the USGS Flow Photo Explorer to produce time series of categorical flow states. These sites are Dry Brook Upper in Massachusetts and USGS streamgage 10247170 on Troy Creek in Nevada. This new appendix is referenced in the revised Discussion section at l. 608, and also references a new Appendix Figure 12, which presents categorical flow state time series from model predictions for both sites.

**Results**

The results are clearly presented, with appropriate use of tables, figures, and statistics. The model performance metrics are convincing, and the comparison between image classifications, observed stage, and modeled discharge is effective in demonstrating the value of the approach.

1. The confusion matrix shows that 'obstructed' images were most often misclassified as 'no water'. Given the ephemerality of the stream, do you think this misclassification might be functionally acceptable in many cases, as it likely reflects a true dry state?

Yes, this is a good point. We mention this in the Results section at l. 351-353 with: "In addition, 'obstructed' images were occasionally misclassified as either 'high water' or 'no water' (1.1% and 1.8% of these classifications, respectively). Due to the stream's ephemerality, it is likely there was in fact no water at PEC in the 'obstructed' images misclassified as 'no water'."

2. Figure 7/9 and the text describe how image classifications identified sensor malfunctions. How many erroneous data points would have been missed without this image-based quality control? A rough percentage or count would powerfully quantify this benefit.

We describe our quality control of data (i.e., removal of erroneous stage data) in Appendix 2; we also revised Section 3.3 to improve clarity based on suggestions from Reviewer #1. Therein, we did not specifically calculate how many erroneous data points would have been removed using some other method, e.g., simple visual inspection or use of a value threshold vs. image-based quality control, but instead note that the image-based approach could either replace other methods or augment them. Even so, we added to Section 3.3 (l. 417-418) that "we flagged and removed almost a month of these data from the record".

3. In Figure 10, the results show a notable discrepancy where the NWM reported zero discharge during periods of observed high water (e.g., Jan-Feb 2018). What is your leading hypothesis for this systematic underestimation by the NWM in this specific Catchment?

In Section 4.2, we discuss reasons for the discrepancies between the NWM, observed discharge, and image classifications that are shown in Figure 10. To respond further to your question, we added the following to Section 4.2 (l. 546-551): "Many 'high-water' image classifications occur during January - February 2018, when the NWM shows little discharge. This is illustrated by two manual discharge measurements from January 2018 (fig. 10), which record substantially higher flows at PEC than those simulated by the NWM. We hypothesize that the NWM struggles to represent early wet season flow processes in the steep slopes and low-infiltration soils of the PEC watershed. Later in the season, when soils in the PEC watershed are likely more saturated, the NWM discharge aligns more closely with PEC stage data and manual discharge measurements, suggesting that the NWM performs better under saturated conditions."

4. The relationship between stage and soil moisture at 5 cm is stronger than at 100 cm. Does this suggest that flow at PEC is primarily driven by shallow subsurface flow or saturation-excess overland flow rather than deeper groundwater contributions? If so, that should be discussed, showing how the changes over time may impact the local hydrology of the watershed and river.

Yes, we discuss the relationship between stage and soil moisture in Section 4.3 and added this to that discussion (l. 573-575): "This, combined with our understanding of the geology of the PEC watershed, suggests that runoff generation at PEC is primarily driven by shallow subsurface flow and saturation- or infiltration-excess overland flow."

5. You mention that high flows remain unmeasured due to safety. Could the image classifications be calibrated against the NWM output or other hydraulic models to provide a rough estimate of discharge during these extreme events?

We interpret this question as asking whether the images and NWM (or another model) could be used to estimate (continuous) discharge during extreme events, given the lack of observed discharge data. This would be beyond the scope of our study. Our model was not trained to predict continuous flow, as some previous studies have done (see references to Gupta et al.,

2022 in the Introduction and Discussion sections). Instead, our model predicts categorical flow states, which we then compared to NWM discharge values and the limited available discharge observations. Because we have very few observed discharge measurements (as described in Section 3.3 l. 439-441), we cannot calibrate or validate hydrologic model estimates of discharge. Consequently, linking image classifications to potentially uncertain modeled discharge estimates would not provide meaningful results in this case.

**Discussion**

The discussion successfully interprets the results, acknowledges limitations, and explores the wider implications of the work. The sections on unique site features and extensibility are particularly thoughtful and elevate the manuscript beyond a simple methods paper.

1. You rightly note that temporal correlation of flow states could be used for further quality control. Could a simple Hidden Markov Model be a natural next step to incorporate this temporal dependency?

Thank you for this question. There are a number of different methods that might be used to incorporate temporal dependency. Because evaluation of the suitability of methods for this is beyond the scope of this paper, we declined to list potential methods to avoid confusion.

2. How does the performance of your logistic regression model (91% accuracy) compare, in your view, to the potential trade-offs of using a more complex but data-hungry model like a CNN for this specific task? That could be a good paragraph in the discussion, showing the drawbacks and positives sides of using a parsimonious but effective model.

In Section 4.1, l. 481 we describe that our method "prioritizes a simple, accurate, site-specific model that requires minimal manually-labeled training data." In accordance with this suggestion, as well as prior feedback from both you and Reviewer #1, we included a discussion of potential applications of our model, including in conjunction with CNN models, in Appendix A4.

3. In the biggening (abstract and objectives) you state the method is "transferable." Could you specify the primary condition for transferability (e.g., the presence of a staff plate or a consistent field of view of the streambed)? Maybe a bit of discussion on the costs of this equipment set up would also help the reader to have an idea of how much it would cost. Perhaps that could be included in the methods?!

In accordance with suggestions made by you and Reviewer #1, we added Appendix A4 that specifically addresses transferability (see above). Furthermore, with respect to the cost of setup, we added Appendix A5 that details our setup costs, and reference that appendix in Section 4.4, l. 608.

4. The discussion on the potential for subsurface flow bypassing the PEC site is fascinating. Could this hypothesis be further supported by comparing the water level in

Lake Mendocino with dry/wet periods at PEC?

We agree that this hypothesis could certainly be explored further, and observations at PEC and other sites in the same watershed suggest that exploration may be worthwhile. However, that exploration is outside the scope of this work.

5. You mention that your code is transferable. What is the minimum number of manually labeled images you would estimate is necessary to achieve reasonable performance at a new, similar site?

Based on suggestions made by you and Reviewer #1, we added Appendix A4 that specifically addresses transferability, including a discussion of performance with a minimal number of manually labeled images (see above).

6. In the context of climate change, how might your method help in detecting shifts in the timing of flow initiation and cessation in IRES, which is a key impact of warming Temperatures?

As described in Section 4.1, this method is well-suited for IRES since it focuses on water presence or absence. Therefore, it is well-suited to identify the timing of flow initiation and cessation in IRES. We made this point more explicit by adding the following to this same Discussion section (l. 497-499): "our method could be expanded to detect IRES-relevant states including wet streambeds, isolated pools, standing water, trickling water, snow, or ice." We discuss the context of climate change in the Introduction, and describe how our method supports IRES monitoring under climate change in the Conclusion.

7. Have you considered referencing studies that have successfully implemented low-cost, image-based methods in data-scarce regions? For instance,
- Noto, S., Tauro, F., Petroselli, A., Apollonio, C., Botter, G., & Grimaldi, S. (2022). Low-cost stage-camera system for continuous water-level monitoring in ephemeral streams. Hydrological Sciences Journal, 67(9), 1439–1448.
https://doi.org/10.1080/02626667.2022.2079415
Yes, we cite this work.

- Rodrigues, R. M., Braga, B. B., & Costa, C. A. G. (2025). Efficiency in river discharge measurement: combining Chiu's method with particle image velocimetry techniques. RBRH, 30, e31.
We added this citation to the introduction section (l. 112-114).

These papers could broaden the perspective on transferability and cost-effectiveness.

**Conclusion**

The conclusion effectively summarizes the main findings and their significance. It

compellingly argues for the role of this low-cost method in improving IRES monitoring and, consequently, water management in a changing climate.

1. Could the conclusion more explicitly state the single most important recommendation for water managers seeking to implement this method?

In response to this and feedback from Reviewer #1, we added a new Conclusion section that summarizes the overall contribution and incorporates discussion of practical recommendations (l. 625).

2. Beyond FIRO, can you speculate on one other specific water management decision (e.g., environmental flow allocations, drought contingency planning) that would benefit from the categorical flow data your method provides?

Yes, our new Conclusion section discusses monitoring for salmon migration (l. 618).

3. While the method is low-cost, the conclusion could acknowledge the ongoing costs and challenges of maintaining field cameras in harsh environments as a consideration for long-term deployment.

Yes, our new Conclusion section recommends including a budget for site maintenance in all long-term monitoring programs (l. 629).

4. What do you see as the next critical technological or methodological advancement needed to make IRES monitoring truly scalable across vast river networks?

While the broader topic of scaling IRES monitoring approaches like ours is beyond the scope of our paper, we addressed this in the Conclusion (l. 629-631) and Appendix A4 (l. 750-760).

I congratulate the authors on an excellent piece of work. I hope that with these revisions, the manuscript will be an even stronger contribution to the literature.

Thank you for your thorough review. The revisions based on your feedback will undoubtedly strengthen the manuscript.

---

## Referee Report (RR1)

Referee #2

Dear Editors,

The authors have addressed the challenges of hydrological monitoring in these environments with clarity, rigor, and creativity. I would like to commend the authors for their thoughtful and detailed responses to reviewer comments and their willingness to substantially revise the manuscript. The improvements in clarity, methodological detail, and contextual discussion are evident throughout, and the new appendices addressing transferability and uncertainty are particularly appreciated.

While the manuscript is now in excellent shape, I would like to offer a few minor suggestions that, if addressed, could further strengthen the work:

**1. Quantifying Benefits of Image-Based Quality Control**

The manuscript describes the important role of image classification in quality control of sensor data. However, it would be helpful to briefly quantify the impact—such as an approximate number or percentage of erroneous data points identified and removed thanks to this approach. This would concretely demonstrate the practical utility of the method.

**2. Discussion of Model Transferability Conditions**

The new appendix on transferability is a strong addition. To further aid practitioners, please consider explicitly summarizing in the main text the primary conditions and limitations for successfully transferring the method to other sites, such as the need for consistent camera positioning, presence of a staff plate, or minimum number of labeled images.

**3. Objective Thresholding for Classification Confidence**

The qualitative approach to determining classification confidence is reasonable and well-explained. For completeness, a brief mention of potential objective, quantitative methods (e.g., maximizing a statistical metric like Youden's J index) could be included in the Discussion, to guide future work in this area.

**4. Ongoing Operational Considerations**

While the manuscript emphasizes the low-cost nature of the method, it would be valuable to briefly acknowledge in the conclusion or discussion the practical challenges and ongoing costs associated with long-term field camera maintenance, especially in harsh or remote environments. This would help set realistic expectations for practitioners considering deployment at scale.

In closing, I wish to thank the authors for their thoughtful revisions and for their clear commitment to advancing hydrological science. I am confident that the manuscript, with these final minor improvements, will be a significant and widely appreciated contribution to the literature.

Sincerely,

Referee #2

---

## Author Response (AR2)

**Point-by-Point Reply to Comments**

Referee #2 and the editor requested minor revisions. The revised manuscript includes minor changes to a few sentences and a couple of additional sentences to address suggestions from Referee #2. Our point-by-point response is organized as a response to the comments from Referee #2, with our responses in blue.

**Response to Referee #2**

Dear Editors,

The authors have addressed the challenges of hydrological monitoring in these environments with clarity, rigor, and creativity. I would like to commend the authors for their thoughtful and detailed responses to reviewer comments and their willingness to substantially revise the manuscript. The improvements in clarity, methodological detail, and contextual discussion are evident throughout, and the new appendices addressing transferability and uncertainty are particularly appreciated.

While the manuscript is now in excellent shape, I would like to offer a few minor suggestions that, if addressed, could further strengthen the work:
Thank you for this thoughtful review of the manuscript. We will address your suggestions below.

1. Quantifying Benefits of Image-Based Quality Control
The manuscript describes the important role of image classification in quality control of sensor data. However, it would be helpful to briefly quantify the impact—such as an approximate number or percentage of erroneous data points identified and removed thanks to this approach. This would concretely demonstrate the practical utility of the method.
Thank you for this feedback. The image classifications were used in tandem with standard quality control. To make the impact of the image classifications clearer, we present figure 9 and then add a sentence describing how the image classifications supported quality control (l. 406-410): "The final corrected and quality-controlled PEC stage time series (fig. 9) is the product of standard quality control (i.e. removal of stage observations taken during sensor maintenance) and using image classifications to support quality control. Specifically, image classifications helped identify when the stage was zero for August 2017 to September 2023, supported the removal of erroneous data for most of January 2023, and revealed that stage observations are likely artificially low from late 2017 to early 2018."

2. Discussion of Model Transferability Conditions
The new appendix on transferability is a strong addition. To further aid practitioners, please consider explicitly summarizing in the main text the primary conditions and limitations for successfully transferring the method to other sites, such as the need for consistent camera positioning, presence of a staff plate, or minimum number of labeled images.

Thank you for this feedback. We provide recommendations to practitioners (including the need for consistent camera positioning) in the final paragraph of the Conclusion section (l. 626-634), with supporting information in Appendices 4 and 5. We added a sentence to the Conclusion to address that a staff plate is not needed for image classification and that as few as 100 labeled images may be sufficient for basic image classification (l. 630-631): "For the classification of categorical flow states, installation of a staff plate is not necessary, and basic image classification can be achieved with a limited number of labeled photos (on the order of 100 per site; see Appendix A4."

3. Objective Thresholding for Classification Confidence
The qualitative approach to determining classification confidence is reasonable and well-explained. For completeness, a brief mention of potential objective, quantitative methods (e.g., maximizing a statistical metric like Youden's J index) could be included in the Discussion, to guide future work in this area.
Thank you for this feedback. In line 604, we mention "In addition, a more objective strategy for evaluating classification confidence for other sites could be developed." We agree that quantitative approaches, such as maximizing a statistical metric (e.g., Youden's J index), could be used in place of the qualitative approach adopted here. However, the selection of a specific quantitative method depends on the objectives of the analysis and the characteristics of the site(s) under study. A systematic evaluation of such objective strategies is therefore beyond the scope of this work.

4. Ongoing Operational Considerations
While the manuscript emphasizes the low-cost nature of the method, it would be valuable to briefly acknowledge in the conclusion or discussion the practical challenges and ongoing costs associated with long-term field camera maintenance, especially in harsh or remote environments. This would help set realistic expectations for practitioners considering deployment at scale.
Thank you for this comment. We address this in the last paragraph of the revised Discussion (l. 608-609), where we reference Appendix 5, which provides estimates of the cost of setting up and maintaining a field camera site. In addition, we provide practical recommendations for implementing our method in the last paragraph of the Conclusion (l. 626-634).

In closing, I wish to thank the authors for their thoughtful revisions and for their clear commitment to advancing hydrological science. I am confident that the manuscript, with these final minor improvements, will be a significant and widely appreciated contribution to the literature.
Thank you for your thorough review and for helping to improve the quality of our manuscript.

Sincerely,
Referee #2